# Hyperspherical Normalization for Scalable Deep Reinforcement Learning

Hojoon Lee [1] [*]  Youngdo Lee [1] [*]  Takuma Seno [2]  Donghu Kim [1]  Peter Stone [2] [3]  Jaegul Choo [1]

## Abstract

Scaling up the model size and computation has brought consistent performance improvements in supervised learning. However, this lesson often fails to apply to reinforcement learning (RL) because training the model on non-stationary data easily leads to overfitting and unstable optimization. In response, we introduce SimbaV2, a novel RL architecture designed to stabilize optimization by (i) constraining the growth of weight and feature norm by *hyperspherical normalization*; and (ii) using a distributional value estimation with reward scaling to maintain stable gradients under varying reward magnitudes. Using the soft actor-critic as a base algorithm, SimbaV2 scales up effectively with larger models and greater compute, achieving state-of-the-art performance on 57 continuous control tasks across 4 domains. The code is available at *dojeon-ai.github.io/SimbaV2*.

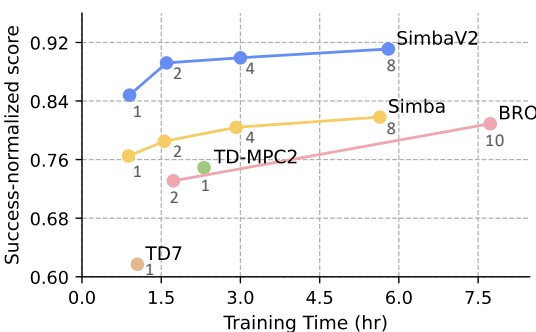

*Figure 1.* **Compute vs RL Performance.** Performance scales with increased compute when using Soft Actor-Critic with SimbaV2 architecture, outperforming other state-of-the-art RL algorithms. SimbaV2 achieves 0.848 normalized return with an update-to-data (UTD) ratio of 1, surpassing TD-MPC2 (0.749 at UTD=1), Simba (0.818 at UTD=8), and BRO (0.807 at UTD=8). Grey numbers below each point indicate the UTD ratio. Results are averaged over 57 continuous control tasks from MuJoCo, DMC, MyoSuite, and HumanoidBench, each trained on 1 million samples.

## 1. Introduction

Over the past decade, a scaling law has emerged as the cornerstone of supervised learning (SL), suggesting that increasing model size, compute, and data consistently improve performance (Kaplan et al., 2020; Dehghani et al., 2023). This paradigm has driven significant breakthroughs, from large language models (Gemini et al., 2023; Achiam et al., 2023) to diffusion models (Ramesh et al., 2021; Rombach et al., 2022), where bigger models reliably translate to better performance.

In contrast, scaling laws often fail to apply in reinforcement learning (RL) (Song et al., 2019; Li et al., 2023). Unlike SL's static data distributions, RL agents must contend with continuously evolving data distributions and shifting objectives throughout their training process (Sutton & Barto, 2018). This fundamental non-stationarity creates a scaling paradox: increasing model capacity or computational re-

sources frequently leads to overfitting to earlier experiences and reduced adaptability to new tasks (Lyle et al., 2022; Dohare et al., 2023).

One of the root causes of RL's scaling challenges lies in uncontrolled norm growth during training, which destabilizes optimization in multiple ways. Feature norms grow uncontrollably due to the implicit bias of TD loss (Kumar et al., 2022), where dominant dimensions cause overfitting and loss of plasticity (Lyle et al., 2022; Ma et al., 2023). Parameter norms grow unbounded, reducing effective learning rates (gradient-to-parameter ratio) and making weight updates increasingly difficult (Dohare et al., 2023; Lyle et al., 2024). Gradient norms fluctuate due to varying reward scales and outliers, further disrupting optimization. These instabilities are compounded with an increased model size or update frequency, making RL harder to scale than SL.

Previous work has addressed these norm instabilities through separate, isolated approaches. Normalization layers such as $\ell_2$-normalization (Bjorck et al., 2021; Hussing et al., 2024), layer normalization (Lei Ba et al., 2016; Lyle et al., 2023), and RL-specific variants (Bhatt et al., 2024; Lee et al., 2024c) control the growth of feature norm. Weight decay (Farebrother et al., 2018) manages the growth of param-

---

[*]Equal contribution  [1]KAIST  [2]Sony AI  [3]UT Austin. Correspondence to: Hojoon Lee <joonleesky@kaist.ac.kr>.

*Proceedings of the 42nd International Conference on Machine Learning*, Vancouver, Canada. PMLR 267, 2025. Copyright 2025 by the author(s).

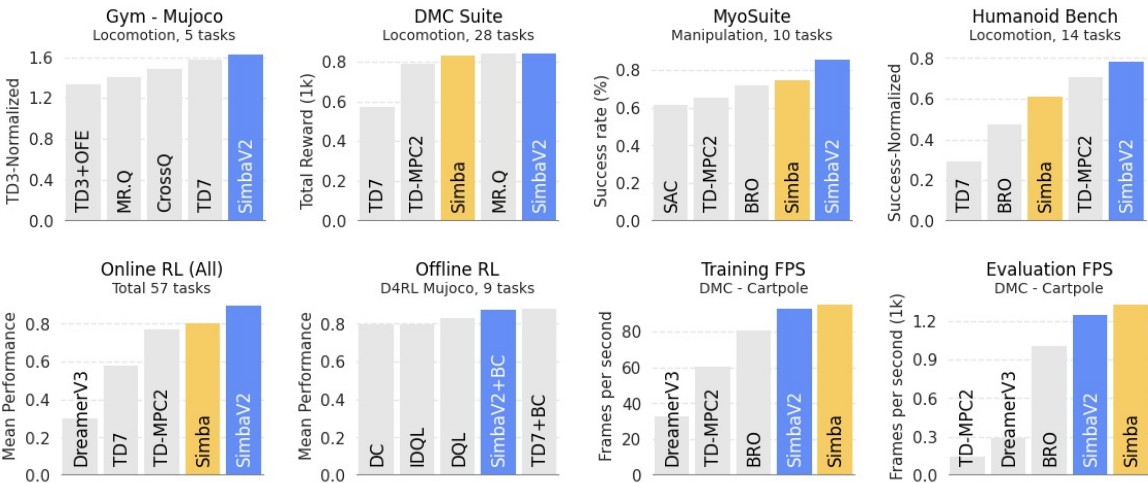

*Figure 2.* **Benchmark Summary. (a)** SimbaV2, with an update-to-data (UTD) ratio of 2, outperforms state-of-the-art RL algorithms across diverse continuous control benchmarks using fixed hyperparameters across all domains. **(b)** SimbaV2 delivers competitive performance in both online and offline RL while requiring significantly less training computation and offering faster inference times.

eter norm. Reward scaling and cross-entropy loss (Schaul et al., 2021; Farebrother et al., 2024) were adopted to control gradient norm fluctuations. However, these techniques are applied individually without a unified framework, making coordination and scaling difficult. Periodic weight reinitialization (Nikishin et al., 2022; D'Oro et al., 2023; Schwarzer et al., 2023) offers an alternative by completely retraining networks periodically. While effective, this approach requires additional training time and causes sharp performance drops, making it impractical for safety-critical applications.

In response, we present SimbaV2, a novel RL architecture that addresses these challenges by simultaneously stabilizing weight, feature, and gradient norms within a unified framework. Building on the Simba architecture (Lee et al., 2024c), which uses pre-layernorm residual blocks (Xiong et al., 2020) and weight decay (Krogh & Hertz, 1991), SimbaV2 introduces three key modifications:

- **Hyperspherical Feature Normalization:** We replace all layer normalization with hyperspherical normalization ($\ell_2$-normalization).

- **Hyperspherical Weight Normalization:** We remove weight decay and instead project weights onto the unit-norm hypersphere after each gradient update (Loshchilov et al., 2024). Combined with hyperspherical feature normalization, this ensures consistent effective learning rates across layers and eliminates the need for weight regularization tuning.

- **Distributional Value Estimation with Reward Scaling:** To address unstable gradient norms caused by varying reward scales and outliers, we integrate a distributional critic (Bellemare et al., 2017) and apply reward scaling to maintain unit variance of the target throughout training.

Using Soft Actor-Critic (Haarnoja et al., 2018) as our base algorithm, SimbaV2 effectively stabilizes all three types of norms while maintaining consistent effective learning rates throughout training (Section 5.2 and Figure 4). We evaluated SimbaV2 on four standard online RL benchmarks: MuJoCo (Todorov et al., 2012), DMC Suite (Tassa et al., 2018), MyoSuite (Caggiano et al., 2022), and Humanoid-Bench (Sferrazza et al., 2024); as well as the D4RL MuJoCo benchmark (Fu et al., 2020) for offline RL. As shown in Figures 1 and 2, SimbaV2 achieves state-of-the-art performance without requiring algorithmic modifications or hyperparameter tuning, and scales effectively with increased model size and computation without using periodic reinitialization.

## 2. Related Work

### 2.1. Regularization in Deep Reinforcement Learning

Deep RL is particularly susceptible to overfitting due to its inherently non-stationary optimization process (Song et al., 2019). To address overfitting, researchers have adapted regularization techniques from SL, including weight decay (Farebrother et al., 2018), dropout (Hiraoka et al., 2021), various normalization layers (Gogianu et al., 2021; Bjorck et al., 2021; Lyle et al., 2023; Gallici et al., 2024; Bhatt et al., 2024; Lee et al., 2024c; Elsayed et al., 2024; Palenicek et al., 2025), and mixture of expert (Obando-Ceron et al., 2024; Willi et al., 2024). However, these methods often prove insufficient when scaling RL models, as larger computational resources and increased model sizes can easily exacerbate overfitting (Li et al., 2023; Nauman et al., 2024a).

To further scale computations and model sizes in RL, recent studies have explored periodic weight reinitialization strategies to rejuvenate learning and escape local minima (D'Oro

et al., 2023; Nauman et al., 2024b). These strategies include reinitializing weights to their initial distributions (Nikishin et al., 2022), interpolating between random and current weights (Xu et al., 2023; Schwarzer et al., 2023), utilizing momentum networks (Lee et al., 2024b), and selectively reinitializing dormant weights (Sokar et al., 2023). While promising, reinitialization has a notable limitation: it can lead to the loss of useful information and incur significant computational overhead as model size increases.

To address these limitations, we introduce SimbaV2, an architecture that explicitly constrains parameter, feature, and gradient norms throughout training. By constraining norms through hyperspherical normalization, SimbaV2 stabilizes an optimization process and eliminates the need for weight decay or periodic weight reinitialization.

## 2.2. Hyperspherical Representations in Deep Learning

Hyperspherical representations are widely used in deep learning across image classification (Salimans & Kingma, 2016; Liu et al., 2017b), face recognition (Wang et al., 2017; Liu et al., 2017a), variational autoencoders (Xu & Durrett, 2018), and contrastive learning (Chen et al., 2020). Using spherical embeddings is known to enhance feature separability (Wang & Isola, 2020), improving performance in tasks requiring precise discrimination. Recently, researchers have applied the hyperspherical normalization to intermediate features and weights to stabilize training in large-scale models such as diffusion models (Karras et al., 2024) and transformers (Loshchilov et al., 2024).

In this work, we apply hyperspherical normalization to RL. Unlike previous studies that focus on training the network on stationary data distributions with discrete inputs and outputs, we demonstrate their effectiveness on non-stationary data distributions with continuous inputs and outputs.

# 3. Preliminaries

As background, we briefly explain the Soft Actor-Critic (SAC) algorithm (Haarnoja et al., 2018) and the Simba architecture (Lee et al., 2024c).

## 3.1. Soft Actor Critic

SAC is a prominent off-policy algorithm for continuous control. It aims to maximize both expected cumulative reward and policy entropy, where $\tau = (o, a, r, o')$ represents a transition tuple. SAC comprises a stochastic policy $\pi_\theta(a|o)$, a Q-function $Q_\phi(o, a)$, and an entropy coefficient $\alpha$ that balances reward and entropy.

The policy network is optimized to maximize the expected return while encouraging entropy, which is formalized as:

$$\mathcal{L}_\pi = \mathbb{E}_{\bar{a} \sim \pi_\theta} \left[ \alpha \log \pi_\theta(\bar{a}|o) - Q_\phi(o, \bar{a}) \right]. \quad (1)$$

The Q-function $Q_\phi(o, a)$ is trained to minimize the Bellman residual loss:

$$\mathcal{L}_Q = (Q_\phi(o, a) - (r + \gamma Q_{\bar{\phi}}(o', a') - \alpha \log \pi_\theta(a'|o')))^2, \quad (2)$$

where $a' \sim \pi_\theta(\cdot|o')$, $\gamma \in [0, 1)$ is the discount factor, and $Q_{\bar{\phi}}$ represents the target Q-network updated via an exponential moving average of $\phi$.

## 3.2. Simba Architecture

Simba (Lee et al., 2024c) is an RL architecture with normalization layers composed of the following stages:

**Input Embedding.** Given an input observation $o_t \in \mathbb{R}^{|\mathcal{O}|}$, Simba applies Running Statistics Normalization (RSNorm) to normalize each dimension to zero mean and unit variance.

At each timestep $t$, the running mean $\mu_t \in \mathbb{R}^{|\mathcal{O}|}$ and variance $\sigma^2 \in \mathbb{R}^{|\mathcal{O}|}$ are updated recursively as:

$$\mu_t = \mu_{t-1} + \frac{1}{t} \delta_t, \quad \sigma_t^2 = \sigma_{t-1}^2 + \frac{1}{t} (\delta_t^2 - \sigma_{t-1}^2) \quad (3)$$

where $\delta_t = o_t - \mu_{t-1}$.

Given running statistics, the observation is normalized as:

$$\bar{o}_t = \text{RSNorm}(o_t) = \frac{o_t - \mu_t}{\sqrt{\sigma_t^2 + \epsilon}}. \quad (4)$$

Then, the normalized observation, $\bar{o}_t$, is embedded with a linear layer $W_h^0 \in \mathbb{R}^{|\mathcal{O}| \times d_h}$ defined as:

$$h_t^0 = W_h^0 \bar{o}_t. \quad (5)$$

**Latent Encoding.** Next, the embedding $h_t^0$ is encoded by a stack of $L$ residual blocks with pre-layer normalization. For $l \in \{1, \ldots, L\}$, each of the $l$-th block is defined as:

$$h_t^l = h_t^{l-1} + \text{MLP}(\text{LayerNorm}(h_t^{l-1})) \quad (6)$$

After the final block, the output is normalized again to obtain the latent feature:

$$z_t = \text{LayerNorm}(h_t^L). \quad (7)$$

**Output Prediction:** Finally, to predict the policy or Q-value, a linear layer $W_o \in \mathbb{R}^{d_h \times d_o}$ maps $z_t$ to:

$$p_t = W_o z_t. \quad (8)$$

# 4. SimbaV2

SimbaV2 builds on Simba by adding constraints on weights, features, and gradients to enhance training stability, particularly when scaling to larger models and more computation. The modifications include:

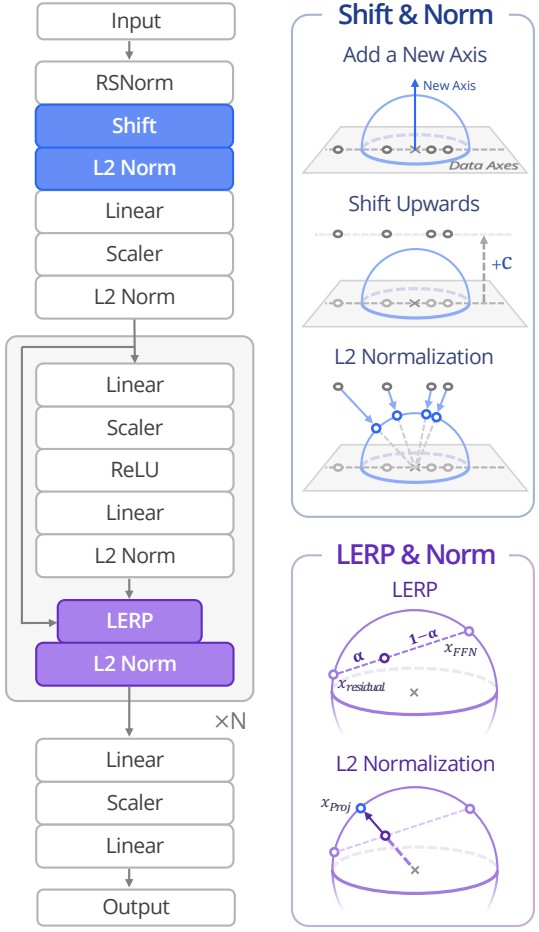

*Figure 3.* **SimbaV2 architecture.** The input observation is first normalized using running statistics, then shifted along a new axis with a constant $c_{\text{shift}}$ to preserve magnitude information before being projected onto the unit hypersphere. The projected observation is passed through a linear layer, followed by a series of non-linear blocks and refined with LERP, serving as a residual connection. A final linear layer predicts the policy or value function.

- **LayerNorm $\rightarrow$ $\ell_2$-Norm**: Layer normalization is replaced with $\ell_2$-normalization, constraining intermediate features to have unit norm.

- **Linear $\rightarrow$ Linear + Scaler**: Standard linear layer is decoupled into a linear layer with weights constrained to a unit norm hypersphere, without a bias, and a learnable scaling vector that performs element-wise scaling.

- **Residual Connection $\rightarrow$ LERP**: Residual connection is replaced with a learnable linear interpolation (LERP), which combines raw and transformed features via a learnable interpolation vector.

- **Weight Decay $\rightarrow$ Weight Projection**: Weight decay is replaced with direct weight projection onto the unit hypersphere after each gradient update.

- **MSE Loss $\rightarrow$ KL-divergence Loss**: MSE-based Bellman loss is replaced with KL-divergence loss, using a categorical critic (Bellemare et al., 2017).

- **No Reward Scaling $\rightarrow$ Reward Scaling**: Rewards are normalized with running statistics to stabilize the scale of both actor loss (Equation.1) and critic loss (Equation.2).

In the following subsections, we describe these modifications in detail.

### 4.1. Input Embedding

Following Simba, SimbaV2 first standardize the raw observations $\boldsymbol{o}_t \in \mathbb{R}^{|\mathcal{O}|}$ using RSNorm, yielding $\bar{\boldsymbol{o}}_t$. To further stabilize training, we map $\bar{\boldsymbol{o}}_t$ onto the unit hypersphere before applying a linear layer.

**Shift + $\ell_2$-Norm.** Direct $\ell_2$-normalization can discard magnitude information (e.g., $\bar{\boldsymbol{o}}_t = [1, 0]$ and $[2, 0]$ both map to $[1, 0]$). To retain magnitude information, we embed $\bar{\boldsymbol{o}}_t$ into an $(|\mathcal{O}| + 1)$-dimensional vector by concatenating a positive constant $c_{\text{shift}} > 0$, then apply $\ell_2$-normalization:

$$\tilde{\boldsymbol{o}}_t = \ell_2\text{-Norm}\big([\bar{\boldsymbol{o}}_t; c_{\text{shift}}]\big). \quad (9)$$

As illustrated in Figure 3, this additional coordinate encodes the original norm of $\bar{\boldsymbol{o}}_t$, preserving magnitude information.

**Linear + Scaler.** We then embed $\tilde{\boldsymbol{o}}_t$ using a linear layer $\boldsymbol{W}_h^0 \in \mathbb{R}^{(|\mathcal{O}|+1) \times d_h}$ and a scaling vector $\boldsymbol{s}_h^0 \in \mathbb{R}^{d_h}$ as:

$$\boldsymbol{h}_t^0 = \ell_2\text{-Norm}(\boldsymbol{s}_h^0 \odot (\boldsymbol{W}_h^0 \, \text{Norm}(\tilde{\boldsymbol{o}}_t)). \quad (10)$$

where the $\ell_2$-normalization projects back to the hypersphere.

### 4.2. Feature Encoding

Starting from the initial hyperspherical embedding $\boldsymbol{h}_t^0$, we apply $L$ consecutive blocks of non-linear transformations. Each $l$-th block transforms $\boldsymbol{h}_t^l$ into $\boldsymbol{h}_t^{l+1}$ as follows:

**MLP + $\ell_2$-Norm.** Each block uses an inverted bottleneck MLP (Vaswani, 2017) followed by $\ell_2$-normalization to project the output back onto the unit hypersphere.

$$\tilde{\boldsymbol{h}}_t^l = \ell_2\text{-Norm}(\boldsymbol{W}_{h,2}^l \, \text{ReLU}\big((\boldsymbol{W}_{h,1}^l \, \boldsymbol{h}_t^l) \odot \boldsymbol{s}_h^l\big)). \quad (11)$$

where $\boldsymbol{W}_{h,1}^l \in \mathbb{R}^{4d_h \times d_h}$ and $\boldsymbol{W}_{h,2}^l \in \mathbb{R}^{d_h \times 4d_h}$ are weight matrices, and $\boldsymbol{s}_h^l \in \mathbb{R}^{4d_h}$ is a learnable scaling vector.

**LERP + $\ell_2$-Norm.** We then linearly interpolate between the original input $\boldsymbol{h}_t^l$ and its non-linearly transformed output $\tilde{\boldsymbol{h}}_t^l$, followed by another $\ell_2$-normalization:

$$\boldsymbol{h}_t^{l+1} = \ell_2\text{-Norm}((\boldsymbol{1} - \boldsymbol{\alpha}^l) \odot \boldsymbol{h}_t^l + \boldsymbol{\alpha}^l \odot \tilde{\boldsymbol{h}}_t^l). \quad (12)$$

where $\boldsymbol{1} \in \mathbb{R}^{d_h}$ and $\boldsymbol{\alpha}^l \in \mathbb{R}^{d_h}$ are one vector and a learnable interpolation vector, respectively.

LERP acts analogous to a learnable residual connection but can also be viewed as a first-order approximation of a Riemannian retraction on the hypersphere (Absil et al., 2008). Please refer to Appendix A.1 for further discussion.

## 4.3. Output Prediction

We use a linear layer to parameterize both the policy distribution and Q-value. Because Simba's single Q-value estimate with an MSE-based Bellman loss is susceptible to outliers, we adopt a categorical critic with KL-divergence loss (Bellemare et al., 2017), which provides smoother gradients and more stable optimization (Imani & White, 2018).

**Distributional Critic.** We represent the Q-value as a categorical distribution over a discrete set of returns:

$$\left\{ \delta_i = G_{\min} + (i-1) \frac{G_{\max} - G_{\min}}{n_{\text{atom}} - 1} \mid i = 1, ..., n_{\text{atom}} \right\},$$
(13)

where $G_{\min}$ and $G_{\max}$ denote the minimum and maximum possible returns, and $n_{\text{atom}}$ is the number of discrete atoms.

Given the encoded representation $\boldsymbol{h}_t^L$, we compute unnormalized logits $\boldsymbol{z}_t \in \mathbb{R}^{|\mathcal{A}| \times n_{\text{atom}}}$ for all actions as follows:

$$\boldsymbol{z}_t = \boldsymbol{W}_{o,2}((\boldsymbol{W}_{o,1}\,\boldsymbol{h}_t^L) \odot \boldsymbol{s}_o),$$
(14)

where $\boldsymbol{W}_{o,1} \in \mathbb{R}^{d_h \times d_h}$, $\boldsymbol{W}_{o,2} \in \mathbb{R}^{|\mathcal{A}| \times n_{\text{atom}} \times d_h}$, and $\boldsymbol{s}_o \in \mathbb{R}^{d_h}$ are trainable parameters.

For each action $\boldsymbol{a} \in \mathcal{A}$, the categorical probability is represented by applying the softmax function to $\boldsymbol{z}_{t,\boldsymbol{a}} \in \mathbb{R}^{n_{\text{atom}}}$:

$$p_{t,\boldsymbol{a}} = \text{softmax}(\boldsymbol{z}_{t,\boldsymbol{a}}).$$
(15)

The resulting Q-value is the expected return under $p_{t,\boldsymbol{a}}$:

$$Q(\boldsymbol{o}_t, \boldsymbol{a}) = \sum_{i=1}^{n_{\text{atom}}} \delta_i\,p_{t,\boldsymbol{a},i}.$$
(16)

**Reward Bounding and Scaling.** To use a categorical critic, we first bound the target returns within $[G_{\min}, G_{\max}]$ and then scale the reward to maintain unit variance, ensuring stable gradients for both the actor and the critic. Unlike previous work (Schaul et al., 2021), which scaled the critic loss, we scale the reward itself, affecting both components simultaneously. Moreover, unlike observation normalization, we do not center the reward, as shifting the reward can alter the optimal policy in episodic tasks (Naik et al., 2024).

Given a reward $r_t$ at time $t$ and a discounted factor $\gamma$, we track a running discounted return:

$$G_t \leftarrow \gamma G_{t-1} + r_t$$
(17)

where $G_t$ is re-initialized to 0 at the start of each episode.

Then, we track the running variance of $G_t$, denoted as $\sigma_{t,G}^2$ and maintain a running maximum:

$$G_{t,\max} \leftarrow \max(G_{t,\max}, G_t).$$
(18)

We then scale the reward as follows:

$$\bar{r}_t \leftarrow \frac{r_t}{\max(\sqrt{\sigma_{t,G}^2 + \epsilon},\ G_{t,\max}/G_{\max})}.$$
(19)

This formula stabilizes gradients for both high-variance and low-variance returns, while thresholding with $G_{t,\max}/G_{\max}$ ensures target returns remain within $[G_{\min}, G_{\max}]$.

## 4.4. Initialization and Update

In this subsection, we outline how weight matrix $\boldsymbol{W}$, scaler $\boldsymbol{s}$, and interpolation vector $\boldsymbol{\alpha}$ are initialized and updated.

**Weight.** All weight vectors are initialized orthogonally and then projected onto the unit hypersphere which forms an orthonormal basis. At each gradient step, we re-project them onto the unit sphere to maintain unit norm.

Formally, let $\boldsymbol{W}$ be the weight matrix before the update, and let $\mathcal{L}$ denote the loss function. The update rule is defined as:

$$\boldsymbol{W} \leftarrow \ell_2\text{-Norm}(\boldsymbol{W} - \eta \frac{\partial \mathcal{L}}{\partial \boldsymbol{W}})$$
(20)

where $\eta > 0$ is a learning rate and $\ell_2$-Norm is the $\ell_2$-normalization operator along the embedding axis.

**Scaler.** Following Loshchilov et al. (2024), we decouple the initialization scale of $\boldsymbol{s}$ from its learning dynamics by using two scalars, $\boldsymbol{s}_{\text{init}}$ and $\boldsymbol{s}_{\text{scale}}$. Although $\boldsymbol{s}$ is initialized to $\boldsymbol{s}_{\text{scale}}$, it behaves as if it was initialized to $\boldsymbol{s}_{\text{init}}$ during the forward pass by:

$$\boldsymbol{s} \leftarrow \boldsymbol{s}_{\text{scale}} \odot (\boldsymbol{s}_{\text{init}} \oslash \boldsymbol{s}_{\text{scale}})$$
(21)

where $\odot$ and $\oslash$ are element-wise product and division, respectively. This formulation lets $\boldsymbol{s}_{\text{scale}}$ control the learning rate of $\boldsymbol{s}$ independently from the global learning rate $\eta$.

When both the feature vector $\boldsymbol{h} \in \mathbb{R}^{d_h}$ and the randomly orthonormal initialized weight matrix $\boldsymbol{W} \in \mathbb{R}^{d_h \times d_h}$ lie on the unit hypersphere, each component of $\boldsymbol{W}\boldsymbol{h} \in \mathbb{R}^{d_h}$ can be approximated by $\cos(\theta)$ with $\mathbb{E}_\theta[\cos^2(\theta)] = 1/2$. Therefore, we set $\boldsymbol{s}_{\text{init}} = \boldsymbol{s}_{\text{scale}} = (\sqrt{2/d_h})\,\boldsymbol{1}$ to maintain unit norm after scaling at initialization. A detailed derivation is in Appendix A.2.

**Interpolation vector.** Analogous to the scaler, the interpolation vector, $\boldsymbol{\alpha}$, also has $\boldsymbol{\alpha}_{init}$ and $\boldsymbol{\alpha}_{scale}$. Following Loshchilov et al. (2024), we initialize $\boldsymbol{\alpha}_{init} = \boldsymbol{1}/(L+1)$ and $\boldsymbol{\alpha}_{scale} = \boldsymbol{1}/\sqrt{d_h}$, to preserve residual feature and gradually integrate non-linear features.

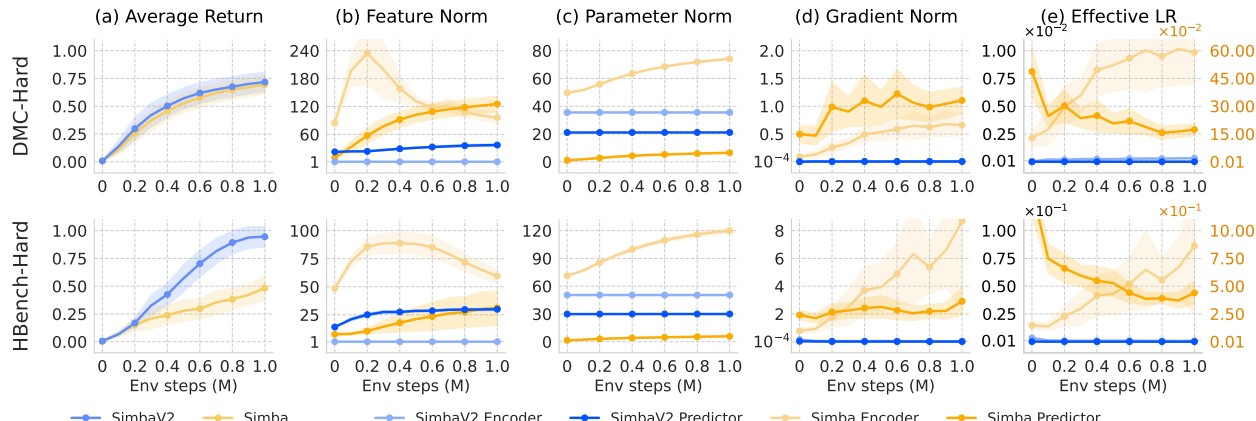

*Figure 4.* **SimbaV2 vs. Simba Training Dynamics.** We track 4 metrics during training to understand the learning dynamics of SimbaV2: **(a)** Average normalized return across tasks. **(b)** Weighted sum of $\ell_2$-norms of all intermediate features in critic. **(c)** Weighted sum of $\ell_2$-norms of all critic parameters **(d)** Weighted sum of $\ell_2$-norms of all gradients in critic **(e)** Effective learning rate (ELR) of the critic. On both environments, SimbaV2 maintains stable norms and ELR, while Simba exhibits divergent fluctuations.

# 5. Experiments

We now present a series of experiments designed to evaluate SimbaV2. Our investigation centers on four main setups:

- **Optinmization Analysis** (Section 5.2). Investigate whether SimbaV2 stabilizes the optimization process.
- **Scaling Analysis** (Section 5.3). Investigate whether SimbaV2 allows scaling model capacity and computation.
- **Comparisons** (Sections 5.4). Compare SimbaV2 against state-of-the-art RL algorithms.
- **Design Study** (Section 5.5.) Conducts ablation studies on individual architectural components of SimbaV2.

## 5.1. Experimental Setup

**Environment.** A total of 57 continuous-control tasks are considered across 4 domains: MuJoCo (Todorov et al., 2012), DMC Suite (Tassa et al., 2018), MyoSuite (Caggiano et al., 2022), and HumanoidBench (Sferrazza et al., 2024). Also, two challenging subsets are defined for an empirical analysis: DMC-Hard (7 tasks involving `dog` and `humanoid` embodiments) and HBench-Hard (5 tasks: `run`, `balance-simple`, `sit-hard`, `stair`, `walk`).

**Baselines.** Comparisons include a broad range of deep RL algorithms, PPO (Schulman et al., 2017), SAC (Haarnoja et al., 2018), TD3 (Fujimoto et al., 2018) TD3+OFE (Ota et al., 2020), TQC (Kuznetsov et al., 2020), DreamerV3 (Hafner et al., 2023), TD7 (Fujimoto et al., 2023), TD-MPC2 (Hansen et al., 2023), Cross-Q (Bhatt et al., 2024), BRO (Nauman et al., 2024b), MAD-TD (Voelcker et al., 2024), MR.Q (Fujimoto et al., 2025), and Simba (Lee et al., 2024c). Whenever available, we report the results from the original paper; otherwise, we run the authors' official code. In addition, to further compare performance before

and after scaling, we evaluate BRO, Simba, and SimbaV2 under both low UTD ratios ($\leq 2$) and high UTD ratios ($\leq 8$). Additional details are described in Appendix E.

**Metrics.** To aggregate performance across diverse domains, each environment's return is normalized to a near $[0, 1)$ range. Specifically, MuJoCo performance is normalized by TD3 (Fujimoto et al., 2018); DMC returns are divided by 1000; MyoSuite scores use success rates; and Humanoid-Bench scores are normalized by their success score.

**Training.** If possible, we tried to closely follow Simba's training configuration aiming to provide an apples-to-apples comparison. Unless otherwise specified, the actor and critic have hidden dimensions of 128 and 512, respectively (approximately 5M parameters). The model is trained for 1M environment steps, using a UTD ratio 2. We used an Adam (Kingma & Ba, 2014) optimizer without weight decay and set the batch size to 256. The learning rate is linearly decayed from $1 \times 10^{-4}$ to $3 \times 10^{-5}$. Full hyperparameter configurations are provided in Appendix C.

## 5.2. Optimization Analysis

To understand the optimizatiom dynamics of SimbaV2, we measure the feature norm, weight norm, gradient norm, and the effective learning rate (ELR), defined as the ratio of the gradient norm to the weight norm (Kodryan et al., 2022; Lyle et al., 2024) (See Appendix G for details). We weighted average each metric across layers where weights correspond to each layer's fraction of total parameters. Additionally, we divide the layers into encoder layers (all layers before the output prediction) and predictor layers (those after) to analyze their respective dynamics.

Figure 4 compares SimbaV2 and Simba on DMC-Hard and HBench-Hard. As shown in Figure 4.(b)-(d), Simba exhibits

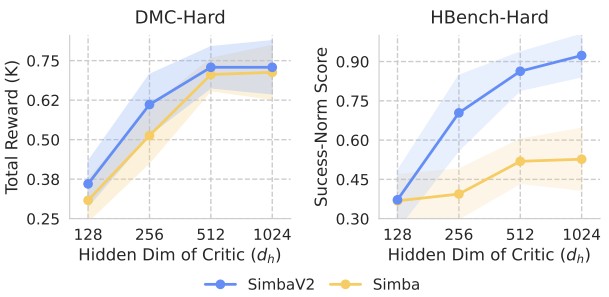

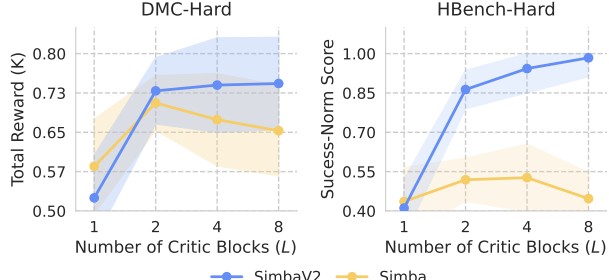

*Figure 5.* **Width Scaling.** We scale the number of model parameters by increasing the width of the critic network. On DMC-Hard, both Simba and SimbaV2 benefit from increased model size. On HBench-Hard, however, Simba plateaus at larger model sizes, whereas SimbaV2 continues to improve.

*Figure 6.* **Depth Scaling.** We scale the number of model parameters by increasing the depth of the critic network. On both DMC-Hard and HBench-Hard, SimbaV2 benefits from increased depth, while Simba's performance degrades beyond a shallow configuration of $L \geq 2$.

large, often divergent fluctuations in feature, weight, and gradient norms between the encoder and predictor. Consequently, Figure 4.(e) shows that the encoder's ELR trending upward while the predictor's ELR declines.

In contrast, SimbaV2 enforces tighter constraints, stabilizing norms and ELRs throughout training. Although certain parameters (e.g., scalers or interpolation vectors) can exceed the unit norm, the majority of parameters remain on the hypersphere, resulting in more robust optimization. A standalone visualization of SimbaV2 is in Appendix G.

### 5.3. Scaling Analysis

For this experiment, we investigate whether SimbaV2's stable training dynamics enable better scaling performance as model parameters or computational resources increase, while reducing overfitting compared to existing methods.

**Experimental Setup.** We conduct two types of scaling experiments. For parameter scaling, we focus on scaling the critic network, as prior studies indicate that scaling the actor provides limited benefits (Nauman et al., 2024b; Lee et al., 2024c). We test two scaling approaches: width scaling by varying the critic's hidden dimension across $\{128, 256, 512, 1024\}$, increasing parameters from $0.3M$ to $17.8M$; and depth scaling by varying the number of critic blocks $L$ across $\{1, 2, 4, 8\}$, growing parameters from $2.2M$ to $17.8M$.

For compute scaling experiments, we vary the update-to-data (UTD) ratio across $\{1, 2, 4, 8\}$. We compare results both with and without periodic weight reinitialization, since prior work suggests that compute scaling requires periodic reinitialization to avoid overfitting (D'Oro et al., 2022). Following Nauman et al. (2024b), we apply reinitialization every 500,000 update steps when used.

**Parameter Scaling.** Figure 5 shows width scaling results on DMC-Hard (left) and HBench-Hard (right). Both Simba and SimbaV2 benefit from larger models on DMC-Hard.

However, on the more challenging HBench-Hard benchmark, while both methods achieve comparable performance at the smallest scale ($d_h = 128$), their scaling behavior diverges significantly. Simba plateaus at larger scales with peak performance at $d_h = 1024$, while SimbaV2 continues to improve with increased width. This demonstrates that SimbaV2's stabilized training dynamics effectively leverages larger model capacity.

Figure 6 presents depth scaling results. In HBench-Hard, SimbaV2 shows consistent performance improvements as the depth of the critic $L$ increases, successfully solving the five complex tasks in $L = 8$. On DMC-Hard, SimbaV2's performance also improves with depth but begins saturating around $L = 4$, likely due to task complexity limitations rather than architectural constraints. In contrast, Simba's performance either plateaus around $L = 2$ or slightly decreases after initial improvement. This clear difference demonstrates SimbaV2's superior depth scalability, which we attribute to its effective regularization mechanisms that enable stable training of deeper networks.

**Compute Scaling.** We next explore compute scaling through increased UTD ratios, a key factor in improving sample efficiency in deep RL (Li et al., 2023). While higher UTD ratios can enhance sample efficiency, they also increase the risk of overfitting. Previous approaches address this through ensembling (Chen et al., 2021b), periodic reinitialization (Lee et al., 2024a; D'Oro et al., 2023; Nauman et al., 2024b), or both (Kim et al., 2023). We investigate whether SimbaV2's stable dynamics enable effective scaling without these additional mechanisms.

Figure 7 shows the effect of varying the UTD ratio on DMC-Hard (left) and HBench-Hard (right), comparing Simba and SimbaV2 with and without reinitialization (solid lines: no reinitialization; dashed lines: reinitialization). In Simba, performance plateaus at a UTD ratio of 2 on DMC-Hard and 1 on HBench-Hard. When combined with reinitialization, but further improves with reinitialization, consistent with

*Table 1.* **Online RL.** Average final performance after 1M environment steps, where $\pm$ captures a 95% confidence interval (CI) computed over all raw benchmark samples. For algorithms with only average scores for each task available, we approximate the CI using these averages (†). Note that this estimation may be inaccurate. The highest performance is highlighted. Any performance that is not statistically worse than the highest performance (according to Welch's $t$-test with significance level 0.05) is highlighted.

| Method | Mujoco (5) TD3.Norm | DMC-Easy (21) Return (1k) | DMC-Hard (7) Return (1k) | MyoSuite (10) Success Rate | HBench (14) Success.Norm | All (57) - |
|---|---|---|---|---|---|---|
| **(a) Low UTD ($\leq 2$)** | | | | | | |
| PPO (Schulman et al., 2017) | $0.447 \pm 0.270^{\dagger}$ | $0.327 \pm 0.128^{\dagger}$ | $0.033 \pm 0.030^{\dagger}$ | - | - | - |
| SAC (Haarnoja et al., 2018) | $1.092 \pm 0.081$ | $0.762 \pm 0.094^{\dagger}$ | $0.136 \pm 0.04$ | $0.607 \pm 0.088$ | $0.279 \pm 0.050$ | $0.554 \pm 0.057$ |
| TD3 (Fujimoto et al., 2018) | $1.000 \pm 0.000^{\dagger}$ | - | - | - | - | - |
| TD3+OFE (Ota et al., 2020) | $1.322 \pm 0.263^{\dagger}$ | - | - | - | - | - |
| TQC (Kuznetsov et al., 2020) | $1.137 \pm 0.125^{\dagger}$ | - | - | - | - | - |
| TD7 (Fujimoto et al., 2023) | $1.570 \pm 0.030$ | $0.689 \pm 0.134^{\dagger}$ | $0.182 \pm 0.137^{\dagger}$ | $0.356 \pm 0.126$ | $0.289 \pm 0.083$ | $0.617 \pm 0.358^{\dagger}$ |
| TD-MPC2 (Hansen et al., 2023) | $1.040 \pm 0.115$ | $0.889 \pm 0.064^{\dagger}$ | $0.465 \pm 0.139^{\dagger}$ | $0.650 \pm 0.148$ | $0.710 \pm 0.149$ | $0.749 \pm 0.168^{\dagger}$ |
| CrossQ (Bhatt et al., 2024) | $1.475 \pm 0.141$ | - | - | - | - | - |
| MR.Q (Fujimoto et al., 2025) | $1.448 \pm 0.156$ | $0.868 \pm 0.026$ | $0.723 \pm 0.061$ | - | - | - |
| BRO (Nauman et al., 2024b) | $1.101 \pm 0.182$ | $0.861 \pm 0.036$ | $0.693 \pm 0.066$ | $0.714 \pm 0.076$ | $0.468 \pm 0.107$ | $0.731 \pm 0.039$ |
| Simba (Lee et al., 2024c) | $1.147 \pm 0.077$ | $0.864 \pm 0.024$ | $0.706 \pm 0.05$ | $0.743 \pm 0.079$ | $0.606 \pm 0.073$ | $0.780 \pm 0.028$ |
| SimbaV2 (ours) | $1.617 \pm 0.103$ | $0.874 \pm 0.025$ | $0.729 \pm 0.065$ | $0.847 \pm 0.066$ | $0.776 \pm 0.064$ | $0.892 \pm 0.032$ |
| **(b) High UTD ($\geq 8$)** | | | | | | |
| REDQ (Chen et al., 2021b) | $1.160 \pm 0.071$ | - | - | - | - | - |
| DroQ (Hiraoka et al., 2021) | $1.134 \pm 0.070$ | - | - | - | - | - |
| DreamerV3 (Hafner et al., 2023) | $0.760 \pm 0.095$ | $0.714 \pm 0.124^{\dagger}$ | $0.009 \pm 0.006^{\dagger}$ | $0.482 \pm 0.166$ | $0.022 \pm 0.023$ | $0.397 \pm 0.289^{\dagger}$ |
| MAD-TD (Voelcker et al., 2024) | - | - | $0.708 \pm 0.065$ | - | - | - |
| BRO (Nauman et al., 2024b) | $1.150 \pm 0.202$ | $0.871 \pm 0.034$ | $0.767 \pm 0.059$ | $0.814 \pm 0.066$ | $0.619 \pm 0.117$ | $0.807 \pm 0.037$ |
| Simba (Lee et al., 2024c) | $1.175 \pm 0.136$ | $0.866 \pm 0.036$ | $0.720 \pm 0.087$ | $0.834 \pm 0.098$ | $0.657 \pm 0.099$ | $0.818 \pm 0.043$ |
| SimbaV2 (ours) | $1.598 \pm 0.176$ | $0.876 \pm 0.035$ | $0.769 \pm 0.089$ | $0.866 \pm 0.090$ | $0.822 \pm 0.099$ | $0.911 \pm 0.044$ |

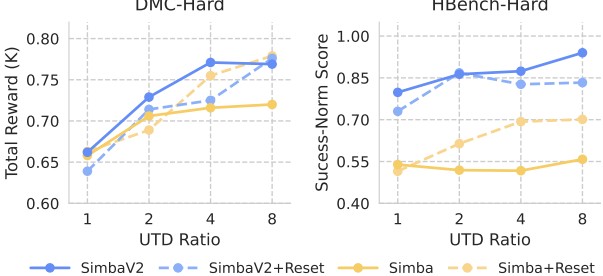

*Figure 7.* **Compute Scaling.**. We scale compute by increasing the UTD ratio. We compare Simba and SimbaV2, both with and without periodic reset. Simba saturates at lower ratios without reset, but improves with reset. In contrast, SimbaV2 scales smoothly even without reset, where using reset slightly degrades its performance.

D'Oro et al. (2023). In contrast, SimbaV2 scales consistently as the UTD ratio increases, even without reinitialization. Notably, reinitialization slightly degrades SimbaV2's performance, as it disrupts training and adds time to recover.

To verify the importance of hyperspherical weight and feature normalization for UTD scaling, we test a variant in Appendix H.1 that includes distributional critics and reward scaling on Simba. This variant fails to scale at higher UTD ratios, confirming the critical role of hyperspherical normalization for effective scaling.

## 5.4. Online RL

Having observed SimbaV2's scalability, we now compare it against standard model-free and model-based RL.

Table 1.(a) presents results at a UTD ratio below 2. SimbaV2 with UTD=2 attains an average normalized score of 0.892, exceeding the previous best of 0.780. Only except for DMC-Easy suite, SimbaV2 outperforms leading model-free (CrossQ (Bhatt et al., 2024), BRO (Nauman et al., 2024b), Simba (Lee et al., 2024c)) and model-based (TD-MPC2 (Hansen et al., 2023), MR.Q (Fujimoto et al., 2025)) baselines, demonstrating superior sample efficiency.

Table 1.(b) evaluates higher UTD settings. Increasing SimbaV2's UTD from 2 to 8 further elevates its average score from 0.892 to 0.911. SimbaV2 also surpasses BRO with UTD=10, which utilizes periodic reinitialization to avoid overfitting at high update rates. These consistent gains at larger UTD ratios underscore the efficacy of hyperspherical normalization in stabilizing training.

For offline RL, we simply add a behavioral cloning loss during training with using identical configurations to the online RL. Despite minimal changes, SimbaV2 performs competitively with existing baselines (Appendix D).

## 5.5. Design Study

Table 2 presents the results from ablation studies isolating the contributions of various architectural choices.

**Input Projection.** Projecting observations onto a hypersphere before passing them through the linear layer is crucial for performance (Table 2.(a)), where omitting this step leads to a significant performance drop. Equally important design is preserving the original magnitude during projection

*Table 2.* **Design Study.** We report the final performance for each design choice in the online RL benchmarks, averaged over 3 random seeds. Performance changes relative to the default SimbaV2 are highlighted according to their percentile difference: mild positive changes $[0.02, 0.05)$ and mild negative changes $(-0.05, -0.02]$ are highlighted lightly, damaging changes $(-0.1, -0.05]$ are highlighted moderately, and catastrophic changes $(-1.0, -0.1]$ are highlighted boldly.

| Design (idx) Original → Changed | Mujoco (5) TD3.Norm | DMC-Easy (21) Return (1k) | DMC-Hard (7) Return (1k) | MyoSuite (10) Success Rate | HBench (14) Success.Norm | All (57) - |
|---|---|---|---|---|---|---|
| **Input Projection** | | | | | | |
| (a) L2 Normalize → No L2 Normalize | $1.370 \pm 0.220$ | $0.779 \pm 0.053$ | $0.700 \pm 0.094$ | $0.815 \pm 0.100$ | $0.711 \pm 0.123$ | $0.809 \pm 0.059$ |
| (b) Shifting → No Shifting | $1.406 \pm 0.233$ | $0.771 \pm 0.056$ | $0.724 \pm 0.088$ | $0.800 \pm 0.105$ | $0.700 \pm 0.125$ | $0.810 \pm 0.061$ |
| (c) $c_{shift} : 3 \rightarrow 1$ | $1.558 \pm 0.167$ | $0.862 \pm 0.039$ | $0.718 \pm 0.089$ | $0.870 \pm 0.085$ | $0.791 \pm 0.130$ | $0.888 \pm 0.058$ |
| (d) Shift Projection → Resize Projection | $1.623 \pm 0.176$ | $0.842 \pm 0.043$ | $0.720 \pm 0.093$ | $0.852 \pm 0.083$ | $0.779 \pm 0.122$ | $0.884 \pm 0.058$ |
| **Output Prediction** | | | | | | |
| (e) Categorical Loss → MSE Loss | $1.343 \pm 0.097$ | $0.868 \pm 0.034$ | $0.708 \pm 0.097$ | $0.757 \pm 0.130$ | $0.767 \pm 0.129$ | $0.846 \pm 0.062$ |
| (f) Reward Scaling → No Scaling | $1.395 \pm 0.151$ | $0.852 \pm 0.034$ | $0.712 \pm 0.092$ | $0.840 \pm 0.077$ | $0.735 \pm 0.085$ | $0.852 \pm 0.042$ |
| (g) Reward Bounding → No Bounding | $1.620 \pm 0.142$ | $0.824 \pm 0.033$ | $0.733 \pm 0.072$ | $0.805 \pm 0.121$ | $0.787 \pm 0.128$ | $0.868 \pm 0.051$ |
| (h) Soft Target → Hard Target | $1.589 \pm 0.175$ | $0.878 \pm 0.037$ | $0.746 \pm 0.081$ | $0.848 \pm 0.086$ | $0.770 \pm 0.094$ | $0.890 \pm 0.051$ |
| **Initialization & Update** | | | | | | |
| (i) LR Decay → No LR Decay | $1.562 \pm 0.162$ | $0.858 \pm 0.042$ | $0.719 \pm 0.065$ | $0.810 \pm 0.114$ | $0.754 \pm 0.119$ | $0.863 \pm 0.064$ |
| (j) $s_{init} : \sqrt{2}/\sqrt{d_h} \rightarrow 1$ | $1.571 \pm 0.105$ | $0.873 \pm 0.022$ | $0.718 \pm 0.052$ | $0.855 \pm 0.062$ | $0.781 \pm 0.074$ | $0.890 \pm 0.032$ |
| (k) $s_{scale} : \sqrt{2}/\sqrt{d_h} \rightarrow 1$ | $1.594 \pm 0.102$ | $0.870 \pm 0.025$ | $0.706 \pm 0.055$ | $0.836 \pm 0.053$ | $0.789 \pm 0.072$ | $0.887 \pm 0.033$ |
| (l) $\alpha_{init} : 1/(L+1) \rightarrow 0.5$ | $1.583 \pm 0.172$ | $0.866 \pm 0.038$ | $0.728 \pm 0.084$ | $0.843 \pm 0.068$ | $0.745 \pm 0.102$ | $0.877 \pm 0.057$ |
| (m) $\alpha_{scale} : 1/\sqrt{d_h} \rightarrow 1$ | $1.520 \pm 0.177$ | $0.856 \pm 0.034$ | $0.714 \pm 0.089$ | $0.875 \pm 0.079$ | $0.792 \pm 0.125$ | $0.885 \pm 0.059$ |
| SimbaV2 | $1.617 \pm 0.103$ | $0.874 \pm 0.025$ | $0.729 \pm 0.064$ | $0.847 \pm 0.066$ | $0.776 \pm 0.071$ | $0.892 \pm 0.032$ |

(Table 2.(b)). We also explore an alternative "resize" projection, where inputs are first divided by $c_{\text{shift}}\sqrt{d_h}$ before being projected onto an $(n + 1)$-dimensional hypersphere. The resize projection yields comparable performance as it can also retain magnitude information (Table 2.(d)).

**Output Projection.** Incorporating a distributional critic and reward scaling improves performance, especially in environments with high reward variance like MuJoCo (Table 2.(e)–(f)). Bounding target returns proves essential for easier tasks (Table 2.(g)), such as cartpole in the DMC-Easy suite (Table 25). Without bounding, consistent high returns can diminish return variance, and scaling returns push target values beyond the range of the distributional critic, leading to collapse in the TD loss.

**Initialization & Update.** Gradually decaying the learning rate is critical. Without decay, the model may struggle to refine its predictions during later training stages, as SimbaV2 maintains an effective constant learning rate throughout training (Table 2.(i)). Tuning initial scaler values has minimal impact on performance where the architecture remains stable by these changes (Table 2.(j)–(m)).

## 6. Lessons and Opportunities

**Lessons.** Historically, RL research has relied on complex regularizations to address overfitting and scalability issues (Klein et al., 2024). Our findings suggest that suitably chosen constraints, exemplified by SimbaV2, can simplify these design complexities while retaining strong performance.

**Opportunities.** Future opportunities include deploying SimbaV2 in real-world robotics (Hwangbo et al., 2019),

where sample efficiency is crucial, and extending it to model-based (Hansen et al., 2023) or visual RL (Kostrikov et al., 2020). Furthermore, with increasing interest in RL for training large language models (Ouyang et al., 2022; Guo et al., 2025), the potential benefits of using stricter normalization for large models remain an exciting open question.

## Impact Statement

This paper presents work aimed at advancing the field of Machine Learning. There are many potential societal consequences of our work, none of which we feel must be specifically highlighted here.

## Acknowledgements

We would like to express our gratitude to Dongyoon Hwang and Hawon Jeong for their valuable feedback on this paper.

This work was supported by Institute for Information & communications Technology Planning & Evaluation (IITP) grant funded by the Korea government (MSIT) (RS-2019-II190075, Artificial Intelligence Graduate School Program (KAIST)). This work was supported by the National Research Foundation of Korea (NRF) grant funded by the Korea government (MSIT) (No. RS-2025-00555621) This work was mainly supported by Institute of Information & communications Technology Planning & Evaluation (IITP) grant funded by the Korea government (MSIT) (No.RS-2021-II212068, Artificial Intelligence Innovation Hub).

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

# Appendix

## A. Architectural Details

### A.1. LERP: A Retraction-based Approximation on Riemannian Manifolds.

During the feature encoding stage in SimbaV2, the input $h$ and its non-linearly transformed output $\tilde{h}$ are linearly interpolated using a learnable interpolation vector $\boldsymbol{\alpha} \in \mathbb{R}^{d_h}$:

$$h \leftarrow \ell_2\text{-Norm}((\mathbf{1} - \boldsymbol{\alpha}) \odot h + \boldsymbol{\alpha} \odot \tilde{h}), \tag{22}$$

followed by $\ell_2$-normalization.

Intuitively, this can be interpreted as a first-order (retraction-based) approximation of the Riemannian update formula on the hypersphere. This section provides a brief introduction to the differential geometry concepts that underpin the Riemannian optimization perspective of $\boldsymbol{\alpha}$. For brevity, we omit the mathematical definitions, derivations, and proofs here. The comprehensive introduction to differential geometry and Riemannian optimization can be found in Spivak (1970), Do Carmo & Flaherty Francis (1992), and Boumal (2023).

Let $\mathbb{S}_{n-1}$ denote the $n$-dimensional hypersphere embedded in $\mathbb{R}^n$, i.e., $\mathbb{S}_{n-1} = \{h \in \mathbb{R}^n \mid \|h\|_2 = 1\}$.

**Manifold.** A *manifold* $\mathcal{M}$ of dimension $n$ is a space that can locally be approximated by a Euclidean space $\mathbb{R}^n$. The simplest examples of a manifold include the open ball $U = \{x \in \mathbb{R}^n \mid \|x\|_2 < r\}$ for $r \in \mathbb{R}_{>0}$, and the hypersphere $\mathbb{S}_{n-1}$ is also a manifold in $\mathbb{R}^n$.

**Tangent Spaces.** At each point $x \in \mathcal{M}$, the *tangent space* $T_x\mathcal{M}$ is an $n$-dimensional vector space that locally approximates $\mathcal{M}$ near $x$. Tangent vectors generalize the concept of directional derivatives. For the hypersphere $\mathbb{S}_{n-1}$, the tangent space at a point $p$ consists of all vectors orthogonal to $p$:

$$T_p\mathbb{S}_{n-1} = \{h \in \mathbb{R}^n \mid \langle p, h \rangle = 0\} \tag{23}$$

where $\langle \cdot, \cdot \rangle$ denotes the Euclidean inner product.

**Riemannian Metrics and Manifolds.** The tangent space $T_x\mathcal{M}$ is not inherently equipped with an inner product. A *Riemannian metric* $\rho$ provides a collection of inner products $\rho_x(\cdot, \cdot) : T_x\mathcal{M} \times T_x\mathcal{M} \to \mathbb{R}$ on the tangent spaces, $\rho := (\rho_x)_{x \in \mathcal{M}}$, which locally define the geometry of $\mathcal{M}$. A *Riemannian manifold* $(\mathcal{M}, \rho)$ is a smooth manifold $\mathcal{M}$ equipped with such a metric. This enables us to define geometric notions such as distance, angle, length, volume, and curvature of manifold. For a detailed explanation of geometrics on Riemannian manifolds, refer to Lee (2006).

**Exponential Mapping and Retraction.** Under some conditions (Do Carmo & Flaherty Francis, 1992), the *exponential map* $\exp_x : T_x\mathcal{M} \to \mathcal{M}$ can be defined at a point $x \in \mathcal{M}$. $\exp_x(v)$ maps a tangent vector $v \in T_x\mathcal{M}$ to a point on the manifold along the geodesic from $x$ in the direction of $v$. Therefore, for small $t \in \mathbb{R}$, $\exp_x(tv)$ represents the shortest path on $\mathcal{M}$ starting at $x$ with initial direction $v$. In Euclidean space $(\mathbb{R}^n, I_n)$, the exponential map $\exp_x(v) = x + v$ is simply defined as a straight path. In practice, for computational efficiency (e.g., the mappings do not have closed-form), we often approximate the exponential map $\exp_x$ by a *retraction* (Absil et al., 2008) $R_x$:

**Definition A.1** (Retraction). A retraction $R$ on a manifold $\mathcal{M}$ is a smooth map:

$$R \colon \bigcup_{x \in \mathcal{M}} T_x\mathcal{M} \to \mathcal{M}$$
$$(x, v) \mapsto R_x(v)$$

with the following properties:

$$R_x(\mathbf{0}) = x \qquad \text{and} \qquad (dR_x)_0 = \text{id}$$

where $R_x$ denotes the restriction of $R$ to $T_x\mathcal{M}$, $(dR_x)_0$ denotes the differential of $R_x$ at $\mathbf{0}$, and id is the identity map.

Intuitively, a retraction $R_x(v)$ provides a first-order approximation of the exponential map $\exp_x(v)$ (Boumal, 2023). Figure 8 illustrates the difference between the exponential map and retraction on $\mathbb{S}_2$. For the hypersphere $\mathbb{S}_{n-1}$, the retraction of a tangent vector $\boldsymbol{\xi} \in T_h\mathbb{S}_{n-1}$ onto $\mathbb{S}_{n-1}$ is given by (Absil et al., 2008):

$$R_h(\boldsymbol{\xi}) = \ell_2\text{-Norm}(h + \boldsymbol{\xi}) = \frac{h + \boldsymbol{\xi}}{\|h + \boldsymbol{\xi}\|_2} \tag{24}$$

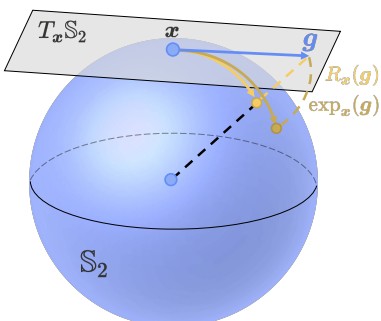

*Figure 8.* **Exponential Map vs. Retraction on 3-dimensional sphere.** Comparison of the exponential map $\exp_{\boldsymbol{x}}$ and the retraction $R_{\boldsymbol{x}}$ on the 3-dimensional sphere $\mathbb{S}_2$. The exponential mapping sends a tangent vector $\boldsymbol{g} \in T_{\boldsymbol{x}}\mathbb{S}_2$ exactly along the geodesic from $\boldsymbol{x}$ to a point on the manifold, while the retraction locally approximates this mapping to first order. Figure adapted from Sutti & Yueh (2024).

**Riemannian Optimization.** On Riemannian manifolds, gradient updates ideally follow the curved geodesics, rather than straight lines as in Euclidean space. To this end, Bonnabel (2013) introduce Riemannian SGD that generalizes SGD to Riemannian manifolds using exponential map:

$$\boldsymbol{h} \leftarrow \exp_{\boldsymbol{h}}(-\alpha \boldsymbol{g}) \tag{25}$$

where $\alpha > 0$ is the global learning rate and $\boldsymbol{g} \in T_{\boldsymbol{h}}\mathcal{M}$ denotes the *Riemannian gradient*.

In our case, $-(\tilde{\boldsymbol{h}} - \boldsymbol{h})$ can be viewed as the gradient $\boldsymbol{g}$ in the Euclidean space. Then, we project the gradient onto the tangent space $T_{\boldsymbol{h}}\mathbb{S}_{n-1}$:

$$\boldsymbol{g}_{\text{proj}} = \boldsymbol{g} - \langle \boldsymbol{g}, \boldsymbol{h} \rangle \boldsymbol{h} \tag{26}$$
$$= -(\tilde{\boldsymbol{h}} - \boldsymbol{h}) - \langle -\tilde{\boldsymbol{h}} + \boldsymbol{h}, \boldsymbol{h} \rangle \boldsymbol{h} \tag{27}$$
$$= -\tilde{\boldsymbol{h}} + \langle \tilde{\boldsymbol{h}}, \boldsymbol{h} \rangle \boldsymbol{h} \tag{28}$$

Applying the retraction $\exp_{\boldsymbol{h}}(-\alpha \boldsymbol{g}_{\text{proj}}) \approx R_{\boldsymbol{h}}(-\alpha \boldsymbol{g}_{\text{proj}})$:

$$\boldsymbol{h} \leftarrow \ell_2\text{-Norm}\left(\boldsymbol{h} + \alpha(\tilde{\boldsymbol{h}} - \langle \tilde{\boldsymbol{h}}, \boldsymbol{h} \rangle \boldsymbol{h})\right) \tag{29}$$
$$= \ell_2\text{-Norm}\left((1 - \alpha \langle \tilde{\boldsymbol{h}}, \boldsymbol{h} \rangle)\boldsymbol{h} + \alpha \tilde{\boldsymbol{h}}\right) \tag{30}$$

Thus, the LERP operation in SimbaV2 can be interpreted as a retraction-based approximation of the Riemannian update rule on the hypersphere, where the learning rate $\alpha$ is replaced by a learnable vector $\boldsymbol{\alpha}$ and the inner product $\langle \tilde{\boldsymbol{h}}, \boldsymbol{h} \rangle$ term is neglected. Also, Loshchilov et al. (2024) empirically show that neglecting the inner product term has no significant impact on performance.

**A.2. Scaler Initialization**

In our algorithm, the *scaler* $s \in \mathbb{R}^{d_h}$ is a learnable vector that element-wise scales the output $z$ of the linear layer:

$$z = s \odot Wh \in \mathbb{R}^{d_h} \tag{31}$$

where $W \in \mathbb{R}^{d_h \times n}$ is the weight matrix of the linear layer, and $h \in \mathbb{R}^n$ is the input vector. To ensure that $z$ (approximately) maintains unit norm at initialization, we initialize $s$ as $s = \sqrt{\frac{2}{d_h}} \cdot \mathbf{1}$. The following section provides the derivation for this initialization.

We assume that each normalized embedding $w_l \in \mathbb{R}^n$ of $W$, and a random $n$-dimensional normalized vector $h \in \mathbb{R}^n$, are uniformly distributed on the $n$-dimensional hypersphere $\mathbb{S}_{n-1}$. Furthermore, we assume that the vectors $w_l$ and $h$ are mutually independent (Feller, 1991). We denote the angle between $w_l$ and $h$ by $\theta_l$, such that $\cos \theta_l = w_l \cdot h$ since $\|w_l\|_2 = \|h\|_2 = 1$.

**Distribution of the Cosine of the Angle.** For simplicity, assume that $w_l$ is fixed. Since $h$ is uniformly distributed on the hypersphere, the distribution of the angle $\theta_l$ depends on the *solid angle* (Weisstein, 2005) subtended by $h$ with respect to $w_l$. The surface area $A_{n-2}$ of an $(n-1)$-dimensional hyperspherical cap (Li, 2010) leads to the probability density function $f(\theta_l)$ (Cai et al., 2013):

$$f(\theta_l) = \frac{A_{n-1}}{S_{n-1}} = \frac{\frac{2\pi^{(n-1)/2}}{\Gamma(\frac{n-1}{2})}}{\frac{2\pi^{n/2}}{\Gamma(\frac{n}{2})}} \sin^{n-2}(\theta_l) = \frac{\Gamma(\frac{n}{2})}{\sqrt{\pi}\Gamma(\frac{n-1}{2})} \sin^{n-2}(\theta_l) \tag{32}$$

where $\theta_l \in [0, \pi]$, $\Gamma$ is the gamma function and $S_{n-1}$ is the surface area of $\mathbb{S}_{n-1} = \frac{2\pi^{n/2}}{\Gamma(\frac{n}{2})}$ (Weisstein, 2002).

**Norm of Output Vector.** Let $z = s \odot Wh \in \mathbb{R}^{d_h}$ be the output of the linear layer. Each element of $z$ and $s$, denoted by $z_l$ and $s_l$, respectively, corresponds to the scaled cosine of the angle $\theta_l$ between $w_l$ and $h$:

$$z = s \odot Wh = \begin{bmatrix} s_1(w_1 \cdot h) \\ s_2(w_2 \cdot h) \\ \vdots \\ s_{d_h}(w_{d_h} \cdot h) \end{bmatrix} = \begin{bmatrix} s_1 \cos \theta_1 \\ s_2 \cos \theta_2 \\ \vdots \\ s_{d_h} \cos \theta_{d_h} \end{bmatrix} \tag{33}$$

The expected squared norm of $z$ is then given by:

$$\mathbb{E}[\|z\|_2^2] = \sum_{l=1}^{d_h} s_l^2 \mathbb{E}[\cos^2 \theta_l] \tag{34}$$

Using the trigonometric identity $\cos^2(\theta) = \frac{1+\cos(2\theta)}{2}$ and the following integrals:

$$\int_0^\pi \sin^{n-2}(\theta) \, d\theta = \frac{\Gamma(\frac{n-1}{2})\Gamma(\frac{1}{2})}{\Gamma(\frac{n}{2})} = \frac{\sqrt{\pi}\Gamma(\frac{n-1}{2})}{\Gamma(\frac{n}{2})} \tag{35}$$

$$\int_0^\pi \cos(2\theta) \sin^{n-2}(\theta) \, d\theta = 0 \tag{36}$$

where the second integral vanishes due to the symmetry of $\cos(2\theta)$ about $\theta = \frac{\pi}{2}$, we compute the expectation:

$$\mathbb{E}[\cos^2(\theta_l)] = \int_0^\pi \cos^2(\theta_l) \underbrace{\frac{\Gamma(\frac{n}{2})}{\sqrt{\pi}\Gamma(\frac{n-1}{2})} \sin^{n-2}(\theta_l)}_{f(\theta_l)} \, d\theta_l \tag{37}$$

$$= \frac{\Gamma(\frac{n}{2})}{\sqrt{\pi}\Gamma(\frac{n-1}{2})} \int_0^\pi \cos^2(\theta_l) \sin^{n-2}(\theta_l) d\theta \tag{38}$$

$$= \frac{\Gamma(\frac{n}{2})}{2\sqrt{\pi}\Gamma(\frac{n-1}{2})} \times \frac{\sqrt{\pi}\Gamma(\frac{n-1}{2})}{\Gamma(\frac{n}{2})} = \frac{1}{2} \tag{39}$$

Thus, by setting $s_l = \sqrt{\frac{2}{d_h}}$ for all $\ell \in \{1, \cdots, d_h\}$, we expect that the expected norm $\mathbb{E}[\|z\|_2^2]$ is 1 at initialization.

# B. Implementation Details

Listings 1, 2 and 3 provide the Google JAX implementation of scaling vector (Section 4.4), input embedding (Section 4.1), and MLP block (Section 4.2), respectively.

```python
import flax.linen as nn

class Scaler(nn.Module):
    dim: int
    init: float
    scale: float

    def setup(self):
        self.scaler = self.param(
            nn.initializers.constant(1.0 * self.scale),
            self.dim,
        )
        self.forward_scaler = self.init / self.scale

    def __call__(self, x: jnp.ndarray) -> jnp.ndarray:
        return self.scaler * self.forward_scaler * x
```

*Listing 1.* A JAX implementation of Scaler (Section 4.4)

```python
import jax.numpy as jnp
import flax.linen as nn

class InputEmbedding(nn.Module):
    observation_dim: int
    hidden_dim: int
    shift_const: float
    input_scaler_init: float
    input_scaler_scale: float

    def setup(self):
        self.obs_rms = RunningMeanStd(
            shape=self.observation_dim
        )
        self.w0 = nn.Dense(
            features=self.hidden_dim,
            use_bias=False
        )
        self.input_scaler = Scaler(
            dim=self.observation_dim,
            init=input_scaler_init,
            scale=input_scaler_scale
        )

    def __call__(self, observation: jnp.ndarray) -> jnp.ndarray:
        # RSNorm
        o = (observations - self.obs_rms.mean) / jnp.sqrt(
            self.obs_rms.var + self.epsilon
        )
        # Shift + l2-Norm
        new_axis = jnp.ones((o.shape[:-1] + (1,))) * self.shift_const
        o = jnp.concatenate([o, new_axis], axis=-1)
        o = l2normalize(o, axis=-1)
        # Linear + Scaler
        h = self.w0(o)
        h = self.input_scaler(h)
        h = l2normalize(h, axis=-1)
        return h
```

*Listing 2.* A JAX implementation of Input Embedding (Section 4.1).

```python
import flax.linen as nn

class SimbaV2Block(nn.Module):
    hidden_dim: int
    ffn_scaler_init: float
    ffn_scaler_scale: float
    alpha_scaler_init: float
    alpha_scaler_scale: float

    def setup(self):
        self.w1 = nn.Dense(
            features=4*self.hidden_dim,
            use_bias=False
        )(x)
        self.mlp_scaler = Scaler(
            dim=4*self.hidden_dim,
            init=ffn_scaler_init,
            scale=ffn_scaler_scale
        )
        self.w2 = nn.Dense(
            features=self.hidden_dim,
            use_bias=False
        )
        self.alpha = Scaler(
            dim=self.hidden_dim,
            init=alpha_scaler_init,
            scale=alpha_scaler_scale
        )

    def __call__(self, x: jnp.ndarray) -> jnp.ndarray:
        residual = x
        # MLP + l2-Norm
        x = self.w1(x)
        x = self.mlp_scaler(x)
        x = nn.relu(x)
        x = self.w2(x)
        x = l2normalize(x, axis=-1)
        # LERP + l2-Norm
        x = l2normalize(residual + self.alpha(x - residual), axis=-1)
        return x
```

*Listing 3.* A JAX implementation of MLP block (Section 4.2).

# C. Hyperparameters

For all experiments, we use consistent hyperparameters across benchmarks. The default settings are listed in Table 3.

*Table 3.* **Hyperparameters Table.** The hyperparameters listed below are used consistently across all tasks using SimbaV2, unless stated otherwise. For the discount factor $\gamma$, we set it automatically using heuristics used by TD-MPC2 (Hansen et al., 2023).

|  | Hyperparameter | Notation | Value |
|---|---|---|---|
| **Input** | Shift constant | $c_{\text{shift}}$ | 3.0 |
| **Output** | Number of return bins | $n_{\text{atoms}}$ | 101 |
|  | Support of return | $[G_{\min}, G_{\max}]$ | $[-5, 5]$ |
|  | Reward scaler epsilon | $\epsilon$ | 1e$-8$ |
| **Training** | Input scaler | $(\boldsymbol{s}_{h,\text{init}}^0, \boldsymbol{s}_{h,\text{scale}}^0)$ | $(\sqrt{2}/\sqrt{d_h}, \sqrt{2}/\sqrt{d_h})$ |
|  | MLP scaler | $(\boldsymbol{s}_{h,\text{init}}^l, \boldsymbol{s}_{h,\text{scale}}^l)$ | $(\sqrt{2}/\sqrt{4d_h}, \sqrt{2}/\sqrt{4d_h})$ |
|  | Output scaler | $(\boldsymbol{s}_{o,\text{init}}, \boldsymbol{s}_{o,\text{scale}})$ | $(\sqrt{2}/\sqrt{d_h}, \sqrt{2}/\sqrt{d_h})$ |
|  | LERP vector | $(\boldsymbol{\alpha}_{\text{init}}, \boldsymbol{\alpha}_{\text{scale}})$ | $(1/(L+1), 1/\sqrt{d_h})$ |
|  | Behavior cloning weight | $\lambda$ | Online: 0.0 |
|  |  |  | Offline: 0.1 |
| **Common** | Discount factor | $\gamma$ | Heuristic (Hansen et al., 2023) |
|  | Replay buffer capacity | - | 1M |
|  | Buffer sampling | - | Uniform |
|  | Batch size | - | 256 |
|  | Update-to-data (UTD) ratio | - | 2 |
|  | TD steps | $k$ | 1 |
| **Actor** | Number of blocks | $L$ | 1 |
|  | Hidden dimension | $d_h$ | 128 |
|  | Initial temperature | $\alpha_0$ | 1e$-2$ |
|  | Target entropy | $\mathbb{H}^*$ | $|\mathcal{A}|/2$ |
| **Critic** | Number of blocks | $L$ | 2 |
|  | Hidden dimension | $d_h$ | 512 |
|  | Number of atoms | $n_{\text{atoms}}$ | 101 |
|  | Target critic momentum | $\tau$ | 5e$-3$ |
|  | Clipped double Q | - | Has Failure Termination (Mujoco, HBench): True |
|  |  |  | No Failure Termination (DMC, MyoSuite): False |
| **Optimizer** | Optimizer | - | Adam |
|  | Optimizer momentum | $(\beta_1, \beta_2)$ | (0.9, 0.999) |
|  | Weight Decay | - | 0.0 |
|  | Learning rate init | $\eta$ | 1e$-4$ |
|  | Learning rate final | - | 3e$-4$ |

# D. Offline RL

In this section, we assess whether the SimbaV2 architecture also provide benefits in offline RL, training from a stationary distribution. We adopt the minimalist offline RL method from (Fujimoto & Gu, 2021), where the behavioral cloning loss is integrated into the reinforcement learning objective. The objective is defined as:

$$\pi \approx \arg\max_{\pi} \mathbb{E}_{(s,a)\sim D} \left[ Q(s, \pi(s)) - \lambda \left| \mathbb{E}_{s\sim D}[Q(s, \pi(s))] \right| \cdot (\pi(s) - a)^2 \right] \tag{40}$$

where we used $\lambda = 0.1$, as in (Fujimoto et al., 2023), and no parameter tuning is performed.

## D.1. Experimental Setup

**Environment.** We use 9 MuJoCo tasks from the D4RL (Fu et al., 2020) benchmark, covering 3 environments (HalfCheetah, Hopper, Walker2d) and 3 difficulty levels (Medium, Medium-Replay, Medium-Expert).

**Baselines.** We compare SimbaV2 against standard offline RL methods: Percentile BC, Decision Transformer (DT, (Chen et al., 2021a)), Diffusion Q-Learning (DQL, (Wang et al., 2022)), Implicit Diffusion Q-Learning (IDQL, (Hansen-Estruch et al., 2023)), Conservative Q-Learning (CQL, (Chen et al., 2021a)), TD3+BC (Fujimoto & Gu, 2021), Implicit Q-Learning (IQL, (Kostrikov et al., 2021)), Extreme Q-Learning ($\mathcal{X}$-QL, (Garg et al., 2023)), and TD7+BC (Fujimoto et al., 2023).

The results for Percentile BC, DT, DQL, and IDQL is from (Hansen-Estruch et al., 2023), while CQL, TD3+BC, IQL, $\mathcal{X}$-QL, and TD7 results come from (Fujimoto et al., 2023).

**Metrics.** Following the standard offline RL protocol (Fu et al., 2020), we normalize the score of each environment based on the expert trajectory in the dataset.

**Training.** We use the same training configuration as in online RL (Appendix C), with a learning rate decaying linearly from $1 \times 10^{-4}$ to $1 \times 10^{-5}$ over 100 epochs, and include an additional behavioral cloning loss.

## D.2. Results

Table 4 reports the performance of SimbaV2 + BC, averaged over 10 random seeds.

*Table 4.* **Offline RL.** Average final performance on the D4RL mujoco benchmark, averaged over 10 trials. Methods are listed in chronological order. For a fair comparison, we used a unified hyperparameter configuration for each method. The highest performance is highlighted. Any performance that is not statistically worse than the highest performance (according to Welch's $t$-test with significance level 0.05) is highlighted. The environment's average variance was used for the statistical test for methods without reported variance.

| Method | HalfCheetah | | | Hopper | | | Walker2d | | | Average |
|---|---|---|---|---|---|---|---|---|---|---|
| | m | m-r | m-e | m | m-r | m-e | m | m-r | m-e | - |
| CQL (Kumar et al., 2020) | 46.7±0.3 | 45.5±0.3 | 76.8±7.4 | 59.3±3.3 | 78.8±10.9 | 79.9±19.8 | 81.4±1.7 | 79.9±3.6 | 108.5±1.2 | 73.0±2.7 |
| Percent BC (Chen et al., 2021a) | 48.4 | 40.6 | 92.9 | 56.9 | 75.9 | 110.9 | 75.0 | 62.5 | 109.0 | 74.0 |
| DT (Chen et al., 2021a) | 42.6 | 36.6 | 86.8 | 67.6 | 82.7 | 110.9 | 74.0 | 66.6 | 108.1 | 74.7 |
| TD3+BC (Fujimoto & Gu, 2021) | 48.1±0.1 | 44.6±0.4 | 93.7±0.9 | 59.1±3.0 | 52.0±10.6 | 98.1±10.7 | 84.3±0.8 | 81.0±3.4 | 110.5±0.4 | 74.6±1.7 |
| IQL (Kostrikov et al., 2021) | 47.4±0.2 | 43.9±1.3 | 89.6±3.5 | 63.9±4.9 | 93.4±7.8 | 64.2±32.0 | 84.2±1.6 | 71.2±8.3 | 108.9±1.4 | 74.1±3.8 |
| DQL (Wang et al., 2022) | 50.6 | 45.8 | 93.3 | 75.2 | 94.5 | 102.1 | 83.4 | 86.7 | 109.6 | 82.4 |
| $\mathcal{X}$-QL (Garg et al., 2023) | 47.4±0.1 | 44.2±0.7 | 90.2±2.7 | 67.7±3.6 | 82.0±14.9 | 92.0±10.0 | 79.2±4.0 | 61.8±7.7 | 110.3±0.2 | 75.0±2.3 |
| IDQL (Hansen-Estruch et al., 2023) | 49.7 | 45.1 | 94.4 | 63.1 | 82.4 | 105.3 | 80.2 | 79.8 | 111.6 | 79.1 |
| TD7+BC (Fujimoto et al., 2023) | 58.0±0.4 | 53.8±0.8 | 104.6±1.6 | 76.1±5.1 | 91.1±8.0 | 108.2±4.8 | 91.1±7.8 | 89.7±4.7 | 111.8±0.6 | 87.2±1.6 |
| SimbaV2+BC (ours) | 54.8±0.5 | 48.6±0.8 | 92.2±1.4 | 98.1±2.4 | 99.9±0.6 | 106.2±1.5 | 82.7±10.3 | 87.7±2.1 | 110.6±0.6 | 86.7±1.6 |

With minimal changes, SimbaV2 performs highly competitively with existing offline RL algorithms, with statistically significantly better performance on Hopper. Again, this experimental results reinforces the importance of architectural design over complex algorithmic modifications. We believe our architectural approach offers exciting future potential for bridging offline and online RL (Ball et al., 2023; Zhou et al., 2024).

# E. Baselines

**PPO** (Lillicrap, 2015). Proximal Policy Optimization (PPO) is an on-policy policy gradient method that constrains updated policies to remain proximal to the old policies to circumvent performance collapse. Results for Gym - MuJoCo and DMC were obtained from Fujimoto et al. (2025), which are averaged over 10 seeds.

**SAC** (Haarnoja et al., 2018). Soft Actor-Critic (SAC) is an off-policy actor-critic algorithm in which the actor simultaneously maximizes expected return and entropy, encouraging both stability and exploration. For the MuJoCo tasks, results averaged over 10 random seeds were obtained directly from the Bhatt et al. (2024) authors, with the update-to-data (UTD) ratio set to 1. For DMC, MyoSuite, and HBench tasks, we use the results from Lee et al. (2024c) which were obtained by running the official repository for 10 random seeds, with the update-to-data (UTD) ratio set to 2.

**TD3** (Fujimoto et al., 2018). Twin Delayed DDPG (TD3) is an off-policy actor-critic algorithm that mitigates Q-overestimation bias via three key techniques: (i) clipped double Q-learning, (ii) delayed policy updates, (iii) target policy smoothing. Results for Gym-MuJoCo were obtained from Table 1 of Fujimoto et al. (2023). These scores are averaged over 10 random seeds.

**TD3+OFE** (Ota et al., 2020). By replacing the encoder with an Online Feature Extractor (OFE)—trained via a dynamics prediction task to produce high-dimensional representations of observation-action pairs—TD3+OFE outperforms the original TD3 without requiring any hyperparameter adjustments. Results for Gym-MuJoCo were obtained from Table 1 of Fujimoto et al. (2023). These scores are averaged over 10 random seeds. We attach these results into Table 1 by TD3-normalizing the scores as outlined in Appendix F.1.

**TQC** (Kuznetsov et al., 2020). Truncated Quantile Critic (TQC) proposes to truncate the return distribution of the distributional critics to flexibly balance between under- and overestimation bias of Q-value. Results for Gym-MuJoCo were taken directly from Table 1 of Fujimoto et al. (2023). We attach these results into Table 1 by TD3-normalizing the scores as described in Appendix F.1.

**REDQ** (Chen et al., 2021b). Randomized Ensembled Double Q-Learning (REDQ) expands clipped double Q-learning from two Q-networks to an ensemble of ten to control estimation bias and variance, and enhance training stability. For the MuJoCo tasks, results averaged over 10 random seeds were obtained directly from the Bhatt et al. (2024) authors, with the update-to-data (UTD) ratio set to 20.

**DroQ** (Chen et al., 2021b). Dropout Q-Function (DroQ) reduces the computational burden of REDQ by using a smaller ensemble of Q functions while employing Dropout and Layer Normalization to stabilize training against Dropout-induced noise. For the MuJoCo tasks, results averaged over 10 random seeds were obtained directly from the Bhatt et al. (2024) authors, with the update-to-data (UTD) ratio set to 20.

**DreamerV3** (Hafner et al., 2023). DreamerV3 encodes sensory inputs into categorical representations to build a learned world model, enabling long-horizon behavior learning in its compact latent space. Results for Gym-MuJoCo and DMC were obtained from Fujimoto et al. (2025), which are averaged over 10 seeds. For MyoSuite, and HBench tasks, we use the results from Lee et al. (2024c) which were obtained by running the official repository (`https://github.com/SonyResearch/simba`) over 3 random seeds.

**TD7** (Fujimoto et al., 2023). TD7 improves TD3 by combining TD3 with four key improvements: (i) state-action representation learning (SALE), (ii) prioritized experience replay, (iii) policy checkpoints, and (iv) additional behavior cloning loss for offline RL. Results for Gym-MuJoCo and DMC were obtained from Fujimoto et al. (2025), which are averaged over 10 seeds. For MyoSuite, and HBench tasks, we use the results from Lee et al. (2024c) which were obtained by running the official repository (`https://github.com/SonyResearch/simba`) over 5 random seeds.

**TD-MPC2** (Hansen et al., 2023). TD-MPC2 is a model-based algorithm that learns an implicit (decoder-free) world model through multiple dynamics prediction tasks and performs local trajectory optimization within the learned latent space. Results for Gym-MuJoCo and DMC were obtained from Fujimoto et al. (2025), which are averaged over 10 seeds. For MyoSuite, and HBench tasks, we use the results from Lee et al. (2024c) which were obtained by running the official repository (`https://github.com/SonyResearch/simba`) over 3 random seeds.

**CrossQ** (Bhatt et al., 2024). CrossQ achieves superior performance and sample efficiency with low replay ratio, by removing target networks and employing careful batch normalization. Results for Gym-MuJoCo were obtained by running the official repository (`https://github.com/adityab/CrossQ`) for 10 random seeds,

**iQRL** (Scannell et al., 2024). Implicitly Quantized Reinforcement Learning (iQRL) is a representation learning technique of model-free RL that prevents representation collapse and improve sample-efficiency via latent quantization. For the DMC hard tasks, results averaged over 3 random seeds were obtained directly from the authors.

**BRO** (Nauman et al., 2024b). Bigger, Regularized, Optimistic (BRO) scales the critic network of SAC by integrating distributional Q-learning, optimistic exploration, and periodic resets. Results for Gym-MuJoCo and DMC Easy were obtained by running the official repository (`https://github.com/naumix/BiggerRegularizedOptimistic`) for 5 random seeds. For DMC hard, MyoSuite, and HBench tasks, we use the results from Lee et al. (2024c) which were obtained by running the official repository (`https://github.com/SonyResearch/simba`) over 5 random seeds for HBench tasks and 10 random seeds for DMC hard and MyoSuite tasks. Unless stated otherwise, we set update-to-data (UTD) ratio to be 2.

**MAD-TD** (Voelcker et al., 2024). Model-Augmented Data for Temporal Difference learning (MAD-TD) aims to stabilize high UTD training by mixing a small fraction $\alpha$ of model-generated on-policy data with real off-policy replay data. For the DMC hard tasks, results averaged over 10 random seeds were obtained directly from the authors using the best algorithm setting (UTD = 8, $\alpha = 0.05$).

**MR.Q** (Fujimoto et al., 2025). Model-based Representations for Q-learning (MR.Q) is a model-free algorithm that uses model-based objectives, such as dynamics and reward prediction, to obtain rich representation for actor-critic agent. We use the results for Gym-MuJoCo and DMC from Fujimoto et al. (2025) which were obtained by running the official repository (`https://github.com/facebookresearch/MRQ`) over 10 random seeds.

**Simba** (Lee et al., 2024c). SimBa is an architecture designed to scale up parameters in deep reinforcement learning by injecting a simplicity bias with observation normalizer, residual blocks, and layer normalizations. For Gym-MuJoCo, DMC, MyoSuite, and HBench tasks, we use the results from Lee et al. (2024c) which were obtained by running the official repository (`https://github.com/SonyResearch/simba`) over 15 random seeds for DMC hard tasks and 10 random seeds otherwise. Unless stated otherwise, we set update-to-data (UTD) ratio to be 2.

*Table 5.* **Environment details.** We list the episode length, action repeat for each domain, total environment steps, and performance metrics used for benchmarking SimbaV2.

|                    | **Gym**        | **DMC**        | **MyoSuite**    | **HumanoidBench** |
|--------------------|----------------|----------------|-----------------|-------------------|
| Episode length     | $1,000$        | $1,000$        | $100$           | $500 - 1,000$     |
| Action repeat      | 1              | 2              | 2               | 2                 |
| Effective length   | $1,000$        | 500            | 50              | $250 - 500$       |
| Total env. steps   | 1M             | 1M             | 1M              | 1M                |
| Performance metric | Average Return | Average Return | Average Success | Average Return    |

# F. Environment Details

This section outlines the benchmark environments used in our evaluation. A complete list of all tasks from each benchmark, including their observation and action dimensions, is provided at the end of this section. Additionally, Table 5 outlines the episode length, action repeat, total number of environment steps, and performance metrics for each task domain.

## F.1. Gym - MuJoCo

Gym (Brockman, 2016; Towers et al., 2024) is a suite of benchmark environments spanning finite MDPs to Multi-Joint dynamics with Contact (Todorov et al., 2012, MuJoCo) simulations. It offers a diverse range of tasks, including classic Atari games, small-scale tasks such as Toy Text and classic controls, as well as physics-based continuous robot control. For our experiments, we focus on 5 locomotion tasks within MuJoCo environments, which simulate complex physical interactions involving multi-body dynamics and contact forces. A complete list of these tasks is provided in Table 6. Note that we use the `v4` version.

For comparison across different score scales of each task, all MuJoCo scores are normalized using TD3 and the random score for each task, as provided in TD7 (Fujimoto et al., 2023).

$$\text{TD3-Normalized}(x) := \frac{x - \text{random score}}{\text{TD3 score} - \text{random score}}$$

| **Task**        | **Random**  | **TD3** |
|-----------------|-------------|---------|
| `Ant-v4`        | $-70.288$   | 3942    |
| `HalfCheetah-v4`| $-289.415$  | 10574   |
| `Hopper-v4`     | $18.791$    | 3226    |
| `Humanoid-v4`   | $120.423$   | 5165    |
| `Walker2d-v4`   | $2.791$     | 3946    |

## F.2. DeepMind Control Suite

DeepMind Control Suite (Tassa et al., 2018, DMC) is a standard continuous control benchmarks, encompassing a variety of locomotion and manipulation tasks with varying levels of complexity. These tasks range from simple low-dimensional settings ($\mathcal{O} \in \mathbb{R}^3$, $\mathcal{A} \in \mathbb{R}^1$) to highly complex scenarios ($\mathcal{O} \in \mathbb{R}^{223}$, $\mathcal{A} \in \mathbb{R}^{38}$). Our evaluation includes 27 DMC tasks, divided into two categories: DMC-Easy&Medium and DMC-Hard. All Humanoid and Dog tasks are grouped as DMC-Hard, while the rest are fall under DMC-Easy&Medium. Comprehensive lists of DMC-Easy&Medium and DMC-Hard are available in Tables 7 and 8, respectively.

## F.3. MyoSuite

MyoSuite (Caggiano et al., 2022) models human motor control using musculoskeletal simulations of the human elbow, wrist, and hand, focusing on physiologically accurate movements. It provides benchmarks for intricate real-world object manipulation, ranging from simple posing tasks to the simultaneous manipulation of two Baoding balls. Our evaluation

focuses on 10 MyoSuite tasks involving the hand. As defined by the authors, each task is categorized as `hard` when the goal is randomized; otherwise the goal is fixed. The full list of MyoSuite tasks is presented in Table 9.

### F.4. HumanoidBench

HumanoidBench (Sferrazza et al., 2024) serves as a high-dimensional simulated robot learning benchmark, leveraging the Unitree H1 humanoid robot equipped with dexterous hands. It encompasses a diverse set of whole-body control tasks, spanning from fundamental locomotion to complex human-like activities that require refined manipulation. In our experiments, we concentrate on 14 locomotion tasks. A comprehensive list of tasks is provided in Table 10.

Note that the locomotion tasks do not necessitate hand dexterity. Therefore, to reduce the complexity arising from high degrees of freedom (DoF) and complex dynamics, we streamline the environments setup by excluding the hands of humanoid. For example, in case of `walk`, this drastically declines the dimension of the observation and action spaces by approximately 66%.

| `walk` | **Without hand** | **With 2 hand** |
|---|---|---|
| Observation dim $|\mathcal{O}|$ | 51 | 151 |
| Action dim $|\mathcal{A}|$ | 19 | 61 |
| DoF (body) | 25 | 25 |
| DoF (two hands) | 0 | 50 |

For comparison across different score scales of each task, all HumanoidBench scores are normalized using each task's target success score provided by the authors and random score. Random scores are measured by the average undiscounted returns over 10 episodes of random agent. Each measurement is repeated over 10 seeds.

$$\text{Success-Normalized}(x) := \frac{x - \text{random score}}{\text{Target success score} - \text{random score}}$$

| Task | Random | Target Success |
|---|---|---|
| h1-balance-simple | 9.391 | 800 |
| h1-balance-hard | 9.044 | 800 |
| h1-crawl | 272.658 | 700 |
| h1-hurdle | 2.214 | 700 |
| h1-maze | 106.441 | 1200 |
| h1-pole | 20.09 | 700 |
| h1-reach | 260.302 | 12000 |
| h1-run | 2.02 | 700 |
| h1-sit-simple | 9.393 | 750 |
| h1-sit-hard | 2.448 | 750 |
| h1-slide | 3.191 | 700 |
| h1-stair | 3.112 | 700 |
| h1-stand | 10.545 | 800 |
| h1-walk | 2.377 | 700 |

*Table 6.* **Gym-MuJoCo.** We evaluate a total of 5 continuous control tasks from the Gym-MuJoCo benchmark. Below, we provide a list of all the tasks considered. The baseline performance for each task is reported at 1M environment steps.

| Task | Observation dim $|\mathcal{O}|$ | Action dim $|\mathcal{A}|$ |
|------|------|------|
| Ant-v4 | 27 | 8 |
| HalfCheetah-v4 | 17 | 6 |
| Hopper-v4 | 11 | 3 |
| Humanoid-v4 | 376 | 17 |
| Walker2d-v4 | 17 | 6 |

*Table 7.* **DMC-Easy Complete List.** We evaluate a total of 21 continuous control tasks from the DMC-Easy benchmark. Below, we provide a list of all the tasks considered. The baseline performance for each task is reported at 1M environment steps.

| Task | Observation dim $|\mathcal{O}|$ | Action dim $|\mathcal{A}|$ |
|------|------|------|
| acrobot-swingup | 6 | 1 |
| ball-in-cup-catch | 6 | 1 |
| cartpole-balance | 5 | 1 |
| cartpole-balance-sparse | 5 | 1 |
| cartpole-swingup | 5 | 1 |
| cartpole-swingup-sparse | 5 | 1 |
| cheetah-run | 17 | 6 |
| finger-spin | 9 | 2 |
| finger-turn-easy | 12 | 2 |
| finger-turn-hard | 12 | 2 |
| fish-swim | 24 | 5 |
| hopper-hop | 15 | 4 |
| hopper-stand | 15 | 4 |
| pendulum-swingup | 3 | 1 |
| quadruped-run | 78 | 12 |
| quadruped-walk | 78 | 12 |
| reacher-easy | 6 | 2 |
| reacher-hard | 6 | 2 |
| walker-run | 24 | 6 |
| walker-stand | 24 | 6 |
| walker-walk | 24 | 6 |

*Table 8.* **DMC-Hard Complete List.** We evaluate a total of 7 continuous control tasks from the DMC-Hard benchmark. Below, we provide a list of all the tasks considered. The baseline performance for each task is reported at 1M environment steps.

| Task | Observation dim $|\mathcal{O}|$ | Action dim $|\mathcal{A}|$ |
|------|------|------|
| dog-run | 223 | 38 |
| dog-trot | 223 | 38 |
| dog-stand | 223 | 38 |
| dog-walk | 223 | 38 |
| humanoid-run | 67 | 24 |
| humanoid-stand | 67 | 24 |
| humanoid-walk | 67 | 24 |

*Table 9.* **MyoSuite Complete List.** We evaluate a total of 10 continuous control tasks from the MyoSuite benchmark including both fixed-goal and randomized-goal (`hard`) settings. Below, we provide a list of all the tasks considered. The baseline performance for each task is reported at 1M environment steps.

| Task | Observation dim $|\mathcal{O}|$ | Action dim $|\mathcal{A}|$ |
|---|---|---|
| `myo-key-turn` | 93 | 39 |
| `myo-key-turn-hard` | 93 | 39 |
| `myo-obj-hold` | 91 | 39 |
| `myo-obj-hold-hard` | 91 | 39 |
| `myo-pen-twirl` | 83 | 39 |
| `myo-pen-twirl-hard` | 83 | 39 |
| `myo-pose` | 108 | 39 |
| `myo-pose-hard` | 108 | 39 |
| `myo-reach` | 115 | 39 |
| `myo-reach-hard` | 115 | 39 |

*Table 10.* **HumanoidBench Complete List.** We evaluate a total of 14 continuous control locomotion tasks from the HumanoidBench benchmark that simulates the UniTree H1 humanoid robot. Below, we provide a list of all the tasks considered. The baseline performance for each task is reported at 1M environment steps.

| Task | Observation dim $|\mathcal{O}|$ | Action dim $|\mathcal{A}|$ |
|---|---|---|
| `h1-balance-hard` | 77 | 19 |
| `h1-balance-simple` | 64 | 19 |
| `h1-crawl` | 51 | 19 |
| `h1-hurdle` | 51 | 19 |
| `h1-maze` | 51 | 19 |
| `h1-pole` | 51 | 19 |
| `h1-reach` | 57 | 19 |
| `h1-run` | 51 | 19 |
| `h1-sit-simple` | 51 | 19 |
| `h1-sit-hard` | 64 | 19 |
| `h1-slide` | 51 | 19 |
| `h1-stair` | 51 | 19 |
| `h1-stand` | 51 | 19 |
| `h1-walk` | 51 | 19 |

# G. Training Stability

In Section 5.2, we investigated the training dynamics of SimbaV2 on DMC-Hard and HBench-Hard via four metrics: feature norm, parameter norm, gradient norm, and effective learning rate (ELR) of neural networks. This section presents these standalone metrics for SimbaV2 to highlight its stable behavior throughout training.

**Effective Learning Rate.** We base our notion of ELR on the *effective step size* of Kodryan et al. (2022), omitting the global learning rate $\eta$ and using dimension-based weighting $w_i = \frac{|\boldsymbol{\theta}_i|}{\sum_{j=1}^N |\boldsymbol{\theta}_j|}$ instead of squared-parameter-norm weighting $w_i = \frac{\|\boldsymbol{\theta}_i\|^2}{\sum_{j=1}^N \|\boldsymbol{\theta}_j\|^2}$.

**Definition G.1** (Effective Learning Rate). Let $\boldsymbol{\theta} = \{\boldsymbol{\theta}_i\}_{i=1}^N$ be the parameter set of a neural network, and $\boldsymbol{g}_i$ be the back-propagated gradient associated with $\boldsymbol{\theta}_i$. The *(total) effective learning rate* ELR of the network is defined as:

$$\text{ELR} \triangleq \sqrt{\sum_{i=1}^N w_i \frac{\|\boldsymbol{g}_i\|^2}{\|\boldsymbol{\theta}_i\|^2}} \tag{41}$$

where $w_i = \frac{|\boldsymbol{\theta}_i|}{\sum_{j=1}^N |\boldsymbol{\theta}_j|}$. Intuitively, our ELR measures the "effective" gradient step—per parameter dimension—before scaled by the global learning rate.

**Metrics.** To reflect dimensional contributions across layers, we also apply the same weighting $w_i$ when computing the feature norm, parameter norm, and gradient norm. For instance, our gradient norm is defined as:

$$\|\boldsymbol{g}_i\|^2 \triangleq \sum_{i=1}^N w_i \|\boldsymbol{g}_i\|_2^2 \tag{42}$$

where $\|\cdot\|_2$ is the standard $\ell_2$-norm (Frobenius norm $\|\cdot\|_F$ in case of matrices). Analogous expressions are applied for feature and parameter norms. We separate encoder layers (all layers preceding the output) from predictor layers (all layers after) to capture their distinct roles in the network. Average returns are normalized by maximum score 1000 for DMC-Hard, by success and random scores for HBench-Hard (Appendix F.4).

**Results.** Figure 9 shows the tracked metrics over 1 million training steps. Certain features (e.g., logits) and parameters (e.g., scalers and interpolation vectors) may occasionally exceed unit norm, the overall parameter norms are tightly controlled (Figure 9.(b)-(c)), and gradient magnitudes are consistently balanced across modules (Figure 9.(d)). This leads to the consistent trend and scales of their ELRs over time. We hypothesize that this stable behavior contributes to SIMBAV2's improved performance and scalability.

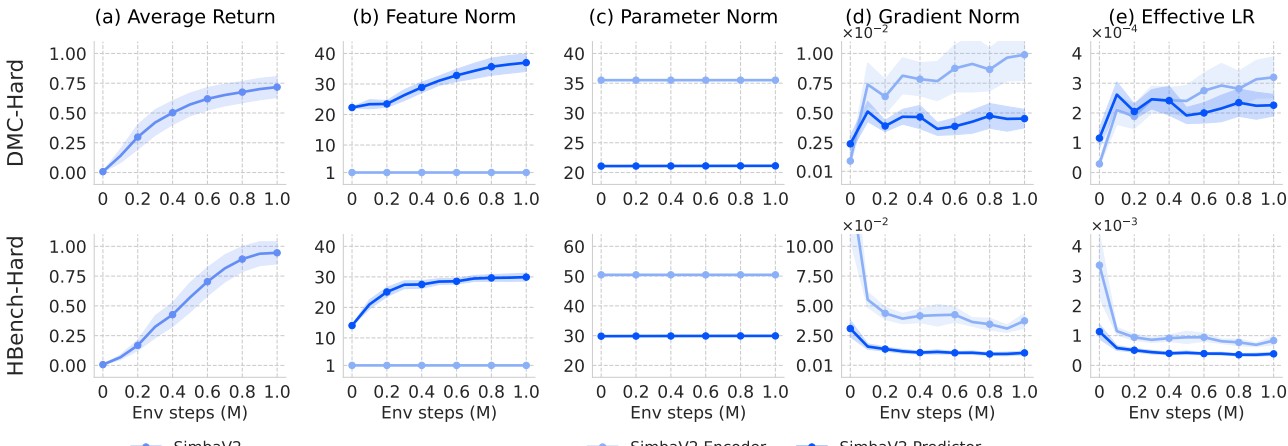

*Figure 9.* **SimbaV2 Training Dynamics.** We track 4 metrics during training to understand the learning dynamics of SimbaV2: **(a)** Average normalized return across tasks. **(b)** Weighted sum of $\ell_2$-norms of all intermediate features in critic. **(c)** Weighted sum of $\ell_2$-norms of all critic parameters **(d)** Weighted sum of $\ell_2$-norms of all gradients in critic **(e)** Effective learning rate (ELR) of the critic. On both environments, SimbaV2 maintains feature and parameter norms aligned, producing consistent gradient norms and ELRs.

# H. Additional Experiments

This section complements Section 5.2 by presenting further experiments probing the properties and robustness of SimbaV2:

- **Scalability Effect of Hyperspherical Normalization** (Section H.1). Investigate the necessity of hyperspherical normalization for achieving SimbaV2's scalability.

- **Effectiveness beyond SAC** (Section H.2.) Assess SimbaV2's broader applicability by substituting SAC with DDPG.

## H.1. Scalability Effect of Hyperspherical Normalization

In Section 5.3, we observe that SimbaV2 consistently scales with an increasing update-to-data (UTD) ratio, even without reinitialization, while Simba saturates at a ratio of 2. However, this raises the question of whether hyperspherical normalization is critical for UTD scaling. This section investigates the effectiveness of hyperspherical normalization in scalability.

**Experimental Setup.** In this experiment, we examine a "Simba-like" variant, named Simba+, which incorporates distributional critic and reward scaling but only excludes the hyperspherical normalization. In other words, Simba+ is identical to SimbaV2 except that it excludes hyperspherical normalization. On DMC-Hard tasks, we compare SimbaV2, Simba, and Simba+ under varying model sizes and UTD ratios to determine the role of hyperspherical normalization in scaling performance.

**Result.** Figure 10 shows the scaling results. In Figure 10 (left), all three methods benefit from increased model capacity, but Simba+ slightly underperforms at larger parameter counts. More critically, Simba and Simba+ both plateau when the UTD ratio surpasses 2 in Figure 10 (right), while SimbaV2 continues to improve. These results confirm that hyperspherical normalization is truly indispensable for UTD scaling.

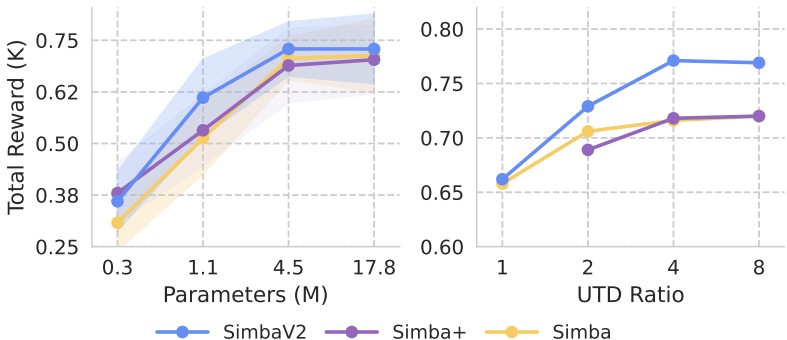

*Figure 10.* **Performance Scaling under DMC-Hard.** We compare SimbaV2, Simba+, and Simba as scaling the number of model parameters by increasing the critic network width and UTD ratio. Simba+ fails to scale effectively at higher UTD ratios, highlighting the essential role of the hyperspherical normalization for scalability.

## H.2. Effectiveness beyond SAC

In Section 5.2, we observe that replacing the neural network of SAC with SimbaV2 consistently improves the performance of a wide range of domains. To assess the broader applicability and robustness of SimbaV2's architectural advantages beyond a single algorithm, we conducted additional experiments using Deep Deterministic Policy Gradient (Lillicrap, 2015, DDPG), another widely adopted off-policy algorithm for continuous control.

**Experimental Setup.** We evaluated SimbaV2 against the original Simba (Lee et al., 2024c) and a standard MLP baseline on two challenging continuous control benchmark suites: DMC-Hard and HBench-Hard. All methods utilized the DDPG algorithm as their underlying learning framework. The MLP architecture we adopted consists of a sequence of linear layers followed by ReLU non-linearities.

**Result.** Comparative results are presented in Figure 11. On the DMC-Hard benchmark, SimbaV2 achieved performance competitive with the original Simba, with both significantly outperforming the MLP baseline. More notably, on the more complex HBench-Hard benchmark, SimbaV2 demonstrated a clear improvement over Simba. These results indicate that SimbaV2 not only generalizes to the DDPG algorithm but also exhibits enhanced stability and generalization capabilities in more demanding environments, likely attributable to its refined architecture and regularization mechanisms.

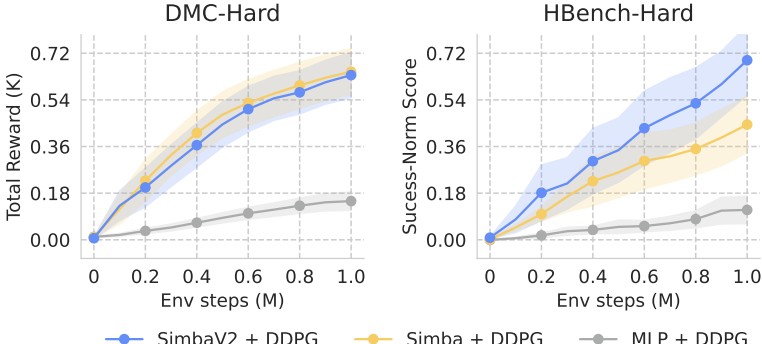

*Figure 11.* **DDPG with SimbaV2.** Learning curves of SimbaV2, Simba (Lee et al., 2024c), and the MLP baseline on DMC-Hard and HBench-Hard benchmarks using DDPG (Lillicrap, 2015). SimbaV2 performs competitively with Simba on DMC-Hard, both significantly outperforming the MLP baseline. In the more challenging HBench-Hard, SimbaV2 shows clear improvements over Simba, indicating enhanced stability and generalization beyond SAC.

# I. Complete UTD Scaling Results

## I.1. Gym - MuJoCo

*Table 11.* **Gym - MuJoCo UTD Scaling Results.** Final average performance at 1M environment steps for each of the 5 locomotion tasks in the Gym - MuJoCo benchmark. The number of evaluated random seeds for each update-to-data (UTD) ratio is 5. The values in [brackets] represent a 95% bootstrap confidence interval. The aggregate mean, median and interquartile mean (IQM) are computed over the TD3-normalized score as described in Appendix F.1.

| Task | UTD = 1 | UTD = 2 | UTD = 4 | UTD = 8 |
|---|---|---|---|---|
| `Ant-v4` | 7405 [7315, 7496] | 7429 [7209, 7649] | 7230 [6968, 7492] | 6940 [6431, 7449] |
| `HalfCheetah-v4` | 11425 [10798, 12052] | 12022 [11640, 12404] | 12007 [11458, 12557] | 11592 [9956, 13229] |
| `Hopper-v4` | 3579 [3311, 3847] | 4053 [3928, 4178] | 4003 [3647, 4359] | 4151 [4033, 4269] |
| `Humanoid-v4` | 7696 [4385, 11008] | 10545 [10195, 10896] | 11133 [10908, 11358] | 11703 [11282, 12125] |
| `Walker2d-v4` | 6069 [5724, 6414] | 6938 [6691, 7185] | 6804 [6459, 7148] | 6163 [4522, 7804] |
| IQM | 1.433 [1.225, 1.648] | 1.637 [1.471, 1.788] | 1.617 [1.402, 1.83] | 1.581 [1.358, 1.82] |
| Median | 1.468 [1.269, 1.625] | 1.616 [1.491, 1.743] | 1.615 [1.438, 1.809] | 1.602 [1.377, 1.821] |
| Mean | 1.418 [1.264, 1.569] | 1.617 [1.513, 1.719] | 1.62 [1.47, 1.773] | 1.598 [1.419, 1.78] |

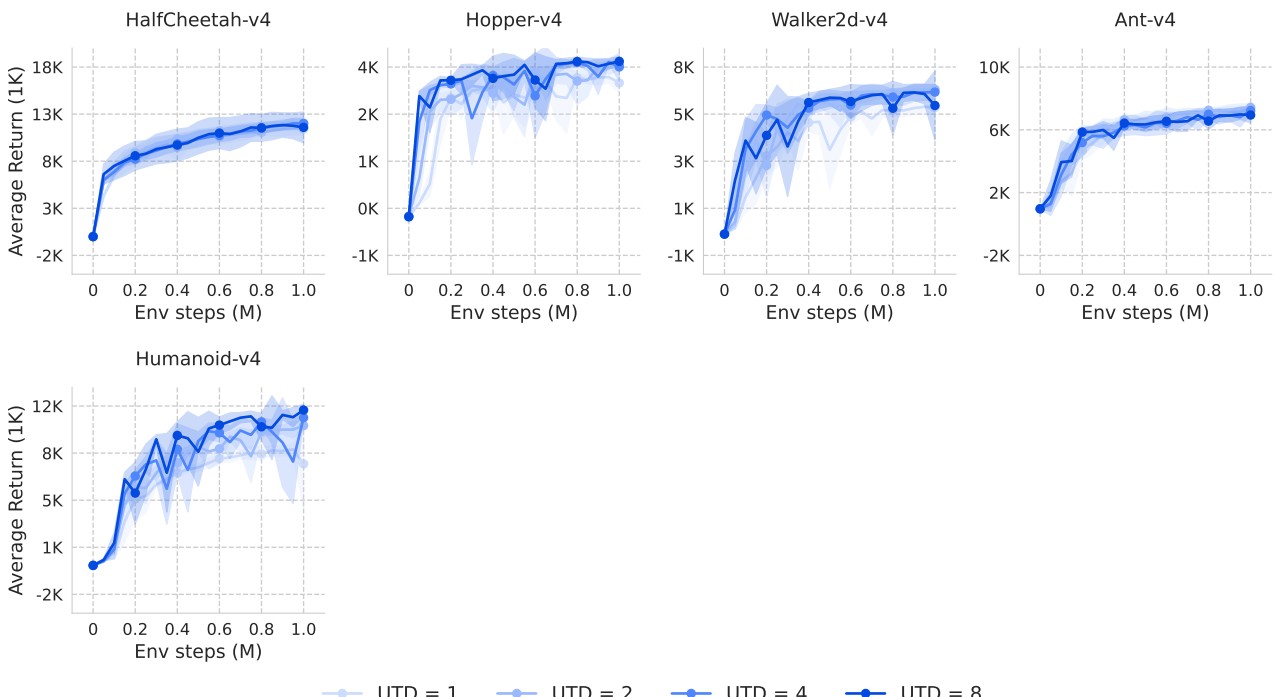

*Figure 12.* **Gym-MuJoCo UTD Scaling Learning Curves.** Average episode return (1k) for the Gym-MuJoCo environment. Results are averaged over 5 random seeds, and the shaded areas indicate 95% bootstrap confidence intervals.

## I.2. Deepmind Control Suite - Easy

*Table 12.* **DMC-Easy UTD Scaling Results.** Final average performance at 1M environment steps for each of the 21 tasks of the DMC-Easy benchmark. The number of evaluated random seeds for each update-to-data (UTD) ratio is provided 5. The values in [brackets] represent a 95% bootstrap confidence interval. The aggregate mean, median and interquartile mean (IQM) are reported in units of 1k.

| Task | UTD = 1 | UTD = 2 | UTD = 4 | UTD = 8 |
|---|---|---|---|---|
| acrobot-swingup | 413 [376, 450] | 436 [391, 482] | 458 [359, 558] | 477 [438, 516] |
| ball-in-cup-catch | 981 [977, 985] | 982 [980, 984] | 982 [979, 986] | 982 [979, 985] |
| cartpole-balance | 999 [999, 999] | 999 [999, 999] | 999 [999, 999] | 999 [999, 999] |
| cartpole-balance-sparse | 1000 [1000, 1000] | 967 [904, 1030] | 1000 [1000, 1000] | 1000 [1000, 1000] |
| cartpole-swingup | 881 [881, 881] | 880 [876, 883] | 880 [879, 881] | 881 [880, 882] |
| cartpole-swingup-sparse | 845 [843, 848] | 848 [848, 849] | 848 [848, 849] | 841 [824, 858] |
| cheetah-run | 917 [913, 920] | 920 [918, 922] | 902 [868, 937] | 916 [912, 920] |
| finger-spin | 940 [895, 985] | 891 [810, 972] | 762 [608, 915] | 910 [790, 1030] |
| finger-turn-easy | 951 [916, 987] | 953 [925, 980] | 954 [917, 992] | 936 [857, 1014] |
| finger-turn-hard | 928 [885, 972] | 951 [925, 977] | 902 [866, 939] | 950 [910, 990] |
| fish-swim | 818 [779, 856] | 826 [806, 846] | 815 [780, 850] | 807 [778, 836] |
| hopper-hop | 379 [224, 535] | 290 [233, 348] | 326 [243, 410] | 317 [230, 404] |
| hopper-stand | 845 [704, 986] | 944 [926, 962] | 781 [449, 1112] | 932 [898, 967] |
| pendulum-swingup | 817 [776, 858] | 827 [805, 849] | 820 [781, 859] | 821 [784, 859] |
| quadruped-run | 931 [922, 940] | 935 [928, 943] | 943 [936, 949] | 935 [930, 940] |
| quadruped-walk | 962 [955, 970] | 962 [955, 969] | 964 [958, 971] | 965 [958, 972] |
| reacher-easy | 963 [927, 1000] | 983 [979, 986] | 975 [958, 992] | 983 [981, 985] |
| reacher-hard | 975 [971, 980] | 967 [946, 987] | 976 [972, 980] | 974 [970, 978] |
| walker-run | 813 [806, 819] | 817 [812, 821] | 821 [819, 823] | 802 [774, 831] |
| walker-stand | 986 [980, 992] | 987 [984, 990] | 988 [984, 992] | 987 [984, 991] |
| walker-walk | 977 [976, 979] | 976 [974, 978] | 976 [973, 979] | 976 [972, 981] |
| IQM | 0.928 [0.906, 0.948] | 0.933 [0.918, 0.948] | 0.925 [0.9, 0.946] | 0.935 [0.91, 0.956] |
| Median | 0.876 [0.834, 0.912] | 0.875 [0.846, 0.904] | 0.866 [0.818, 0.905] | 0.878 [0.835, 0.917] |
| Mean | 0.873 [0.838, 0.905] | 0.874 [0.848, 0.897] | 0.861 [0.823, 0.896] | 0.876 [0.841, 0.908] |

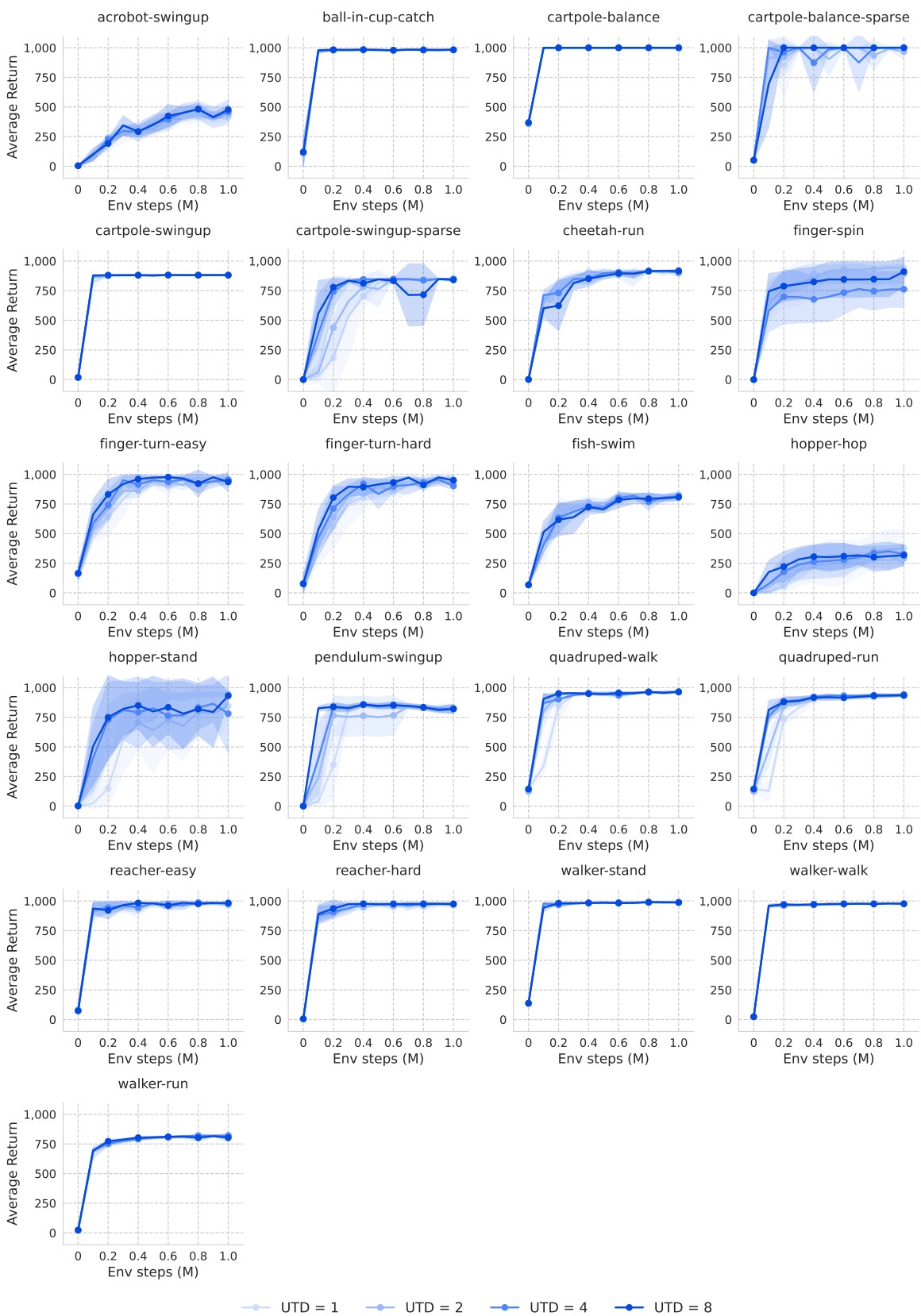

*Figure 13.* **DMC-Easy UTD Scaling Learning Curves.** Average episode return for the DMC-Easy environment. Results are averaged over 5 random seeds, and the shaded areas indicate 95% bootstrap confidence intervals.

## I.3. Deepmind Control Suite - Hard

*Table 13.* **DMC-Hard UTD Scaling Results.** Final average performance at 1M environment steps for each of the 7 tasks of the DMC-Hard benchmark. The number of evaluated random seeds for each update-to-data (UTD) ratio is provided 5. The values in [brackets] represent a 95% bootstrap confidence interval. The aggregate mean, median and interquartile mean (IQM) are reported in units of 1k.

| Task | UTD = 1 | UTD = 2 | UTD = 4 | UTD = 8 |
|---|---|---|---|---|
| dog-run | 477 [429, 525] | 562 [516, 608] | 655 [620, 691] | 555 [523, 587] |
| dog-stand | 967 [959, 974] | 981 [977, 985] | 967 [960, 974] | 972 [967, 976] |
| dog-trot | 850 [810, 890] | 861 [772, 950] | 846 [782, 910] | 898 [888, 909] |
| dog-walk | 921 [912, 930] | 935 [927, 944] | 923 [905, 941] | 949 [945, 953] |
| humanoid-run | 183 [164, 203] | 194 [182, 207] | 272 [230, 313] | 253 [228, 278] |
| humanoid-stand | 660 [585, 734] | 916 [886, 945] | 928 [926, 930] | 933 [924, 941] |
| humanoid-walk | 568 [533, 603] | 651 [590, 713] | 818 [751, 885] | 819 [762, 877] |
| IQM | 0.713 [0.598, 0.809] | 0.808 [0.725, 0.88] | 0.851 [0.755, 0.916] | 0.849 [0.727, 0.924] |
| Median | 0.666 [0.563, 0.774] | 0.729 [0.655, 0.81] | 0.771 [0.678, 0.868] | 0.767 [0.652, 0.861] |
| Mean | 0.669 [0.581, 0.753] | 0.729 [0.663, 0.791] | 0.769 [0.687, 0.845] | 0.759 [0.67, 0.84] |

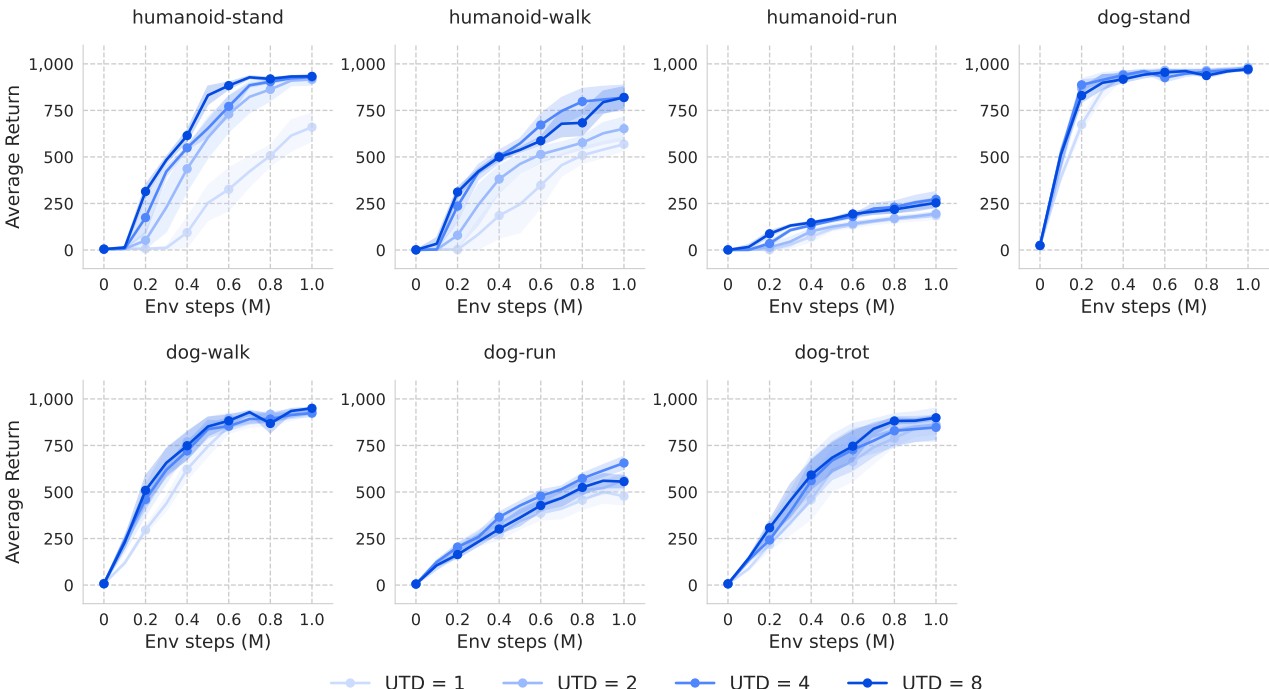

*Figure 14.* **DMC-Hard UTD Scaling Learning Curves.** Average episode return for the DMC-Hard environment. Results are averaged over 5 random seeds, and the shaded areas indicate 95% bootstrap confidence intervals.

## I.4. MyoSuite

*Table 14.* **MyoSuite UTD Scaling Results.** Final average performance at 1M environment steps across each of the 10 continuous control tasks in the MyoSuite benchmark, including both fixed-goal and randomized-goal (`hard`) settings. The number of evaluated random seeds for each update-to-data (UTD) ratio is 5. The values in [brackets] represent a 95% bootstrap confidence interval. Performance is measured by the average success rate of each task.

| Task | UTD = 1 | UTD = 2 | UTD = 4 | UTD = 8 |
|---|---|---|---|---|
| `myo-pen-twirl-hard` | 76.0 [57.8, 94.2] | 93.0 [88.8, 97.2] | 92.0 [84.7, 99.3] | 98.0 [94.1, 101.9] |
| `myo-pen-twirl` | 100.0 [100.0, 100.0] | 100.0 [100.0, 100.0] | 100.0 [100.0, 100.0] | 100.0 [100.0, 100.0] |
| `myo-key-turn-hard` | 46.0 [8.5, 83.5] | 62.0 [42.7, 81.3] | 70.0 [34.9, 105.1] | 80.0 [49.6, 110.4] |
| `myo-key-turn` | 80.0 [40.8, 119.2] | 100.0 [100.0, 100.0] | 100.0 [100.0, 100.0] | 100.0 [100.0, 100.0] |
| `myo-obj-hold-hard` | 100.0 [100.0, 100.0] | 98.0 [95.4, 100.6] | 98.0 [94.1, 101.9] | 92.0 [84.7, 99.3] |
| `myo-obj-hold` | 100.0 [100.0, 100.0] | 100.0 [100.0, 100.0] | 100.0 [100.0, 100.0] | 100.0 [100.0, 100.0] |
| `myo-pose-hard` | 0.0 [0.0, 0.0] | 0.0 [0.0, 0.0] | 0.0 [0.0, 0.0] | 0.0 [0.0, 0.0] |
| `myo-pose` | 100.0 [100.0, 100.0] | 100.0 [100.0, 100.0] | 100.0 [100.0, 100.0] | 100.0 [100.0, 100.0] |
| `myo-reach-hard` | 92.0 [84.7, 99.3] | 94.0 [87.3, 100.7] | 98.0 [94.1, 101.9] | 96.0 [91.2, 100.8] |
| `myo-reach` | 100.0 [100.0, 100.0] | 100.0 [100.0, 100.0] | 100.0 [100.0, 100.0] | 100.0 [100.0, 100.0] |
| IQM | 96.9 [85.8, 100.0] | 99.0 [96.8, 100.0] | 99.2 [96.2, 100.0] | 99.6 [96.9, 100.0] |
| Median | 79.0 [67.0, 91.0] | 84.5 [78.0, 93.0] | 87.0 [76.0, 98.0] | 88.0 [77.0, 98.0] |
| Mean | 79.4 [68.6, 88.8] | 84.7 [78.3, 90.6] | 85.8 [76.4, 94.0] | 86.6 [77.4, 94.6] |

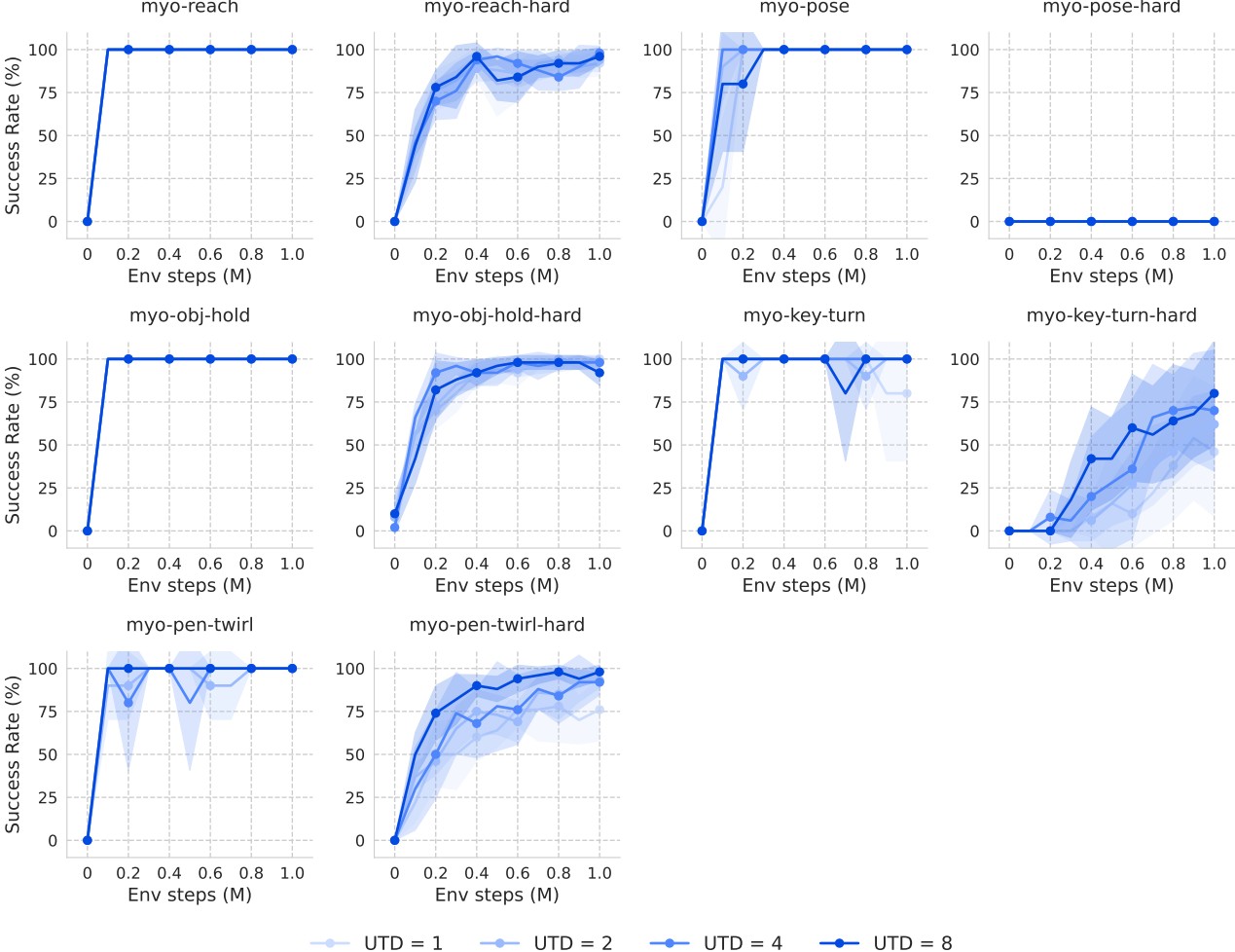

*Figure 15.* **MyoSuite UTD Scaling Learning Curves.** Average episode success rate (%) for the MyoSuite environment. Results are averaged over 5 random seeds, and the shaded areas indicate 95% bootstrap confidence intervals.

## I.5. Humanoid Bench

*Table 15.* **HumanoidBench UTD Scaling Results.** Final average performance at 1M environment steps for each of the 14 locomotion tasks in the HumanoidBench benchmark. The number of evaluated random seeds for each update-to-data (UTD) ratio is 5. The values in [brackets] represent a 95% bootstrap confidence interval. The aggregate mean, median and interquartile mean (IQM) are computed over the success normalized score as described in Appendix F.4.

| Task | UTD = 1 | UTD = 2 | UTD = 4 | UTD = 8 |
|---|---|---|---|---|
| h1-sit-hard-v0 | 681 [506, 857] | 679 [548, 811] | 719 [664, 773] | 810 [784, 836] |
| h1-walk-v0 | 732 [522, 941] | 845 [840, 850] | 846 [841, 851] | 844 [840, 847] |
| h1-stair-v0 | 473 [444, 503] | 493 [467, 518] | 546 [541, 550] | 532 [512, 552] |
| h1-run-v0 | 247 [152, 342] | 415 [307, 524] | 318 [176, 461] | 425 [293, 558] |
| h1-balance-simple-v0 | 806 [773, 839] | 723 [651, 795] | 775 [719, 831] | 813 [797, 828] |
| h1-pole-v0 | 769 [758, 780] | 791 [785, 797] | 799 [780, 817] | 827 [787, 868] |
| h1-slide-v0 | 412 [279, 544] | 487 [404, 571] | 544 [500, 588] | 534 [505, 563] |
| h1-balance-hard-v0 | 135 [111, 160] | 143 [128, 157] | 128 [118, 139] | 167 [157, 178] |
| h1-sit-simple-v0 | 873 [868, 879] | 875 [870, 880] | 908 [861, 955] | 867 [839, 894] |
| h1-maze-v0 | 350 [332, 368] | 313 [287, 340] | 343 [327, 359] | 338 [325, 351] |
| h1-crawl-v0 | 923 [884, 962] | 946 [933, 959] | 939 [927, 951] | 954 [923, 984] |
| h1-hurdle-v0 | 193 [171, 215] | 202 [167, 236] | 244 [230, 259] | 246 [206, 287] |
| h1-reach-v0 | 4166 [3706, 4627] | 3850 [3272, 4427] | 4003 [3614, 4392] | 3449 [2541, 4358] |
| h1-stand-v0 | 771 [669, 873] | 814 [770, 857] | 765 [695, 835] | 855 [828, 882] |
| IQM | 0.734 [0.574, 0.887] | 0.799 [0.685, 0.905] | 0.813 [0.657, 0.958] | 0.873 [0.706, 1.002] |
| Median | 0.71 [0.612, 0.859] | 0.781 [0.691, 0.863] | 0.782 [0.665, 0.914] | 0.824 [0.699, 0.944] |
| Mean | 0.737 [0.637, 0.836] | 0.776 [0.704, 0.846] | 0.791 [0.687, 0.893] | 0.822 [0.72, 0.92] |

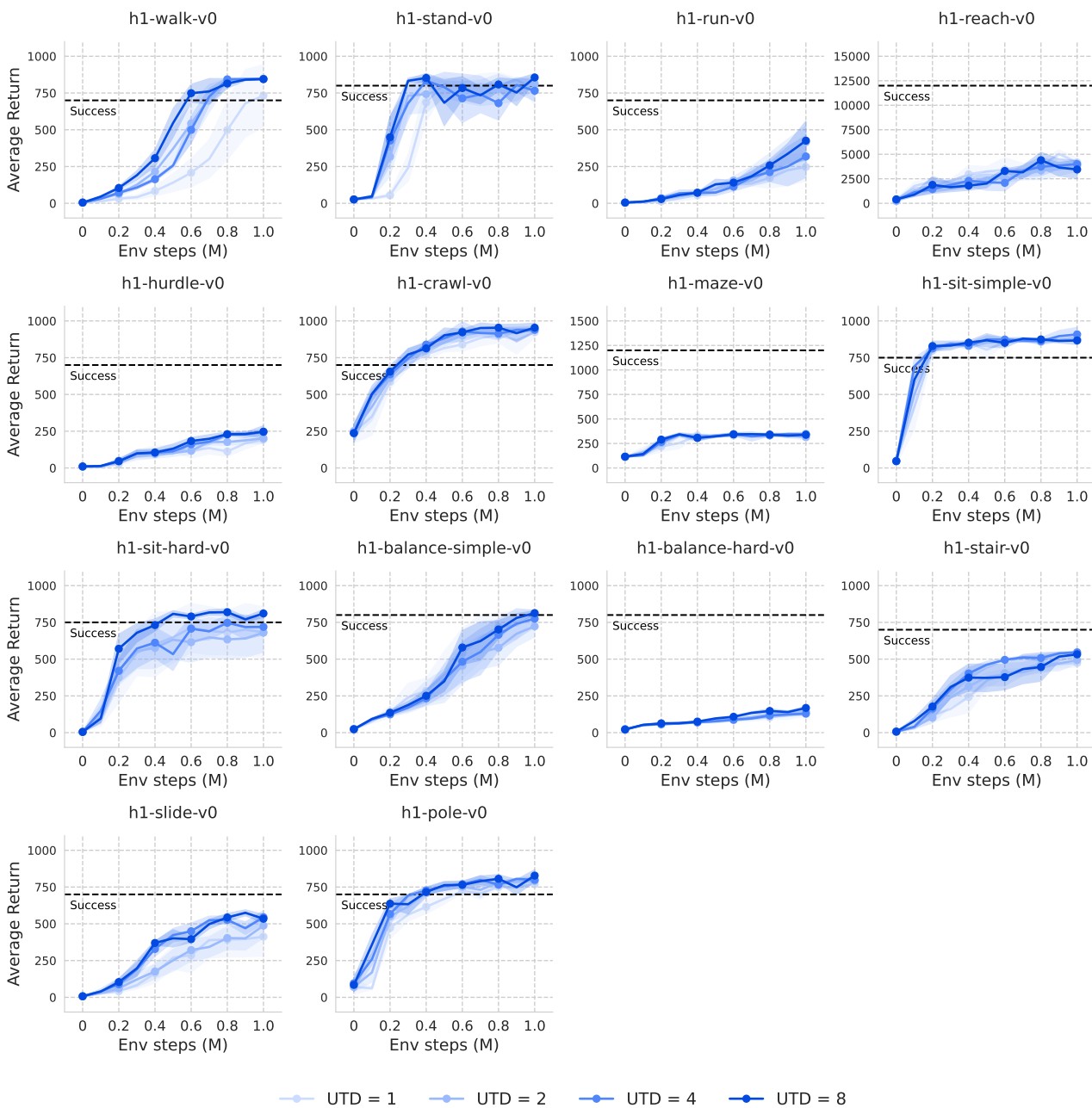

*Figure 16.* **Humanoidbench UTD Scaling Learning Curves.** Average episode return for the HumanoidBench environment. Results are averaged over 5 random seeds, and the shaded areas indicate 95% bootstrap confidence intervals. The black dotted line indicates the success score of each tasks (Appendix F.4)

# J. Complete Main Results

This section provides learning curves and final performance for each online RL task across the evaluated algorithms.

**Learning Curve.** For visibility of learning curve, we focus on DreamerV3 (Hafner et al., 2023), TD7 (Fujimoto et al., 2023), TD-MPC2 (Hansen et al., 2023), MR.Q (Fujimoto et al., 2025), and Simba (Lee et al., 2024c) as main baselines, selected for their strong performance and community adoption. We omit curves for algorithms with unavailable raw samples at each task.

**Confidence Interval.** The light-colored area in the figures and the gray-shaded, bracketed terms in the tables represent 95% bootstrap confidence intervals. For each task evaluated over $n$ random seeds, the 95% bootstrap confidence interval CI is computed as:

$$\text{CI} = \left[\mu - 1.96 \times \frac{\sigma}{\sqrt{n}}, \mu + 1.96 \times \frac{\sigma}{\sqrt{n}}\right]$$

where $\mu$ and $\sigma$ are the sample mean and standard deviation (with Bessel's correction) of the evaluation, respectively. For aggregated scores (mean, median, and interquartile mean), confidence intervals are computed over all $n \times T$ raw samples, where $n$ and $T$ are the number of evaluated random seeds and tasks in the benchmark, respectively. For algorithms with only average scores for each task available, we approximate the CI of aggregated scores using these averages (denoted with gray-colored †). We caution that this estimation may be inaccurate.

## J.1. Gym - MuJoCo

*Table 16.* **Gym - MuJoCo.** Final average performance at 1M environment steps for each of the 5 locomotion tasks in the Gym - MuJoCo benchmark. The number of evaluated random seeds for each algorithm is provided in Appendix E. The values in [brackets] represent a 95% bootstrap confidence interval. The aggregate mean, median and interquartile mean (IQM) are computed over the TD3-normalized score as described in Appendix F.1.

| Task | DreamerV3 | TD7 | TD-MPC2 | MR.Q | Simba | SimbaV2 |
|---|---|---|---|---|---|---|
| Ant-v4 | 1947 [1076, 2813] | 8509 [8168, 8844] | 4751 [2988, 6145] | 6989 [6203, 7617] | 5882 [5354, 6411] | 7429 [7209, 7649] |
| HalfCheetah-v4 | 5502 [3717, 7123] | 17433 [17301, 17559] | 15078 [14065, 15932] | 13305 [11841, 14140] | 9422 [8745, 10100] | 12022 [11640, 12404] |
| Hopper-v4 | 2666 [2106, 3210] | 3511 [3236, 3736] | 2081 [1197, 2921] | 2684 [2154, 3269] | 3231 [3004, 3458] | 4054 [3929, 4179] |
| Humanoid-v4 | 4217 [2785, 5523] | 7428 [7304, 7553] | 6071 [5770, 6333] | 7259 [5080, 9336] | 6513 [5634, 7392] | 10546 [10195, 10897] |
| Walker2d-v4 | 4519 [3692, 5244] | 6096 [5621, 6547] | 3008 [1706, 4321] | 6629 [5816, 7493] | 4290 [3864, 4716] | 6938 [6691, 7185] |
| IQM | 0.720 [0.620, 0.850] | 1.540 [1.500, 1.580] | 1.050 [0.890, 1.190] | 1.450 [1.270, 1.580] | 1.114 [1.043, 1.200] | 1.637 [1.470, 1.791] |
| Median | 0.810 [0.580, 0.930] | 1.550 [1.450, 1.630] | 1.180 [0.830, 1.220] | 1.420 [1.190, 1.710] | 1.143 [1.063, 1.227] | 1.616 [1.49, 1.744] |
| Mean | 0.760 [0.670, 0.860] | 1.570 [1.540, 1.600] | 1.040 [0.920, 1.150] | 1.390 [1.270, 1.490] | 1.147 [1.075, 1.223] | 1.617 [1.513, 1.718] |

| Task | PPO | SAC | TD3 | TD3+OFE | TQC | REDQ | DroQ | CrossQ | BRO |
|---|---|---|---|---|---|---|---|---|---|
| Ant-v4 | 1584 [1360, 1815] | 5733 [5316, 6151] | 3942 [2912, 4972] | 7398 [7280, 7516] | 3582 [2489, 4675] | 5314 [4539, 6090] | 5965 [5560, 6370] | 6980 [6834, 7126] | 7027 [6710, 7343] |
| HalfCheetah-v4 | 1744 [1523, 2118] | 11320 [10634, 12007] | 10574 [9677, 11471] | 13758 [13214, 14302] | 12349 [11471, 13227] | 11505 [10213, 12798] | 11070 [10272, 11867] | 12893 [11771, 14015] | 13747 [12621, 14873] |
| Hopper-v4 | 3022 [2633, 3339] | 2787 [2249, 3325] | 3226 [2911, 3541] | 3121 [2615, 3627] | 3526 [3302, 3750] | 3299 [2730, 3869] | 2797 [2387, 3208] | 2122 [1655, 2588] | 2122 [1655, 2588] |
| Humanoid-v4 | 477 [436, 518] | 4825 [3784, 5866] | 5165 [5020, 5310] | 6032 [5698, 6366] | 6029 [5498, 6560] | 5278 [5127, 5430] | 5380 [5353, 5407] | 10480 [10307, 10653] | 4757 [3139, 6376] |
| Walker2d-v4 | 2487 [1907, 3022] | 4536 [4229, 4843] | 3946 [3654, 4238] | 5195 [4683, 5707] | 5321 [4999, 5643] | 5228 [4836, 5620] | 4781 [4539, 5024] | 6257 [5277, 7237] | 3432 [2064, 4801] |
| IQM | 0.410 [0.110, 0.834]† | 1.097 [1.050, 1.155] | 1.000 [1.000, 1.000]† | 1.261 [1.035, 1.680]† | 1.143 [0.971, 1.290]† | 1.135 [1.086, 1.194] | 1.108 [1.055, 1.170] | 1.565 [1.394, 1.710] | 1.071 [0.828, 1.333] |
| Median | 0.412 [0.071, 0.936]† | 1.093 [1.028, 1.180] | 1.000 [1.000, 1.000]† | 1.293 [0.967, 1.861]† | 1.163 [0.910, 1.349]† | 1.188 [1.086, 1.241] | 1.133 [1.068, 1.199] | 1.489 [1.317, 1.643] | 1.071 [0.884, 1.322] |
| Mean | 0.447 [0.186, 0.725]† | 1.092 [1.013, 1.166] | 1.000 [1.000, 1.000]† | 1.322 [1.090, 1.615]† | 1.137 [1.012, 1.261]† | 1.160 [1.096, 1.224] | 1.134 [1.067, 1.205] | 1.475 [1.330, 1.608] | 1.101 [0.927, 1.278] |

*Figure 17.* **Gym-MuJoCo Learning Curves.** Average episode return (1k) for the Gym-MuJoCo environment. Results are averaged over random seeds of each algorithm, and the shaded areas indicate 95% bootstrap confidence intervals.

## J.2. Deepmind Control Suite - Easy

*Table 17.* **DMC Easy.** Final average performance at 1M environment steps for each of the 21 tasks of the DMC Easy benchmark. The number of evaluated random seeds for each algorithm is provided in Appendix E. The values in [brackets] represent a 95% bootstrap confidence interval. The aggregate mean, median and interquartile mean (IQM) are reported in units of 1k.

| Task | DreamerV3 | TD7 | TD-MPC2 | MR.Q | BRO | Simba | SimbaV2 |
|---|---|---|---|---|---|---|---|
| acrobot-swingup | 230 [193, 266] | 58 [38, 75] | 584 [551, 615] | 567 [523, 616] | 529 [504, 555] | 431 [379, 482] | 436 [391, 482] |
| ball-in-cup-catch | 968 [965, 973] | 984 [982, 986] | 983 [981, 985] | 981 [979, 984] | 982 [981, 984] | 981 [978, 983] | 982 [980, 984] |
| cartpole-balance | 998 [997, 1000] | 999 [998, 1000] | 996 [995, 998] | 999 [999, 1000] | 999 [998, 999] | 998 [998, 999] | 999 [999, 999] |
| cartpole-balance-sparse | 1000 [1000, 1000] | 999 [1000, 1000] | 1000 [1000, 1000] | 1000 [1000, 1000] | 852 [563, 1141] | 991 [973, 1008] | 967 [904, 1030] |
| cartpole-swingup | 736 [591, 838] | 869 [866, 873] | 875 [870, 880] | 866 [866, 866] | 879 [877, 882] | 876 [871, 881] | 880 [876, 883] |
| cartpole-swingup-sparse | 702 [560, 792] | 573 [333, 806] | 845 [839, 849] | 798 [780, 818] | 840 [827, 852] | 825 [795, 854] | 848 [848, 849] |
| cheetah-run | 917 [915, 920] | 699 [655, 744] | 914 [911, 917] | 877 [849, 905] | 863 [822, 904] | 920 [918, 922] | 821 [642, 913] |
| finger-spin | 666 [577, 763] | 335 [99, 596] | 986 [986, 988] | 937 [917, 956] | 988 [987, 989] | 849 [758, 939] | 891 [810, 972] |
| finger-turn-easy | 906 [883, 927] | 912 [774, 983] | 979 [975, 983] | 953 [931, 974] | 957 [923, 992] | 935 [903, 968] | 953 [925, 980] |
| finger-turn-hard | 864 [812, 900] | 470 [199, 727] | 947 [916, 977] | 950 [910, 974] | 957 [920, 993] | 915 [859, 972] | 951 [925, 977] |
| fish-swim | 813 [808, 819] | 86 [64, 120] | 659 [615, 706] | 792 [773, 810] | 618 [523, 713] | 823 [799, 846] | 826 [806, 846] |
| hopper-hop | 116 [66, 165] | 87 [25, 160] | 425 [368, 500] | 251 [195, 301] | 295 [273, 316] | 385 [322, 449] | 290 [233, 348] |
| hopper-stand | 747 [669, 806] | 670 [466, 829] | 952 [944, 958] | 951 [948, 955] | 949 [941, 957] | 929 [900, 957] | 944 [926, 962] |
| pendulum-swingup | 774 [740, 802] | 500 [251, 743] | 846 [830, 862] | 748 [597, 829] | 829 [795, 864] | 737 [575, 899] | 827 [805, 849] |
| quadruped-run | 130 [92, 169] | 645 [567, 713] | 942 [938, 947] | 947 [940, 954] | 859 [824, 895] | 928 [916, 939] | 935 [928, 943] |
| quadruped-walk | 193 [137, 243] | 949 [939, 957] | 963 [959, 967] | 963 [959, 967] | 958 [949, 967] | 957 [951, 963] | 962 [955, 969] |
| reacher-easy | 966 [964, 970] | 970 [951, 982] | 983 [980, 986] | 983 [983, 985] | 983 [983, 984] | 983 [981, 986] | 983 [979, 986] |
| reacher-hard | 919 [864, 955] | 898 [861, 936] | 960 [936, 979] | 977 [975, 980] | 974 [970, 978] | 966 [947, 984] | 967 [946, 987] |
| walker-run | 510 [430, 588] | 804 [783, 825] | 854 [851, 859] | 793 [765, 815] | 790 [776, 805] | 796 [792, 801] | 817 [812, 821] |
| walker-stand | 941 [934, 948] | 983 [974, 989] | 991 [990, 994] | 988 [987, 990] | 990 [986, 994] | 985 [982, 989] | 987 [984, 990] |
| walker-walk | 898 [875, 919] | 977 [975, 980] | 981 [979, 984] | 978 [978, 980] | 979 [975, 983] | 975 [972, 978] | 976 [974, 978] |
| IQM | 0.813 [0.621, 0.899]† | 0.771 [0.570, 0.907]† | 0.941 [0.880, 0.973]† | 0.927 [0.858, 0.966]† | 0.928 [0.899, 0.952] | 0.922 [0.905, 0.938] | 0.933 [0.918, 0.948] |
| Median | 0.813 [0.702, 0.917]† | 0.804 [0.573, 0.949]† | 0.952 [0.875, 0.981]† | 0.950 [0.866, 0.977]† | 0.872 [0.819, 0.906] | 0.870 [0.841, 0.896] | 0.875 [0.847, 0.905] |
| Mean | 0.714 [0.584, 0.832]† | 0.689 [0.548, 0.816]† | 0.889 [0.819, 0.946]† | 0.871 [0.788, 0.935]† | 0.861 [0.823, 0.896] | 0.864 [0.84, 0.887] | 0.874 [0.849, 0.897] |

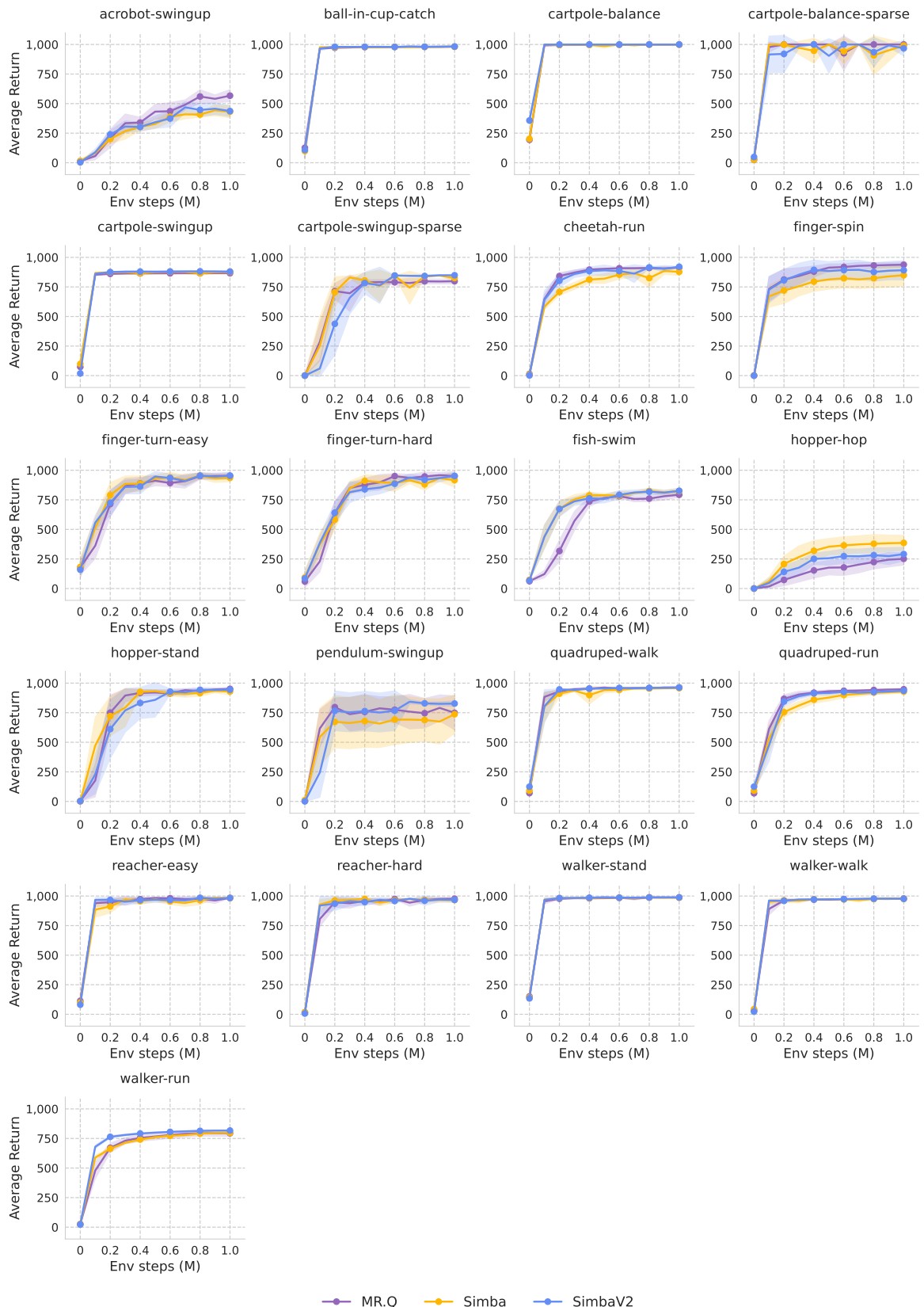

*Figure 18.* **DMC-Easy Learning Curves.** Average episode return for the DMC-Easy environment. Results are averaged over random seeds of each algorithm, and the shaded areas indicate 95% bootstrap confidence intervals.

## J.3. Deepmind Control Suite - Hard

*Table 18.* **DMC Hard.** Final average performance at 1M environment steps for each of the 7 tasks of the DMC Hard benchmark. The number of evaluated random seeds for each algorithm is provided in Appendix E. The values in [brackets] represent a 95% bootstrap confidence interval. The aggregate mean, median and interquartile mean (IQM) are reported in units of 1k.

| Task | DreamerV3 | TD7 | TD-MPC2 | MR.Q | Simba | SimbaV2 |
|------|-----------|-----|---------|------|-------|---------|
| dog-run | 4 [4, 5] | 69 [36, 101] | 265 [166, 342] | 569 [547, 595] | 544 [525, 564] | 562 [516, 608] |
| dog-stand | 22 [20, 27] | 582 [432, 741] | 506 [266, 715] | 967 [960, 975] | 960 [951, 969] | 981 [977, 985] |
| dog-trot | 10 [6, 17] | 21 [13, 30] | 407 [265, 530] | 877 [845, 898] | 824 [773, 876] | 861 [772, 950] |
| dog-walk | 17 [15, 21] | 52 [19, 116] | 486 [240, 704] | 916 [908, 924] | 916 [905, 928] | 935 [927, 944] |
| humanoid-run | 0 [1, 1] | 57 [23, 92] | 181 [121, 231] | 200 [170, 236] | 181 [171, 191] | 194 [182, 207] |
| humanoid-stand | 5 [5, 6] | 317 [117, 516] | 658 [506, 745] | 868 [822, 903] | 846 [801, 890] | 916 [886, 945] |
| humanoid-walk | 1 [1, 2] | 176 [42, 320] | 754 [725, 791] | 662 [610, 724] | 668 [608, 728] | 651 [590, 713] |
| IQM | 0.008 [0.002, 0.016]† | 0.134 [0.047, 0.343]† | 0.464 [0.305, 0.632]† | 0.778 [0.500, 0.911]† | 0.773 [0.713, 0.83] | 0.808 [0.726, 0.879] |
| Median | 0.005 [0.001, 0.018]† | 0.069 [0.052, 0.317]† | 0.486 [0.265, 0.658]† | 0.868 [0.569, 0.916]† | 0.706 [0.647, 0.772] | 0.729 [0.655, 0.808] |
| Mean | 0.009 [0.003, 0.015]† | 0.182 [0.062, 0.336]† | 0.465 [0.329, 0.606]† | 0.723 [0.516, 0.886]† | 0.706 [0.656, 0.755] | 0.729 [0.664, 0.791] |

| Task | PPO | SAC | iQRL | BRO | MAD-TD |
|------|-----|-----|------|-----|--------|
| dog-run | 26 [26, 28] | 36 [3, 69] | 380 [336, 424] | 374 [338, 411] | 437 [396, 478] |
| dog-stand | 129 [122, 139] | 102 [39, 164] | 926 [897, 955] | 966 [956, 977] | 967 [952, 982] |
| dog-trot | 31 [30, 34] | 29 [6, 52] | 713 [516, 909] | 783 [717, 848] | 867 [805, 929] |
| dog-walk | 40 [37, 43] | 38 [11, 65] | 866 [827, 905] | 931 [920, 942] | 924 [906, 943] |
| humanoid-run | 0 [1, 1] | 116 [89, 144] | 188 [167, 210] | 204 [186, 223] | 200 [180, 220] |
| humanoid-stand | 5 [5, 6] | 352 [225, 479] | 727 [655, 799] | 920 [909, 931] | 870 [840, 901] |
| humanoid-walk | 1 [1, 2] | 273 [128, 418] | 688 [642, 735] | 672 [619, 725] | 684 [609, 759] |
| IQM | 0.021 [0.003, 0.069]† | 0.069 [0.042, 0.114] | 0.694 [0.528, 0.805] | 0.772 [0.662, 0.854] | 0.787 [0.691, 0.865] |
| Median | 0.026 [0.001, 0.040]† | 0.159 [0.08, 0.183] | 0.64 [0.516, 0.766] | 0.694 [0.615, 0.774] | 0.707 [0.634, 0.786] |
| Mean | 0.033 [0.009, 0.068]† | 0.136 [0.098, 0.175] | 0.642 [0.531, 0.747] | 0.693 [0.625, 0.757] | 0.708 [0.642, 0.771] |

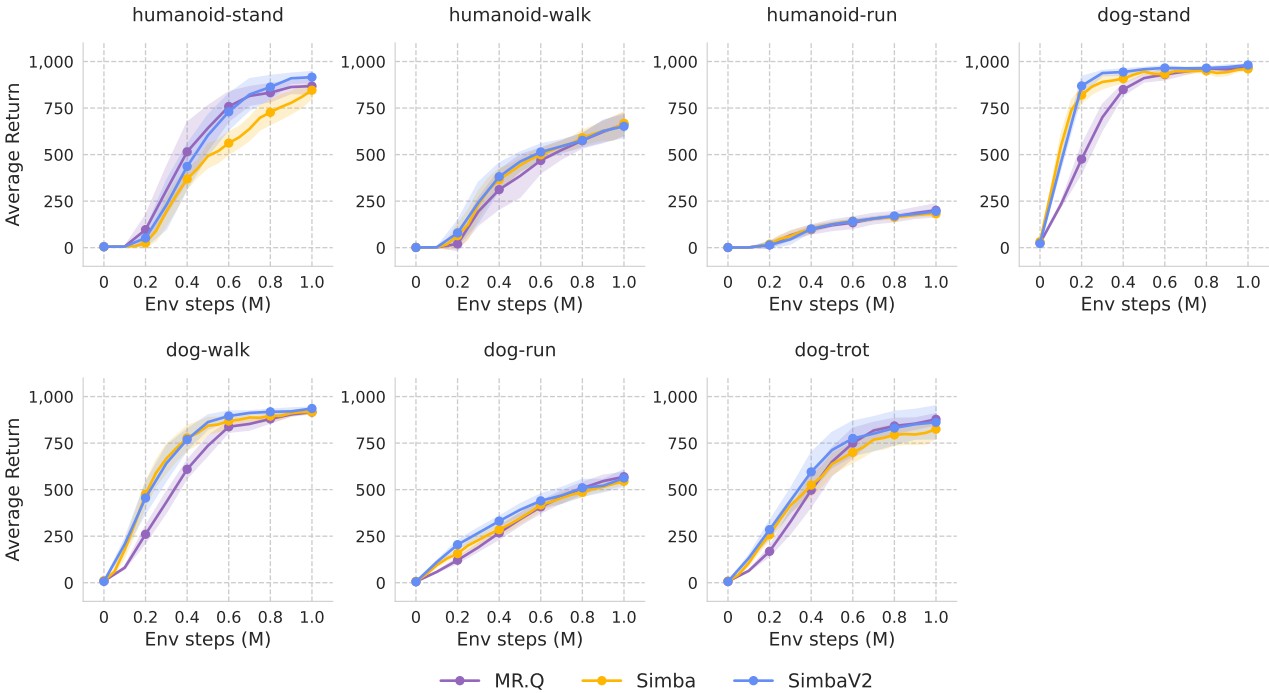

*Figure 19.* **DMC-Hard Learning Curves.** Average episode return for the DMC-Hard environment. Results are averaged over random seeds of each algorithm, and the shaded areas indicate 95% bootstrap confidence intervals.

## J.4. MyoSuite

*Table 19.* **MyoSuite.** Final average performance at 1M environment steps across each of the 10 continuous control tasks in the MyoSuite benchmark, including both fixed-goal and randomized-goal (`hard`) settings. The number of evaluated random seeds for each algorithm is provided in Appendix E. The values in [brackets] represent a 95% bootstrap confidence interval. Performance is measured by the average success rate of each task.

| Task | DreamerV3 | TD7 | TD-MPC2 | Simba | SimbaV2 |
|---|---|---|---|---|---|
| `myo-pen-twirl-hard` | 53.3 [29.8, 76.9] | 12.0 [2.4, 21.6] | 40.0 [40.0, 40.0] | 77.0 [66.4, 87.6] | 93.0 [88.8, 97.2] |
| `myo-pen-twirl` | 96.7 [90.1, 103.2] | 100.0 [100.0, 100.0] | 70.0 [11.2, 128.8] | 80.0 [53.9, 106.1] | 100.0 [100.0, 100.0] |
| `myo-key-turn-hard` | 0.0 [0.0, 0.0] | 0.0 [0.0, 0.0] | 0.0 [0.0, 0.0] | 7.0 [-3.1, 17.1] | 62.0 [42.7, 81.3] |
| `myo-key-turn` | 88.9 [67.1, 110.7] | 100.0 [100.0, 100.0] | 100.0 [100.0, 100.0] | 100.0 [100.0, 100.0] | 100.0 [100.0, 100.0] |
| `myo-obj-hold-hard` | 9.4 [8.4, 10.5] | 10.0 [-0.7, 20.7] | 56.7 [39.4, 74.0] | 96.0 [92.8, 99.2] | 98.0 [95.4, 100.6] |
| `myo-obj-hold` | 33.3 [-32.0, 98.7] | 20.0 [-19.2, 59.2] | 100.0 [100.0, 100.0] | 90.0 [70.4, 109.6] | 100.0 [100.0, 100.0] |
| `myo-pose-hard` | 0.0 [0.0, 0.0] | 0.0 [0.0, 0.0] | 0.0 [0.0, 0.0] | 0.0 [0.0, 0.0] | 0.0 [0.0, 0.0] |
| `myo-pose` | 100.0 [100.0, 100.0] | 0.0 [0.0, 0.0] | 100.0 [100.0, 100.0] | 100.0 [100.0, 100.0] | 100.0 [100.0, 100.0] |
| `myo-reach-hard` | 0.0 [0.0, 0.0] | 14.0 [0.7, 27.3] | 83.3 [66.0, 100.6] | 93.0 [86.4, 99.6] | 94.0 [87.3, 100.7] |
| `myo-reach` | 100.0 [100.0, 100.0] | 100.0 [100.0, 100.0] | 100.0 [100.0, 100.0] | 100.0 [100.0, 100.0] | 100.0 [100.0, 100.0] |
| IQM | 46.6 [16.7, 76.9] | 22.3 [4.2, 46.2] | 77.5 [50.6, 94.4] | 95.2 [82.6, 98.8] | 99.0 [96.8, 100.0] |
| Median | 47.0 [29.0, 67.0] | 34.0 [21.0, 51.0] | 65.0 [47.0, 82.0] | 77.0 [66.5, 85.5] | 84.5 [78.0, 93.0] |
| Mean | 48.2 [31.8, 64.6] | 35.6 [23.8, 48.0] | 65.0 [50.3, 78.7] | 74.3 [66.3, 81.7] | 84.7 [78.2, 90.6] |

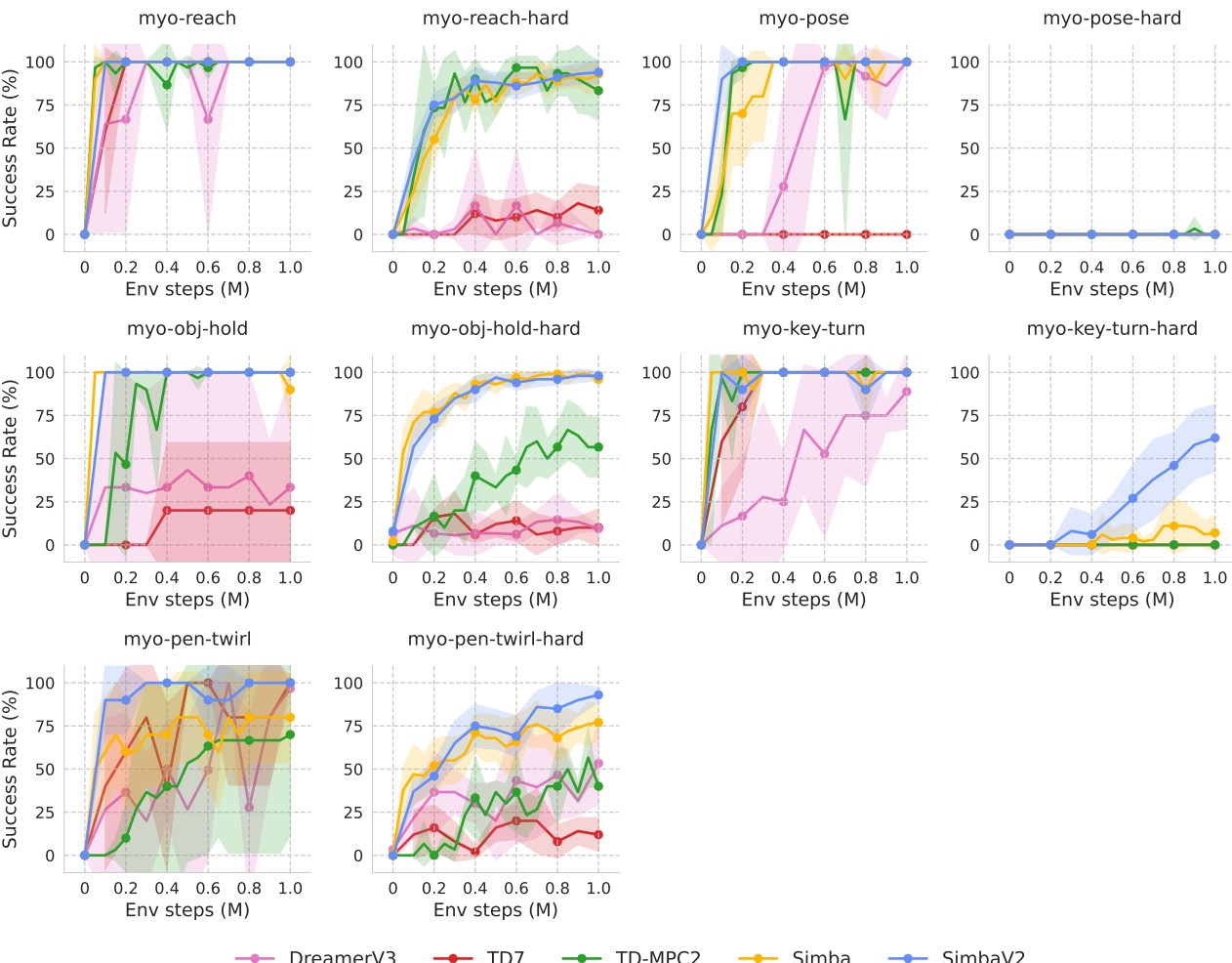

*Figure 20.* **MyoSuite Learning Curves.** Average episode success rate (%) for the MyoSuite environment. Results are averaged over random seeds of each algorithm, and the shaded areas indicate 95% bootstrap confidence intervals.

## J.5. Humanoid Bench

*Table 20.* **HumanoidBench.** Final average performance at 1M environment steps for each of the 14 locomotion tasks in the Humanoid-Bench benchmark. The number of evaluated random seeds for each algorithm is provided in Appendix E. The values in [brackets] represent a 95% bootstrap confidence interval. The aggregate mean, median and interquartile mean (IQM) are computed over the success normalized score as described in Appendix F.4.

| Task | DreamerV3 | TD7 | TD-MPC2 | Simba | SimbaV2 |
|---|---|---|---|---|---|
| h1-pole-v0 | 41 [28, 54] | 441 [320, 563] | 744 [609, 879] | 716 [667, 765] | 791 [785, 797] |
| h1-slide-v0 | 11 [7, 15] | 39 [26, 53] | 334 [304, 364] | 277 [252, 303] | 487 [404, 571] |
| h1-stair-v0 | 7 [2, 12] | 52 [31, 74] | 378 [108, 648] | 269 [153, 385] | 493 [467, 518] |
| h1-balance-hard-v0 | 11 [7, 15] | 79 [51, 107] | 31 [5, 56] | 75 [71, 80] | 143 [128, 157] |
| h1-balance-simple-v0 | 9 [6, 12] | 69 [58, 80] | 42 [14, 70] | 337 [193, 482] | 723 [651, 795] |
| h1-sit-hard-v0 | 15 [-4, 35] | 235 [154, 315] | 723 [660, 786] | 512 [354, 670] | 679 [548, 811] |
| h1-sit-simple-v0 | 19 [9, 28] | 874 [869, 879] | 790 [772, 809] | 833 [814, 853] | 875 [870, 880] |
| h1-maze-v0 | 113 [107, 118] | 147 [137, 156] | 244 [106, 383] | 354 [342, 366] | 313 [287, 340] |
| h1-crawl-v0 | 248 [176, 319] | 582 [563, 600] | 962 [959, 965] | 923 [904, 942] | 946 [933, 959] |
| h1-hurdle-v0 | 4 [3, 5] | 60 [18, 102] | 387 [254, 519] | 175 [150, 201] | 202 [167, 236] |
| h1-reach-v0 | 3203 [2824, 3581] | 1409 [998, 1821] | 2654 [1951, 3357] | 3874 [3220, 4527] | 3850 [3272, 4427] |
| h1-run-v0 | 4 [2, 6] | 91 [54, 128] | 778 [763, 793] | 232 [185, 279] | 415 [307, 524] |
| h1-stand-v0 | 15 [7, 22] | 433 [138, 727] | 798 [779, 817] | 772 [701, 843] | 814 [770, 857] |
| h1-walk-v0 | 8 [1, 16] | 33 [22, 45] | 814 [813, 815] | 550 [391, 709] | 845 [840, 850] |
| IQM | 0.007 [0.004, 0.012] | 0.134 [0.088, 0.245] | 0.734 [0.510, 0.936] | 0.521 [0.413, 0.633] | 0.799 [0.686, 0.908] |
| Median | 0.021 [0.000, 0.047] | 0.284 [0.183, 0.392] | 0.696 [0.536, 0.881] | 0.598 [0.514, 0.692] | 0.781 [0.693, 0.865] |
| Mean | 0.022 [0.000, 0.046] | 0.289 [0.207, 0.375] | 0.710 [0.562, 0.858] | 0.606 [0.536, 0.678] | 0.776 [0.705, 0.849] |

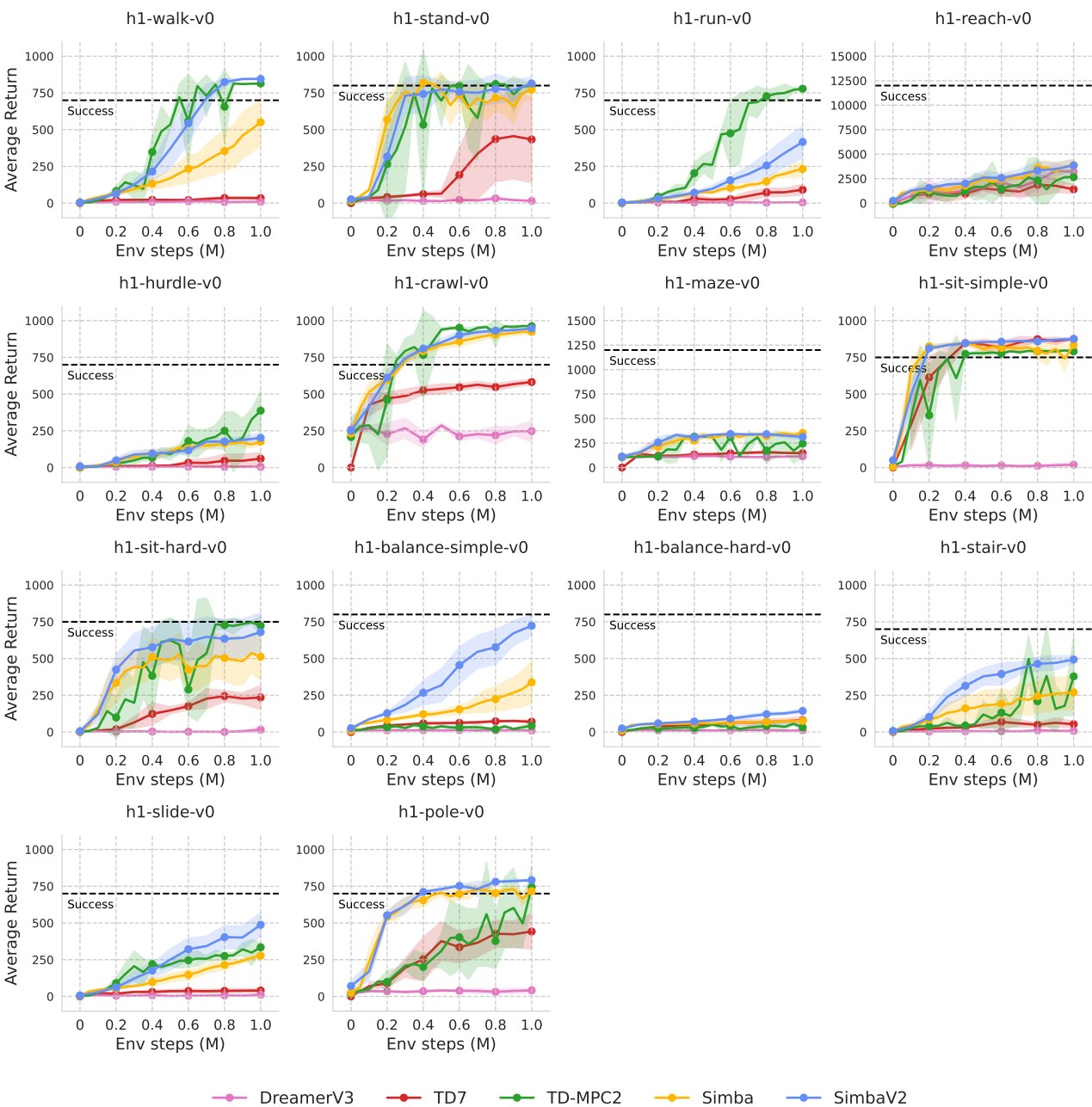

*Figure 21.* **Humanoidbench Learning Curves.** Average episode return for the HumanoidBench environment. Results are averaged over random seeds of each algorithm, and the shaded areas indicate 95% bootstrap confidence intervals. The black dotted line indicates the success score of each tasks (Appendix F.4)

# K. Complete Ablation Results

This section presents a per-environment analysis of the design variations discussed in Section 5.5. Each table includes raw scores for individual environments, with [bracketed values] indicating 95% bootstrap confidence intervals. The aggregate mean, median, and interquartile mean (IQM) are calculated based on the differences in normalized scores. To illustrate the magnitude of these differences, we use the following highlight scale:

- $(\geq 0.1)$
- $[0.05, 0.1)$
- $[0.02, 0.05)$
- $[-0.02, -0.05)$
- $[-0.05, -0.1)$
- $(\leq -0.05)$

## K.1. Gym - MuJoCo

*Table 21.* **Mujoco (Input Design).** Final average performance at 1M environment steps averaged over 3 seeds. The [bracketed values] represent a 95% bootstrap confidence interval. The aggregate mean, median and interquartile mean (IQM) are computed over the TD3-normalized score as described in Appendix F.1.

| Task | SimbaV2 | No L2 Normalize | No Shifting | $c_{shift} : 1$ | Resize Projection |
|---|---|---|---|---|---|
| Ant-v4 | 7429 [7209, 7649] | 7267 [7065, 7469] | 7203 [6765, 7641] | 7367 [7302, 7433] | 7834 [7374, 8294] |
| HalfCheetah-v4 | 12022 [11640, 12404] | 5386 [4901, 5870] | 5464 [5178, 5750] | 11913 [11241, 12584] | 12047 [11315, 12778] |
| Hopper-v4 | 4053 [3928, 4178] | 3764 [3550, 3978] | 3676 [3619, 3734] | 3703 [3217, 4189] | 3788 [3602, 3974] |
| Humanoid-v4 | 10545 [10195, 10896] | 9820 [8982, 10658] | 10655 [10050, 11259] | 10680 [10586, 10775] | 10530 [10308, 10753] |
| Walker2d-v4 | 6938 [6691, 7185] | 5560 [4757, 6364] | 5750 [5368, 6131] | 6192 [5986, 6399] | 6985 [6378, 7591] |
| IQM | 1.637 [1.471, 1.792] | 1.448 [1.102, 1.715] | 1.463 [1.115, 1.746] | 1.552 [1.304, 1.809] | 1.645 [1.34, 1.919] |
| Median | 1.616 [1.495, 1.746] | 1.364 [1.122, 1.618] | 1.411 [1.14, 1.67] | 1.573 [1.372, 1.749] | 1.623 [1.434, 1.813] |
| Mean | 1.617 [1.514, 1.72] | 1.37 [1.137, 1.591] | 1.406 [1.164, 1.644] | 1.558 [1.39, 1.728] | 1.623 [1.45, 1.796] |

*Table 22.* **Mujoco (Output Design).** Final performance at 1M environment steps averaged over 3 seeds. The [bracketed values] represent a 95% bootstrap confidence interval. The aggregate mean, median and interquartile mean (IQM) are computed over the TD3-normalized score as described in Appendix F.1.

| Task | SimbaV2 | MSE Loss | No Reward Scaling | No Return Bounding | Hard Target |
|---|---|---|---|---|---|
| Ant-v4 | 7429 [7209, 7649] | 6195 [5459, 6931] | 7087 [6935, 7239] | 7622 [7486, 7757] | 7373 [7342, 7405] |
| HalfCheetah-v4 | 12022 [11640, 12404] | 12222 [11753, 12691] | 12775 [12608, 12941] | 12724 [12133, 13315] | 11986 [11434, 12538] |
| Hopper-v4 | 4053 [3928, 4178] | 3507 [3333, 3682] | 2932 [2007, 3858] | 4113 [3999, 4228] | 3623 [3445, 3800] |
| Humanoid-v4 | 10545 [10195, 10896] | 7764 [7227, 8302] | 8265 [5867, 10664] | 10583 [10506, 10660] | 9973 [9763, 10184] |
| Walker2d-v4 | 6938 [6691, 7185] | 5267 [4667, 5866] | 5786 [5116, 6456] | 6442 [6112, 6772] | 7428 [6463, 8393] |
| IQM | 1.637 [1.474, 1.792] | 1.334 [1.162, 1.5] | 1.405 [1.236, 1.597] | 1.612 [1.367, 1.864] | 1.624 [1.32, 1.874] |
| Median | 1.616 [1.495, 1.745] | 1.341 [1.202, 1.483] | 1.426 [1.201, 1.609] | 1.616 [1.453, 1.79] | 1.593 [1.399, 1.771] |
| Mean | 1.617 [1.516, 1.719] | 1.343 [1.225, 1.462] | 1.395 [1.247, 1.544] | 1.62 [1.471, 1.773] | 1.589 [1.414, 1.757] |

*Table 23.* **Mujoco (Training Design).** Final performance at 1M environment steps averaged over 3 seeds. The [bracketed values] represent a 95% bootstrap confidence interval. The aggregate mean, median and interquartile mean (IQM) are computed over the TD3-normalized score as described in Appendix F.1.

| Task | SimbaV2 | No LR Decay | $s_{init} : 1$ | $s_{scale} : 1$ | $\alpha_{init} : 0.5$ | $\alpha_{scale} : 1$ |
|---|---|---|---|---|---|---|
| Ant-v4 | 7429 [7209, 7649] | 7553 [6914, 8192] | 7429 [7237, 7621] | 7296 [7146, 7447] | 7552 [7323, 7781] | 7258 [6959, 7556] |
| HalfCheetah-v4 | 12022 [11640, 12404] | 12227 [11760, 12694] | 12090 [11758, 12422] | 11548 [10470, 12626] | 12538 [12472, 12604] | 11982 [11585, 12378] |
| Hopper-v4 | 4053 [3928, 4178] | 3635 [3000, 4269] | 4046 [3944, 4148] | 4008 [3869, 4146] | 3568 [3279, 3857] | 4023 [3830, 4216] |
| Humanoid-v4 | 10545 [10195, 10896] | 9907 [8412, 11401] | 9819 [8615, 11023] | 10636 [10465, 10807] | 10828 [10426, 11229] | 8624 [6418, 10831] |
| Walker2d-v4 | 6938 [6691, 7185] | 6661 [6010, 7311] | 6583 [5893, 7274] | 6770 [6541, 6998] | 6328 [5726, 6929] | 6744 [6476, 7013] |
| IQM | 1.637 [1.478, 1.789] | 1.556 [1.31, 1.81] | 1.588 [1.418, 1.749] | 1.613 [1.456, 1.767] | 1.563 [1.29, 1.848] | 1.53 [1.296, 1.746] |
| Median | 1.616 [1.494, 1.743] | 1.541 [1.374, 1.745] | 1.605 [1.451, 1.709] | 1.606 [1.467, 1.721] | 1.584 [1.388, 1.78] | 1.546 [1.362, 1.681] |
| Mean | 1.617 [1.518, 1.718] | 1.562 [1.393, 1.73] | 1.571 [1.467, 1.675] | 1.594 [1.488, 1.699] | 1.583 [1.405, 1.757] | 1.52 [1.377, 1.659] |

## K.2. Deepmind Control Suite - Easy

*Table 24.* **DMC-Easy (Input Design).** Final performance at 1M environment steps averaged over 3 seeds. The [bracketed values] represent a 95% bootstrap confidence interval. The aggregate mean, median and interquartile mean are computed over the default reward.

| Task | SimbaV2 | No L2 Normalize | No Shifting | $c_{shift}:1$ | Resize Projection |
|---|---|---|---|---|---|
| acrobot-swingup | 436 [391, 482] | 385 [364, 406] | 399 [276, 522] | 466 [402, 530] | 293 [184, 401] |
| ball-in-cup-catch | 982 [980, 984] | 564 [92, 1036] | 586 [145, 1026] | 983 [979, 987] | 983 [979, 986] |
| cartpole-balance | 999 [999, 999] | 999 [999, 999] | 999 [999, 999] | 999 [999, 999] | 999 [999, 999] |
| cartpole-balance-sparse | 967 [904, 1030] | 1000 [1000, 1000] | 992 [978, 1007] | 1000 [1000, 1000] | 1000 [1000, 1000] |
| cartpole-swingup | 880 [876, 883] | 881 [881, 882] | 881 [880, 882] | 882 [881, 882] | 799 [646, 951] |
| cartpole-swingup-sparse | 848 [848, 849] | 838 [829, 846] | 829 [796, 862] | 846 [843, 848] | 803 [719, 888] |
| cheetah-run | 920 [918, 922] | 499 [426, 572] | 519 [477, 560] | 919 [914, 924] | 917 [914, 920] |
| finger-spin | 891 [810, 972] | 699 [485, 913] | 620 [404, 836] | 855 [674, 1036] | 883 [737, 1030] |
| finger-turn-easy | 953 [925, 980] | 922 [858, 986] | 924 [831, 1017] | 922 [874, 970] | 925 [866, 985] |
| finger-turn-hard | 951 [925, 977] | 888 [796, 980] | 937 [895, 979] | 870 [861, 879] | 917 [864, 970] |
| fish-swim | 826 [806, 846] | 450 [316, 584] | 442 [326, 558] | 806 [783, 830] | 758 [706, 811] |
| hopper-hop | 290 [233, 348] | 347 [270, 424] | 208 [132, 284] | 212 [79, 345] | 350 [191, 509] |
| hopper-stand | 944 [926, 962] | 804 [578, 1031] | 680 [444, 915] | 925 [883, 967] | 753 [440, 1066] |
| pendulum-swingup | 827 [805, 849] | 610 [208, 1011] | 620 [213, 1026] | 810 [762, 859] | 678 [453, 902] |
| quadruped-run | 935 [928, 943] | 937 [923, 950] | 917 [902, 932] | 900 [850, 949] | 935 [927, 943] |
| quadruped-walk | 962 [955, 969] | 962 [949, 975] | 963 [958, 967] | 960 [945, 974] | 957 [947, 967] |
| reacher-easy | 983 [979, 986] | 949 [905, 992] | 970 [952, 989] | 984 [982, 985] | 982 [980, 985] |
| reacher-hard | 967 [946, 987] | 881 [707, 1054] | 935 [890, 979] | 973 [966, 979] | 975 [972, 978] |
| walker-run | 817 [812, 821] | 762 [672, 853] | 793 [778, 809] | 816 [811, 820] | 813 [809, 816] |
| walker-stand | 987 [984, 990] | 987 [984, 990] | 987 [983, 991] | 986 [978, 995] | 979 [967, 991] |
| walker-walk | 976 [974, 978] | 974 [968, 981] | 978 [975, 981] | 976 [973, 980] | 969 [963, 975] |
| IQM | 0.933 [0.918, 0.948] | 0.874 [0.795, 0.922] | 0.871 [0.789, 0.921] | 0.919 [0.891, 0.942] | 0.92 [0.888, 0.946] |
| Median | 0.875 [0.847, 0.905] | 0.787 [0.717, 0.839] | 0.781 [0.712, 0.837] | 0.867 [0.818, 0.905] | 0.844 [0.792, 0.892] |
| Mean | 0.874 [0.849, 0.898] | 0.779 [0.722, 0.832] | 0.771 [0.713, 0.826] | 0.862 [0.819, 0.9] | 0.842 [0.794, 0.885] |

*Table 25.* **DMC-Easy (Output Design).** Final performance at 1M environment steps averaged over 3 seeds. The [bracketed values] represent a 95% bootstrap confidence interval. The aggregate mean, median and interquartile mean are computed over the default reward.

| Task | SimbaV2 | MSE Loss | No Reward Scaling | No Return Bounding | Hard Target |
|---|---|---|---|---|---|
| acrobot-swingup | 436 [391, 482] | 384 [273, 494] | 383 [329, 438] | 439 [386, 492] | 423 [350, 496] |
| ball-in-cup-catch | 982 [980, 984] | 982 [979, 985] | 981 [978, 984] | 983 [979, 986] | 983 [979, 986] |
| cartpole-balance | 999 [999, 999] | 999 [998, 1000] | 999 [999, 999] | 694 [659, 730] | 999 [999, 999] |
| cartpole-balance-sparse | 967 [904, 1030] | 1000 [1000, 1000] | 970 [913, 1027] | 998 [994, 1001] | 1000 [1000, 1000] |
| cartpole-swingup | 880 [876, 883] | 881 [880, 882] | 881 [880, 881] | 758 [737, 779] | 881 [880, 882] |
| cartpole-swingup-sparse | 848 [848, 849] | 845 [843, 847] | 715 [493, 937] | 844 [839, 849] | 846 [844, 848] |
| cheetah-run | 920 [918, 922] | 796 [563, 1030] | 869 [813, 925] | 887 [829, 946] | 905 [873, 937] |
| finger-spin | 891 [810, 972] | 959 [937, 980] | 824 [684, 963] | 774 [632, 916] | 954 [919, 989] |
| finger-turn-easy | 953 [925, 980] | 973 [965, 981] | 916 [873, 959] | 970 [964, 976] | 970 [964, 977] |
| finger-turn-hard | 951 [925, 977] | 886 [793, 978] | 853 [761, 945] | 917 [861, 972] | 966 [958, 973] |
| fish-swim | 826 [806, 846] | 838 [827, 849] | 844 [834, 853] | 819 [790, 847] | 809 [792, 826] |
| hopper-hop | 290 [233, 348] | 380 [198, 561] | 294 [237, 350] | 211 [114, 309] | 316 [293, 339] |
| hopper-stand | 944 [926, 962] | 920 [858, 982] | 928 [897, 959] | 710 [464, 956] | 918 [862, 973] |
| pendulum-swingup | 827 [805, 849] | 773 [754, 792] | 814 [776, 852] | 809 [763, 854] | 809 [760, 858] |
| quadruped-run | 935 [928, 943] | 938 [914, 962] | 929 [901, 956] | 923 [903, 944] | 943 [930, 956] |
| quadruped-walk | 962 [955, 969] | 955 [943, 967] | 968 [960, 976] | 959 [949, 968] | 964 [953, 974] |
| reacher-easy | 983 [979, 986] | 968 [944, 992] | 966 [938, 994] | 983 [981, 985] | 983 [982, 985] |
| reacher-hard | 967 [946, 987] | 978 [977, 979] | 976 [972, 981] | 976 [972, 981] | 969 [949, 989] |
| walker-run | 817 [812, 821] | 795 [793, 797] | 818 [815, 820] | 732 [687, 777] | 817 [812, 821] |
| walker-stand | 987 [984, 990] | 992 [990, 993] | 987 [983, 991] | 945 [912, 978] | 986 [979, 994] |
| walker-walk | 976 [974, 978] | 977 [974, 980] | 974 [970, 978] | 972 [966, 979] | 978 [976, 981] |
| IQM | 0.933 [0.918, 0.948] | 0.928 [0.894, 0.954] | 0.914 [0.89, 0.937] | 0.887 [0.843, 0.922] | 0.938 [0.911, 0.96] |
| Median | 0.875 [0.846, 0.905] | 0.866 [0.813, 0.918] | 0.853 [0.808, 0.896] | 0.823 [0.777, 0.871] | 0.878 [0.836, 0.918] |
| Mean | 0.874 [0.848, 0.898] | 0.868 [0.821, 0.909] | 0.852 [0.814, 0.888] | 0.824 [0.78, 0.865] | 0.878 [0.838, 0.913] |

*Table 26.* **DMC-Easy (Training Design).** Final performance at 1M environment steps averaged over 3 seeds. The [bracketed values] represent a 95% bootstrap confidence interval. The aggregate mean, median and interquartile mean are computed over the default reward.

| Task | SimbaV2 | No LR Decay | $s_{init} : 1$ | $s_{scale} : 1$ | $\alpha_{init} : 0.5$ | $\alpha_{scale} : 1$ |
|---|---|---|---|---|---|---|
| acrobot-swingup | 436 [391, 482] | 393 [269, 517] | 490 [453, 527] | 448 [408, 488] | 452 [359, 545] | 449 [391, 506] |
| ball-in-cup-catch | 982 [980, 984] | 982 [978, 986] | 982 [980, 984] | 982 [980, 984] | 983 [979, 987] | 983 [979, 987] |
| cartpole-balance | 999 [999, 999] | 999 [999, 999] | 999 [999, 999] | 999 [999, 999] | 999 [999, 999] | 999 [999, 999] |
| cartpole-balance-sparse | 967 [904, 1030] | 1000 [1000, 1000] | 999 [997, 1000] | 1000 [1000, 1000] | 1000 [1000, 1000] | 1000 [1000, 1000] |
| cartpole-swingup | 880 [876, 883] | 880 [878, 882] | 881 [881, 882] | 882 [881, 882] | 881 [881, 882] | 881 [881, 882] |
| cartpole-swingup-sparse | 848 [848, 849] | 699 [409, 989] | 847 [846, 848] | 787 [697, 878] | 848 [847, 849] | 847 [845, 849] |
| cheetah-run | 920 [918, 922] | 917 [913, 922] | 917 [913, 921] | 914 [903, 925] | 918 [914, 921] | 866 [758, 974] |
| finger-spin | 891 [810, 972] | 871 [716, 1025] | 957 [941, 973] | 895 [813, 977] | 950 [921, 979] | 928 [852, 1004] |
| finger-turn-easy | 953 [925, 980] | 895 [846, 943] | 945 [904, 986] | 955 [931, 979] | 864 [794, 934] | 878 [800, 956] |
| finger-turn-hard | 951 [925, 977] | 869 [755, 982] | 941 [910, 971] | 959 [937, 981] | 939 [895, 983] | 943 [898, 988] |
| fish-swim | 826 [806, 846] | 805 [766, 844] | 819 [804, 834] | 821 [792, 850] | 806 [759, 853] | 825 [814, 835] |
| hopper-hop | 290 [233, 348] | 321 [305, 338] | 282 [217, 347] | 304 [265, 343] | 171 [59, 283] | 313 [293, 333] |
| hopper-stand | 944 [926, 962] | 922 [882, 963] | 840 [705, 976] | 879 [730, 1027] | 929 [882, 976] | 622 [240, 1004] |
| pendulum-swingup | 827 [805, 849] | 810 [763, 858] | 827 [806, 848] | 820 [797, 843] | 811 [766, 857] | 812 [767, 857] |
| quadruped-run | 935 [928, 943] | 940 [923, 958] | 929 [921, 937] | 929 [910, 948] | 931 [911, 952] | 930 [922, 938] |
| quadruped-walk | 962 [955, 969] | 967 [962, 973] | 966 [962, 971] | 954 [939, 969] | 953 [946, 961] | 954 [948, 959] |
| reacher-easy | 983 [979, 986] | 982 [979, 985] | 983 [980, 985] | 983 [981, 986] | 983 [980, 985] | 984 [981, 986] |
| reacher-hard | 967 [946, 987] | 977 [971, 982] | 949 [920, 977] | 958 [933, 983] | 973 [968, 979] | 970 [958, 982] |
| walker-run | 817 [812, 821] | 816 [810, 822] | 814 [810, 818] | 819 [817, 821] | 819 [817, 822] | 819 [817, 821] |
| walker-stand | 987 [984, 990] | 987 [986, 989] | 990 [988, 991] | 988 [987, 990] | 989 [985, 992] | 990 [988, 993] |
| walker-walk | 976 [974, 978] | 976 [969, 984] | 975 [972, 979] | 977 [973, 981] | 975 [974, 976] | 977 [974, 981] |
| IQM | 0.933 [0.918, 0.948] | 0.923 [0.894, 0.947] | 0.932 [0.916, 0.947] | 0.934 [0.918, 0.949] | 0.927 [0.901, 0.949] | 0.922 [0.892, 0.947] |
| Median | 0.875 [0.847, 0.904] | 0.858 [0.811, 0.902] | 0.878 [0.847, 0.906] | 0.871 [0.842, 0.902] | 0.863 [0.818, 0.911] | 0.859 [0.809, 0.902] |
| Mean | 0.874 [0.849, 0.897] | 0.858 [0.813, 0.896] | 0.873 [0.848, 0.897] | 0.87 [0.843, 0.894] | 0.866 [0.819, 0.905] | 0.856 [0.812, 0.895] |

## K.3. Deepmind Control Suite - Hard

*Table 27.* **DMC-Hard (Input Design).** Final performance at 1M environment steps averaged over 3 seeds. The [bracketed values] represent a 95% bootstrap confidence interval. The aggregate mean, median and interquartile mean are computed over the default reward.

| Task | SimbaV2 | No L2 Normalize | No Shifting | $c_{shift}$ : 1 | Resize Projection |
|---|---|---|---|---|---|
| dog-run | 562 [516, 608] | 573 [515, 632] | 530 [415, 644] | 610 [575, 645] | 541 [418, 664] |
| dog-stand | 981 [977, 985] | 953 [931, 975] | 963 [955, 971] | 964 [939, 989] | 977 [970, 985] |
| dog-trot | 861 [772, 950] | 813 [747, 878] | 823 [753, 892] | 772 [640, 904] | 875 [842, 907] |
| dog-walk | 935 [927, 944] | 895 [868, 922] | 902 [894, 909] | 938 [933, 943] | 914 [899, 929] |
| humanoid-run | 194 [182, 207] | 189 [173, 205] | 236 [213, 259] | 182 [169, 196] | 204 [177, 231] |
| humanoid-stand | 916 [886, 945] | 716 [341, 1092] | 913 [895, 932] | 876 [794, 958] | 904 [886, 921] |
| humanoid-walk | 651 [590, 713] | 756 [695, 817] | 700 [553, 846] | 683 [541, 824] | 621 [585, 658] |
| IQM | 0.808 [0.726, 0.88] | 0.789 [0.65, 0.868] | 0.805 [0.667, 0.893] | 0.783 [0.663, 0.882] | 0.795 [0.659, 0.894] |
| Median | 0.729 [0.655, 0.808] | 0.732 [0.595, 0.816] | 0.73 [0.619, 0.826] | 0.717 [0.606, 0.823] | 0.724 [0.61, 0.825] |
| Mean | 0.729 [0.665, 0.79] | 0.7 [0.6, 0.795] | 0.724 [0.629, 0.814] | 0.718 [0.619, 0.809] | 0.72 [0.619, 0.811] |

*Table 28.* **DMC-Hard (Output Design).** Final performance at 1M environment steps averaged over 3 seeds. The [bracketed values] represent a 95% bootstrap confidence interval. The aggregate mean, median and interquartile mean are computed over the default reward.

| Task | SimbaV2 | MSE Loss | No Reward Scaling | No Return Bounding | Hard Target |
|---|---|---|---|---|---|
| dog-run | 562 [516, 608] | 545 [450, 639] | 478 [402, 554] | 617 [537, 696] | 676 [654, 699] |
| dog-stand | 981 [977, 985] | 976 [959, 993] | 967 [956, 978] | 969 [958, 981] | 980 [971, 989] |
| dog-trot | 861 [772, 950] | 841 [796, 886] | 737 [614, 859] | 884 [803, 964] | 848 [763, 934] |
| dog-walk | 935 [927, 944] | 905 [883, 928] | 925 [915, 935] | 922 [899, 945] | 928 [891, 964] |
| humanoid-run | 194 [182, 207] | 173 [154, 200] | 237 [181, 293] | 182 [154, 209] | 209 [159, 260] |
| humanoid-stand | 916 [886, 945] | 786 [612, 960] | 879 [821, 936] | 851 [744, 958] | 928 [920, 937] |
| humanoid-walk | 651 [590, 713] | 729 [577, 880] | 754 [643, 865] | 706 [563, 849] | 645 [602, 689] |
| IQM | 0.808 [0.728, 0.881] | 0.78 [0.62, 0.877] | 0.778 [0.655, 0.871] | 0.812 [0.69, 0.901] | 0.814 [0.706, 0.902] |
| Median | 0.729 [0.655, 0.809] | 0.705 [0.57, 0.836] | 0.715 [0.61, 0.817] | 0.731 [0.621, 0.84] | 0.746 [0.641, 0.847] |
| Mean | 0.729 [0.665, 0.79] | 0.708 [0.586, 0.813] | 0.712 [0.626, 0.792] | 0.733 [0.632, 0.824] | 0.746 [0.648, 0.833] |

*Table 29.* **DMC-Hard (Training Design).** Final performance at 1M environment steps averaged over 3 seeds. The [bracketed values] represent a 95% bootstrap confidence interval. The aggregate mean, median and interquartile mean are computed over the default reward.

| Task | SimbaV2 | No LR Decay | $s_{init}$ : 1 | $s_{scale}$ : 1 | $\alpha_{init}$ : 0.5 | $\alpha_{scale}$ : 1 |
|---|---|---|---|---|---|---|
| dog-run | 562 [516, 608] | 516 [407, 625] | 586 [550, 621] | 543 [488, 598] | 556 [420, 691] | 546 [493, 599] |
| dog-stand | 981 [977, 985] | 957 [933, 980] | 976 [970, 982] | 972 [964, 979] | 951 [912, 989] | 979 [969, 989] |
| dog-trot | 861 [772, 950] | 770 [663, 878] | 836 [756, 916] | 850 [798, 901] | 870 [842, 898] | 814 [692, 937] |
| dog-walk | 935 [927, 944] | 927 [919, 935] | 938 [921, 955] | 949 [937, 960] | 924 [903, 945] | 931 [921, 942] |
| humanoid-run | 194 [182, 207] | 221 [165, 277] | 187 [178, 196] | 194 [177, 211] | 193 [170, 217] | 182 [172, 192] |
| humanoid-stand | 916 [886, 945] | 932 [916, 948] | 874 [812, 936] | 823 [749, 897] | 900 [877, 924] | 918 [892, 944] |
| humanoid-walk | 651 [590, 713] | 706 [590, 822] | 624 [592, 656] | 610 [589, 630] | 697 [583, 812] | 622 [611, 633] |
| IQM | 0.808 [0.725, 0.879] | 0.798 [0.658, 0.895] | 0.783 [0.708, 0.851] | 0.766 [0.687, 0.838] | 0.819 [0.685, 0.892] | 0.781 [0.642, 0.891] |
| Median | 0.729 [0.655, 0.808] | 0.716 [0.616, 0.822] | 0.719 [0.646, 0.794] | 0.711 [0.634, 0.782] | 0.724 [0.62, 0.833] | 0.715 [0.603, 0.823] |
| Mean | 0.729 [0.664, 0.791] | 0.719 [0.623, 0.809] | 0.718 [0.656, 0.777] | 0.706 [0.644, 0.767] | 0.728 [0.627, 0.819] | 0.714 [0.61, 0.81] |

## K.4. Myosuite

*Table 30.* **Myosuite (Input Design).** Final performance at 1M environment steps averaged over 3 seeds. The [bracketed values] represent a 95% bootstrap confidence interval. The aggregate mean, median and interquartile mean are computed over the default reward.

| Task | SimbaV2 | No L2 Normalize | No Shifting | $c_{shift} : 1$ | Resize Projection |
|---|---|---|---|---|---|
| myo-key-turn | 1.0 [1.0, 1.0] | 1.0 [1.0, 1.0] | 1.0 [1.0, 1.0] | 1.0 [1.0, 1.0] | 1.0 [1.0, 1.0] |
| myo-key-turn-hard | 0.62 [0.427, 0.813] | 0.325 [-0.009, 0.659] | 0.25 [-0.044, 0.544] | 0.8 [0.661, 0.939] | 0.85 [0.752, 0.948] |
| myo-obj-hold | 1.0 [1.0, 1.0] | 1.0 [1.0, 1.0] | 1.0 [1.0, 1.0] | 1.0 [1.0, 1.0] | 1.0 [1.0, 1.0] |
| myo-obj-hold-hard | 0.98 [0.954, 1.006] | 0.975 [0.926, 1.024] | 0.975 [0.926, 1.024] | 1.0 [1.0, 1.0] | 0.975 [0.926, 1.024] |
| myo-pen-twirl | 1.0 [1.0, 1.0] | 1.0 [1.0, 1.0] | 1.0 [1.0, 1.0] | 1.0 [1.0, 1.0] | 1.0 [1.0, 1.0] |
| myo-pen-twirl-hard | 0.93 [0.888, 0.972] | 0.9 [0.82, 0.98] | 0.875 [0.689, 1.061] | 0.975 [0.926, 1.024] | 0.8 [0.604, 0.996] |
| myo-pose | 1.0 [1.0, 1.0] | 1.0 [1.0, 1.0] | 1.0 [1.0, 1.0] | 1.0 [1.0, 1.0] | 1.0 [1.0, 1.0] |
| myo-pose-hard | 0.0 [0.0, 0.0] | 0.0 [0.0, 0.0] | 0.0 [0.0, 0.0] | 0.0 [0.0, 0.0] | 0.0 [0.0, 0.0] |
| myo-reach | 1.0 [1.0, 1.0] | 1.0 [1.0, 1.0] | 1.0 [1.0, 1.0] | 1.0 [1.0, 1.0] | 1.0 [1.0, 1.0] |
| myo-reach-hard | 0.94 [0.873, 1.007] | 0.95 [0.893, 1.007] | 0.9 [0.82, 0.98] | 0.925 [0.831, 1.019] | 0.9 [0.82, 0.98] |
| IQM | 0.99 [0.968, 1.0] | 0.98 [0.885, 1.0] | 0.98 [0.86, 1.0] | 1.0 [0.955, 1.0] | 0.975 [0.925, 1.0] |
| Median | 0.845 [0.78, 0.925] | 0.815 [0.695, 0.935] | 0.805 [0.68, 0.92] | 0.875 [0.765, 0.975] | 0.86 [0.75, 0.955] |
| Mean | 0.847 [0.782, 0.906] | 0.815 [0.702, 0.915] | 0.8 [0.682, 0.905] | 0.87 [0.77, 0.953] | 0.852 [0.75, 0.938] |

*Table 31.* **Myosuite (Output Design).** Final performance at 1M environment steps averaged over 3 seeds. The [bracketed values] represent a 95% bootstrap confidence interval. The aggregate mean, median and interquartile mean are computed over the default reward.

| Task | SimbaV2 | MSE Loss | No Reward Scaling | No Return Bounding | Hard Target |
|---|---|---|---|---|---|
| myo-key-turn | 1.0 [1.0, 1.0] | 1.0 [1.0, 1.0] | 1.0 [1.0, 1.0] | 1.0 [1.0, 1.0] | 1.0 [1.0, 1.0] |
| myo-key-turn-hard | 0.62 [0.427, 0.813] | 0.2 [-0.192, 0.592] | 0.76 [0.66, 0.86] | 0.225 [-0.153, 0.603] | 0.675 [0.417, 0.933] |
| myo-obj-hold | 1.0 [1.0, 1.0] | 1.0 [1.0, 1.0] | 1.0 [1.0, 1.0] | 1.0 [1.0, 1.0] | 1.0 [1.0, 1.0] |
| myo-obj-hold-hard | 0.98 [0.954, 1.006] | 0.933 [0.868, 0.999] | 0.98 [0.941, 1.019] | 1.0 [1.0, 1.0] | 0.975 [0.926, 1.024] |
| myo-pen-twirl | 1.0 [1.0, 1.0] | 0.667 [0.013, 1.32] | 1.0 [1.0, 1.0] | 1.0 [1.0, 1.0] | 1.0 [1.0, 1.0] |
| myo-pen-twirl-hard | 0.93 [0.888, 0.972] | 0.867 [0.605, 1.128] | 0.98 [0.941, 1.019] | 0.9 [0.82, 0.98] | 0.9 [0.82, 0.98] |
| myo-pose | 1.0 [1.0, 1.0] | 1.0 [1.0, 1.0] | 0.8 [0.408, 1.192] | 1.0 [1.0, 1.0] | 1.0 [1.0, 1.0] |
| myo-pose-hard | 0.0 [0.0, 0.0] | 0.0 [0.0, 0.0] | 0.0 [0.0, 0.0] | 0.0 [0.0, 0.0] | 0.0 [0.0, 0.0] |
| myo-reach | 1.0 [1.0, 1.0] | 1.0 [1.0, 1.0] | 1.0 [1.0, 1.0] | 1.0 [1.0, 1.0] | 1.0 [1.0, 1.0] |
| myo-reach-hard | 0.94 [0.873, 1.007] | 0.9 [0.787, 1.013] | 0.88 [0.766, 0.994] | 0.925 [0.831, 1.019] | 0.925 [0.831, 1.019] |
| IQM | 0.99 [0.968, 1.0] | 0.944 [0.712, 1.0] | 0.985 [0.931, 1.0] | 0.985 [0.87, 1.0] | 0.98 [0.93, 1.0] |
| Median | 0.845 [0.78, 0.93] | 0.77 [0.58, 0.94] | 0.85 [0.74, 0.96] | 0.79 [0.68, 0.93] | 0.845 [0.74, 0.955] |
| Mean | 0.847 [0.783, 0.906] | 0.757 [0.613, 0.887] | 0.84 [0.746, 0.922] | 0.805 [0.685, 0.912] | 0.848 [0.745, 0.938] |

*Table 32.* **Myosuite (Training Design).** Final performance at 1M environment steps averaged over 3 seeds. The [bracketed values] represent a 95% bootstrap confidence interval. The aggregate mean, median and interquartile mean are computed over the default reward.

| Task | SimbaV2 | No LR Decay | $s_{init} : 1$ | $s_{scale} : 1$ | $\alpha_{init} : 0.5$ | $\alpha_{scale} : 1$ |
|---|---|---|---|---|---|---|
| myo-key-turn | 1.0 [1.0, 1.0] | 1.0 [1.0, 1.0] | 1.0 [1.0, 1.0] | 0.9 [0.704, 1.096] | 1.0 [1.0, 1.0] | 1.0 [1.0, 1.0] |
| myo-key-turn-hard | 0.62 [0.427, 0.813] | 0.65 [0.345, 0.955] | 0.74 [0.585, 0.895] | 0.69 [0.502, 0.878] | 0.675 [0.581, 0.769] | 0.85 [0.662, 1.038] |
| myo-obj-hold | 1.0 [1.0, 1.0] | 1.0 [1.0, 1.0] | 1.0 [1.0, 1.0] | 1.0 [1.0, 1.0] | 1.0 [1.0, 1.0] | 1.0 [1.0, 1.0] |
| myo-obj-hold-hard | 0.98 [0.954, 1.006] | 1.0 [1.0, 1.0] | 0.95 [0.897, 1.003] | 0.98 [0.954, 1.006] | 0.95 [0.893, 1.007] | 1.0 [1.0, 1.0] |
| myo-pen-twirl | 1.0 [1.0, 1.0] | 0.75 [0.26, 1.24] | 1.0 [1.0, 1.0] | 1.0 [1.0, 1.0] | 1.0 [1.0, 1.0] | 1.0 [1.0, 1.0] |
| myo-pen-twirl-hard | 0.93 [0.888, 0.972] | 0.775 [0.451, 1.099] | 0.89 [0.81, 0.97] | 0.88 [0.816, 0.944] | 0.925 [0.831, 1.019] | 1.0 [1.0, 1.0] |
| myo-pose | 1.0 [1.0, 1.0] | 1.0 [1.0, 1.0] | 1.0 [1.0, 1.0] | 1.0 [1.0, 1.0] | 1.0 [1.0, 1.0] | 1.0 [1.0, 1.0] |
| myo-pose-hard | 0.0 [0.0, 0.0] | 0.0 [0.0, 0.0] | 0.0 [0.0, 0.0] | 0.0 [0.0, 0.0] | 0.0 [0.0, 0.0] | 0.0 [0.0, 0.0] |
| myo-reach | 1.0 [1.0, 1.0] | 1.0 [1.0, 1.0] | 1.0 [1.0, 1.0] | 1.0 [1.0, 1.0] | 1.0 [1.0, 1.0] | 1.0 [1.0, 1.0] |
| myo-reach-hard | 0.94 [0.873, 1.007] | 0.925 [0.831, 1.019] | 0.97 [0.928, 1.012] | 0.91 [0.848, 0.972] | 0.875 [0.752, 0.998] | 0.9 [0.82, 0.98] |
| IQM | 0.99 [0.968, 1.0] | 0.985 [0.875, 1.0] | 0.99 [0.964, 1.0] | 0.982 [0.948, 1.0] | 0.975 [0.905, 1.0] | 1.0 [0.97, 1.0] |
| Median | 0.845 [0.78, 0.925] | 0.84 [0.7, 0.94] | 0.865 [0.785, 0.935] | 0.84 [0.77, 0.92] | 0.85 [0.735, 0.945] | 0.875 [0.77, 0.98] |
| Mean | 0.847 [0.782, 0.906] | 0.81 [0.698, 0.908] | 0.855 [0.792, 0.913] | 0.836 [0.77, 0.896] | 0.842 [0.742, 0.928] | 0.875 [0.772, 0.96] |

## K.5. Humanoid Bench

*Table 33.* **HumanoidBench (Input Design).** Final performance at 1M environment steps averaged over 3 seeds. The [bracketed values] represent a 95% bootstrap confidence interval. The aggregate mean, median and interquartile mean are computed over the default reward.

| Task | SimbaV2 | No L2 Normalize | No Shifting | $c_{shift} : 1$ | Resize Projection |
|---|---|---|---|---|---|
| h1-balance-hard-v0 | 143 [128, 157] | 113 [62, 164] | 94 [80, 109] | 89 [74, 104] | 150 [105, 194] |
| h1-balance-simple-v0 | 723 [651, 795] | 760 [666, 854] | 512 [236, 789] | 799 [744, 854] | 760 [658, 861] |
| h1-crawl-v0 | 946 [933, 959] | 901 [870, 932] | 838 [813, 864] | 948 [927, 968] | 884 [723, 1044] |
| h1-hurdle-v0 | 202 [167, 236] | 177 [168, 187] | 172 [162, 183] | 191 [160, 222] | 218 [152, 285] |
| h1-maze-v0 | 313 [287, 340] | 337 [313, 361] | 302 [230, 374] | 341 [337, 345] | 338 [311, 365] |
| h1-pole-v0 | 791 [785, 797] | 746 [722, 769] | 759 [754, 764] | 784 [781, 786] | 805 [795, 816] |
| h1-reach-v0 | 3850 [3272, 4427] | 4564 [3107, 6020] | 3263 [2743, 3783] | 5415 [4354, 6475] | 3722 [2105, 5339] |
| h1-run-v0 | 415 [307, 524] | 212 [208, 217] | 218 [196, 240] | 365 [115, 614] | 585 [377, 792] |
| h1-sit-hard-v0 | 679 [548, 811] | 493 [227, 758] | 721 [499, 944] | 757 [667, 847] | 667 [402, 933] |
| h1-sit-simple-v0 | 875 [870, 880] | 829 [783, 874] | 803 [745, 860] | 869 [858, 880] | 890 [872, 909] |
| h1-slide-v0 | 487 [404, 571] | 518 [499, 537] | 497 [483, 511] | 520 [467, 573] | 435 [318, 551] |
| h1-stair-v0 | 493 [467, 518] | 477 [460, 494] | 475 [433, 517] | 540 [480, 600] | 515 [504, 525] |
| h1-stand-v0 | 814 [770, 857] | 821 [786, 856] | 835 [833, 837] | 848 [845, 851] | 764 [661, 867] |
| h1-walk-v0 | 845 [840, 850] | 675 [560, 789] | 832 [814, 850] | 842 [830, 853] | 845 [835, 855] |
| IQM | 0.799 [0.684, 0.907] | 0.709 [0.525, 0.881] | 0.708 [0.509, 0.908] | 0.833 [0.635, 0.997] | 0.808 [0.61, 0.977] |
| Median | 0.781 [0.69, 0.862] | 0.698 [0.566, 0.851] | 0.698 [0.552, 0.85] | 0.801 [0.639, 0.944] | 0.801 [0.631, 0.934] |
| Mean | 0.776 [0.704, 0.847] | 0.711 [0.589, 0.833] | 0.7 [0.578, 0.825] | 0.791 [0.66, 0.921] | 0.779 [0.653, 0.905] |

*Table 34.* **HumanoidBench (Output Design).** Final performance at 1M environment steps averaged over 3 seeds. The [bracketed values] represent a 95% bootstrap confidence interval. The aggregate mean, median and interquartile mean are computed over the default reward.

| Task | SimbaV2 | MSE Loss | No Reward Scaling | No Return Bounding | Hard Target |
|---|---|---|---|---|---|
| h1-balance-hard-v0 | 143 [128, 157] | 89 [73, 106] | 92 [82, 101] | 136 [117, 155] | 173 [133, 213] |
| h1-balance-simple-v0 | 723 [651, 795] | 765 [664, 865] | 779 [680, 877] | 832 [825, 839] | 739 [553, 924] |
| h1-crawl-v0 | 946 [933, 959] | 945 [921, 969] | 953 [934, 971] | 942 [896, 989] | 938 [913, 964] |
| h1-hurdle-v0 | 202 [167, 236] | 243 [241, 245] | 140 [105, 174] | 192 [132, 252] | 207 [186, 228] |
| h1-maze-v0 | 313 [287, 340] | 324 [282, 367] | 273 [220, 327] | 320 [291, 350] | 339 [309, 370] |
| h1-pole-v0 | 791 [785, 797] | 774 [773, 776] | 724 [625, 823] | 796 [791, 800] | 787 [786, 788] |
| h1-reach-v0 | 3850 [3272, 4427] | 5044 [2733, 7355] | 4032 [2583, 5480] | 4396 [2059, 6732] | 3984 [2231, 5738] |
| h1-run-v0 | 415 [307, 524] | 485 [199, 771] | 359 [150, 568] | 360 [234, 487] | 341 [175, 508] |
| h1-sit-hard-v0 | 679 [548, 811] | 611 [350, 873] | 729 [648, 811] | 727 [681, 773] | 684 [362, 1005] |
| h1-sit-simple-v0 | 875 [870, 880] | 869 [864, 875] | 871 [859, 883] | 868 [863, 872] | 871 [870, 871] |
| h1-slide-v0 | 487 [404, 571] | 352 [260, 445] | 410 [313, 506] | 534 [484, 583] | 581 [550, 613] |
| h1-stair-v0 | 493 [467, 518] | 512 [470, 554] | 450 [382, 517] | 510 [495, 526] | 515 [495, 536] |
| h1-stand-v0 | 814 [770, 857] | 742 [621, 862] | 775 [684, 865] | 751 [570, 933] | 645 [251, 1038] |
| h1-walk-v0 | 845 [840, 850] | 851 [840, 863] | 742 [556, 928] | 845 [834, 855] | 838 [835, 842] |
| IQM | 0.799 [0.684, 0.907] | 0.778 [0.594, 0.95] | 0.745 [0.584, 0.896] | 0.819 [0.623, 0.977] | 0.778 [0.572, 0.965] |
| Median | 0.781 [0.692, 0.863] | 0.762 [0.614, 0.917] | 0.722 [0.605, 0.864] | 0.776 [0.636, 0.933] | 0.772 [0.614, 0.923] |
| Mean | 0.776 [0.704, 0.848] | 0.767 [0.637, 0.894] | 0.735 [0.629, 0.842] | 0.787 [0.658, 0.915] | 0.77 [0.64, 0.897] |

*Table 35.* **HumanoidBench (Training Design).** Final performance at 1M env steps averaged over 3 seeds. The [bracketed values] represent a 95% bootstrap confidence interval. The aggregate mean, median and interquartile mean are computed over the default reward.

| Task | SimbaV2 | No LR Decay | $s_{init}:1$ | $s_{scale}:1$ | $\alpha_{init}:0.5$ | $\alpha_{scale}:1$ |
|---|---|---|---|---|---|---|
| h1-balance-hard-v0 | 143 [128, 157] | 118 [86, 150] | 139 [119, 159] | 139 [116, 162] | 152 [140, 164] | 139 [119, 159] |
| h1-balance-simple-v0 | 723 [651, 795] | 812 [758, 867] | 815 [794, 836] | 813 [793, 833] | 763 [620, 907] | 732 [645, 820] |
| h1-crawl-v0 | 946 [933, 959] | 955 [933, 977] | 956 [947, 965] | 946 [929, 963] | 933 [911, 954] | 962 [954, 970] |
| h1-hurdle-v0 | 202 [167, 236] | 150 [64, 235] | 228 [218, 238] | 188 [159, 218] | 203 [187, 219] | 209 [204, 214] |
| h1-maze-v0 | 313 [287, 340] | 283 [150, 417] | 347 [339, 355] | 337 [318, 356] | 367 [352, 382] | 333 [324, 342] |
| h1-pole-v0 | 791 [785, 797] | 736 [625, 847] | 774 [760, 788] | 767 [736, 798] | 791 [777, 804] | 792 [785, 798] |
| h1-reach-v0 | 3850 [3272, 4427] | 5986 [3542, 8430] | 4295 [3423, 5167] | 4988 [4251, 5724] | 4312 [2719, 5905] | 5495 [3782, 7207] |
| h1-run-v0 | 415 [307, 524] | 424 [172, 675] | 357 [277, 436] | 351 [246, 457] | 234 [209, 259] | 337 [171, 503] |
| h1-sit-hard-v0 | 679 [548, 811] | 678 [382, 975] | 686 [575, 796] | 734 [604, 863] | 507 [257, 757] | 702 [493, 911] |
| h1-sit-simple-v0 | 875 [870, 880] | 845 [793, 897] | 875 [869, 881] | 877 [871, 882] | 868 [852, 885] | 870 [868, 872] |
| h1-slide-v0 | 487 [404, 571] | 388 [268, 507] | 486 [420, 553] | 525 [475, 574] | 447 [380, 514] | 533 [512, 554] |
| h1-stair-v0 | 493 [467, 518] | 512 [499, 525] | 507 [491, 523] | 504 [483, 525] | 513 [505, 521] | 503 [468, 538] |
| h1-stand-v0 | 814 [770, 857] | 603 [434, 773] | 751 [697, 804] | 789 [700, 879] | 793 [707, 878] | 799 [720, 877] |
| h1-walk-v0 | 845 [840, 850] | 843 [832, 855] | 843 [837, 849] | 839 [828, 850] | 834 [829, 839] | 827 [802, 851] |
| IQM | 0.799 [0.683, 0.908] | 0.764 [0.578, 0.93] | 0.796 [0.688, 0.901] | 0.823 [0.709, 0.927] | 0.725 [0.532, 0.917] | 0.817 [0.615, 0.978] |
| Median | 0.781 [0.69, 0.862] | 0.769 [0.599, 0.912] | 0.776 [0.7, 0.866] | 0.792 [0.705, 0.876] | 0.746 [0.595, 0.893] | 0.802 [0.64, 0.938] |
| Mean | 0.776 [0.704, 0.847] | 0.754 [0.625, 0.883] | 0.781 [0.711, 0.852] | 0.789 [0.719, 0.861] | 0.745 [0.619, 0.872] | 0.792 [0.659, 0.916] |

