# OpenReview forum: "Hyperspherical Normalization for Scalable Deep Reinforcement Learning"
_ICML.cc/2025/Conference — ICML 2025 spotlightposter_

### Official Review · Reviewer_fmAi · 2025-02-27

**Overall Recommendation:** 4

**Summary:**

The main claim of the paper is that a novel architecture (SimbaV2) can improve the scaling of RL algorithms. The benefits of using SAC and SimbaV2 are demonstrated across a variety of domains.

**Claims And Evidence:**

I find that the authors do an excellent job at demonstrating the empirical performance benefits of using SimbaV2 and SAC. The empirical evidence is overwhelming, across a large number of domains.

However, the authors also make some more general claims about scaling (SimbaV2 enables better scaling), that are not as clearly justified.
1. SimbaV2 improves scaling of **RL algorithms** $\rightarrow$ The authors only demonstrate this claim for SAC.

2. SimbaV2 improves **scaling** of RL algorithms $\rightarrow$ The authors only demonstrate that SimbaV2 scales better in width than the original Simba, and not SAC. The authors also don’t demonstrate this claim for depth (increasing the number of blocks). In the UTD dimension, the scaling results are less convincing. Looking at Appendix I, we see that often an increase in UTD offers no performance gain (and sometimes slightly harms). From many previous works, we know that this is still an improvement (naively increasing UTD often harms performance), but this isn’t exactly a strong defense of UTD scaling. As a baseline, the authors only compare against one setting of resets and the original Simba.

3. X design change is beneficial $\rightarrow$  The authors make a number of design choices to improve over the original Simba (listed in Section 4). Many of these design choices are not defended or motivated outside of empirical performance (Table 2). Including the use of a distributional critic, which is specifically outlined as a key contribution in the abstract of the paper.

**Essential References Not Discussed:**

Hussing, Marcel, et al. "Dissecting deep rl with high update ratios: Combatting value divergence." arXiv preprint arXiv:2403.05996 (2024).

This paper analyses high UTD in more detail and also suggests l2 normalization as a solution.

**Experimental Designs Or Analyses:**

I find the analysis limited. In Figure 4, the authors demonstrate that SimbaV2 improves over the original Simba in a number of dimensions (feature norm, parameter norm, gradient norm, effective LR). However, it’s unclear to me whether any of these dimensions are necessary for stability, scaling, or better performance. Furthermore, it’s unclear which design choices contribute to these changes.

**Methods And Evaluation Criteria:**

The SOTA claims are well-defended: the authors cover a lot of popular benchmarks and compare against many SOTA algorithms.

As mentioned above, some of the evaluation of the broader claims made by the authors fall short (only a single base algorithm, limited baselines for analytical claims).

**Other Comments Or Suggestions:**

Typos
- Output Preidiction Line395
- ”resize” (inverse quotes) Line434

**Other Strengths And Weaknesses:**

**Strengths:**

So far, the construction of this review process has mostly forced me to list weaknesses, but I think the paper is a valuable contribution to the community. As a scientific paper, the authors make some overclaims, but as an empirical contribution, the authors are introducing a powerful, widely applicable algorithm that has been thoroughly tested.

Given the impact we have seen from the same types of papers in the model-based space (DreamerV3, TD-MPC2), I see no reason why a model-free algorithm should be any less useful. Furthermore, many of the design choices are likely widely applicable to other algorithms, although I believe the authors could do a better job defending that claim.

**Weaknesses:**

I learned very little from this paper, other than the fact that it works well.

**Questions For Authors:**

Empirical:
- Does SimbaV2 scale in depth? (i.e., adding more blocks).
- Does SimbaV2 work with other algorithms (besides SAC).

Analytical:
- Why does SimbaV2 (and each of its components) improve performance?
- What was the motivation behind the design choices, especially the more subtle ones like Linear + Scalar?

Other reviewers may disagree, but I would also be satisfied if some of the broader claims were reduced, and the contribution of the paper was reduced to just the specific SimbaV2+SAC algorithm.

**Relation To Broader Scientific Literature:**

SOTA algorithm is obviously beneficial. UTD and scaling laws are very popular topics in the community right now. I think the authors could do a better job at comparing against existing work in this dimension.

**Theoretical Claims:**

N/A.

---

> ### Author Rebuttal · Authors · 2025-04-01
>
> Dear reviewer fmAi,
>
> Thank you for your thoughtful and constructive feedback. We address your concerns below and would be happy to clarify further.
>
> > **Question 4.1**
> The scalability claims are not fully justified:
> (1) Only SAC is tested, (2) only width scaling is shown, (3) UTD scaling shows limited benefit.
>
> (1) Algorithm: To evaluate generality beyond SAC, we ran SimbaV2 with DDPG on DMC-Hard and HBench-Hard (UTD = 2):
>
> | Method   | DMC-Hard | HBench-Hard |
> |-|-|-|
> | SimbaV2  | 0.636  | 0.693 |
> | Simba    | 0.649  | 0.445 |
> | MLP      | 0.149  | 0.115 |
>
> SimbaV2 performs comparably to Simba on DMC-Hard and significantly better on HBench-Hard, confirming its effectiveness beyond SAC.
>
> (2) Depth scaling: We conducted additional experiments on DMC-Hard by varying the critic depth (1, 2, 4, 8 layers) using 5 random seeds.
>
> | Method  | 1 | 2 | 4 | 8 |
> |-|-|-|-|-|
> | SimbaV2  | 0.525 | 0.729 | 0.740 | **0.743** |
> | Simba    | 0.512 | 0.706 | 0.675 | -  |
>
> Unlike SimbaV1, which degraded with depth, SimbaV2 improves consistently. This supports our view that effective regularization enables stable scaling in depth. Due to a limited time, HBench-Hard results are underway and will be included in the final manuscript.
>
> (3) UTD scaling: While simple tasks (e.g., Cartpole-Balance) saturate quickly, more complex tasks such as HumanoidBench-Hard continue improving with higher UTD (see Fig. 6). Notably, prior work [1, 2] shows that high UTD often degrades performance unless combined with weight reinitialization. SimbaV2 maintains stable learning at high UTD without reinitialization, which we believe is a key contribution.
>
> - [1] Sample-Efficient Reinforcement Learning by Breaking the Replay Ratio Barrier, ICLR'23.
> - [2] Bigger, Regularized, Optimistic: scaling for compute and sample-efficient continuous control, NeurIPS'24.
>
> > **Question 4.2**
> It's unclear whether the reported metrics are necessary for stability and scaling.
> Many architectural choices in Section 4 are not motivated beyond empirical results.
>
> The architectural decisions in SimbaV2 are based on training instabilities observed in SimbaV1 and supported by prior literature.
>
> 1. Feature Norm: TD learning introduces an implicit bias toward increasing feature norm [1], which can lead to overfitting and a reduction in feature rank. Both effects are closely linked to loss of plasticity during training [2, 3].
> 2. Parameter Norm, Gradient Norm, and ELR: As parameter norms grow, the ELR decreases, impeding gradient flow and leading to stagnation in learning [4]. This dynamic was observed in SimbaV1, particularly in the encoder.
>
> These insights motivated the following designs in SimbaV2:
> 1. Hyperspherical normalization to control feature norm
> 2. Weight projection to constrain parameter norm.
> 3. Distributional critic with reward scaling to stabilize gradient norm.
>
> Together, these components maintain a stable ELR across layers, preserve plasticity, and eliminate the need for weight reinitialization. We agree that this motivation was not clearly described and will revise the introduction to clarify these points.
>
> - [1] DR3: Value-Based Deep RL Requires Explicit Regularization, ICLR'22.
> - [2] Understanding and Preventing Capacity Loss in RL, ICLR'22.
> - [3] Dissecting Deep RL with High Update Ratios: Combatting Value Divergence, RLC'24.
> - [4] Normalization and effective learning rates in RL, NeurIPS'24.
>
>
> > **Question 4.3**
> > The use of a distributional critic is not defended.
>
> As shown in [this ablation](https://drive.google.com/file/d/15nCDgxhbPL20LnrPCa6k_gNYfoR5smzO/view?usp=sharing), removing the distributional critic and reward scaling degrades performance and leads to a sharp decline in gradient norm, which in turn reduces the ELR. This supports its role in stabilizing training and preserving plasticity.
>
> > **Question 4.4**
> What was the motivation behind Linear + Scaler?
>
> Linear + Scaler serves two purposes:
> (i) weight projection controls parameter norm and ELR, and
> (ii) The learnable scalar amplifies important features.
>
> Without the scalar, projection limits expressivity; without projection, training becomes unstable. Their combination is essential for balancing stability and flexibility.
>
> > **Question 4.5**
> I learned very little from this paper, other than the fact that it works well.
>
> Thank you for the candid feedback. Beyond empirical gains, our central insight is that stabilizing norm dynamics enables scalable, stable, and sample-efficient RL without relying on resets—a common workaround in the RL community. SimbaV2 offers a principled, architecture-level solution to this challenge.
>
> We also believe these insights may transfer to other domains. For example, recent [NLP work](https://arxiv.org/abs/2503.19206) suggests long pretraining reduces model plasticity. SimbaV2’s techniques may inform the design of more robust architectures for large-scale training in other domains.

---

> > ### Comment · Reviewer_fmAi · 2025-04-02
> >
> > Thank you for the response.
> >
> > While I still feel that the analysis could be strengthened, the results are significant and convincing. I have increased my score.

---

### Official Review · Reviewer_4564 · 2025-03-09

**Overall Recommendation:** 3

**Summary:**

This paper introduces SimbaV2, an RL architecture that improves scalability and stability in deep RL. The authors use hyperspherical normalization to control weight and feature norm growth, alongside distributional value estimation with reward scaling to maintain stable gradients. Using SAC as the base algorithm, SimbaV2 outperforms existing RL methods across a wide range of continuous control tasks.

## Update after Rebuttal

After the rebuttal, I am keeping my positive score for this paper.

**Claims And Evidence:**

Most claims made in the submission are supported by clear and convincing evidence. However, the claim that this leads to 'scalable' RL remains incorrect, as scaling limits are reached very quickly (See Fig. 5).

**Essential References Not Discussed:**

-

**Experimental Designs Or Analyses:**

Experimental design and analysis is sound.

**Methods And Evaluation Criteria:**

The methods and evaluation criteria are good, although I would have liked to see experiments also in another domain or algorithm, as it feels somewhat repetitive to Simba V1.

**Other Comments Or Suggestions:**

-

**Other Strengths And Weaknesses:**

Strengths: Strong performance on the state-based continuous control benchmarks. Well written paper.

Weaknesses: This paper only feels like a minor step up of SimbaV1. I would have liked to see experiments in a different set of environments, such as pixel-based control. However, at this level of performance a step up is also hard to accomplish, so there is still a valid contribution (Especially if we compare it to Supervised Learning papers which make minuscule step ups.)

**Questions For Authors:**

Why have the authors not attempted to do a performance analysis in at least a few pixel-based Mujoco tasks? How would the authors implement their techniques on CNN's ?

**Relation To Broader Scientific Literature:**

This paper is at the forefront of performance-based RL in continuous control tasks.

**Theoretical Claims:**

-

---

> ### Author Rebuttal · Authors · 2025-04-01
>
> Dear Reviewer 4564,
>
> Thank you for suggesting future research direction! We respond to each of your comments below and are happy to clarify further if needed.
>
>
> > **Question 3.1**
> The claim that this leads to 'scalable' RL remains incorrect, as scaling limits are reached very quickly (See Fig. 5).
>
> We appreciate your concern. While it is true that performance on DMC-Hard saturates earlier, this is largely due to task-specific ceilings rather than architectural constraints. Most tasks in DMC-Hard (except humanoid-run) reach near-optimal scores around 4.5 million parameters. In contrast, HBench-Hard continues to benefit from increased capacity up to 17.8 million parameters, demonstrating substantial headroom for scaling.
>
> This level of parameter scalability is atypical in RL. Standard SAC or DDPG architectures often use around 2 million parameters, and larger models tend to degrade performance. In contrast, SimbaV2 scales consistently and robustly, without the need for reinitialization or tuning.
>
> Similarly, in Fig. 6, compute scaling on DMC-Hard saturates around UTD = 4 due to task limitations, but HBench-Hard continues improving up to UTD = 8. This behavior highlights SimbaV2’s stable training dynamics under both model and compute scaling, which we believe justifies its claim to scalability within the current RL landscape.
>
>
> > **Question 3.2**
> This paper only feels like a minor step up of SimbaV1. I would have liked to see experiments in a different set of environments, such as pixel-based control.
>
> > Why have the authors not attempted to do a performance analysis in at least a few pixel-based Mujoco tasks? How would the authors implement their techniques on CNN's ?
>
> Thank you for this suggestion. We agree that applying SimbaV2 to pixel-based control is a promising direction. However, extending our techniques to convolutional architectures presents unique challenges. In CNNs, a shared kernel operates over overlapping spatial regions, making it nontrivial to enforce hyperspherical constraints on both features and weights, as we do with MLPs.
>
> One possible approach is to project each C-dimensional fiber of the feature map and kernel onto a hypersphere, preserving the underlying normalization principle. However, this would require considerable architectural tuning, which we consider an important direction for future work.
>
> That said, we believe the core ideas behind SimbaV2, *controlling feature and parameter norms, and maintaining stable effective learning rates*, can extend to vision-based RL. We look forward to exploring this as a future work.

---

> > ### Comment · Reviewer_4564 · 2025-04-08
> >
> > Thanks for the additional clarifications.
> >
> > "
> > We appreciate your concern. While it is true that performance on DMC-Hard saturates earlier, this is largely due to task-specific ceilings rather than architectural constraints. Most tasks in DMC-Hard (except humanoid-run) reach near-optimal scores around 4.5 million parameters. In contrast, HBench-Hard continues to benefit from increased capacity up to 17.8 million parameters, demonstrating substantial headroom for scaling.
> > "
> >
> > I think it would be beneficial to the paper to more explicitly mention this (Abstract, Introduction).
> >
> > My concerns have been solved and I will keep my (positive) score! Looking forward to further improvements in this area.

---

### Official Review · Reviewer_ZJka · 2025-03-12

**Overall Recommendation:** 4

**Summary:**

This paper proposes SimbaV2, an improved version of Simba, by replacing several key components of Simba with a scale-preserved l2-normalization (i.e., hyperspherical normalization), distributional value function approximation, reward scaling, etc. The authors presents a comprehensive experimental study with 57 continuous control tasks across 4 domains, against a wide range of existing online RL methods. The experimental results demonstrate the superiority of SimbaV2 in scaling with larger networks and higher UTDs effectively, along with careful analysis on learning dynamics metrics and design choices.


## update after rebuttal

I've read all the other reviewers' comments, and the authors' rebuttal provided additional experimental evidence that addressed my questions well. Therefore, I will keep the rating.

**Claims And Evidence:**

Most claims made in this paper are well supported with experimental results. The necessity or effects of the design changes “Linear → Linear + Scaler” and “Residual Connection → LERP” are not supported with direct evidence (Please correct me if I missed them).

**Essential References Not Discussed:**

Most essential related works are included. There are some other related papers not included:

- Deep Policy Gradient Methods Without Batch Updates, Target Networks, or Replay Buffers. arXiv 2411.15370
- Adam on Local Time: Addressing Nonstationarity in RL with Relative Adam Timesteps. arXiv 2412.17113
- Improving Deep Reinforcement Learning by Reducing the Chain Effect of Value and Policy Churn. arXiv 2409.04792

**Experimental Designs Or Analyses:**

The experiments are well organized with comprehensive content, from the analysis of learning dynamics, to performance comparison and then the ablation study of design choices. Training and inference cost are also included.

**Methods And Evaluation Criteria:**

The proposed methods are mainly for better addressing non-stationarity in observation, intermediate features and network output (i.e., target values).

The evaluation criteria are diverse in this work. The performance metrics are well normalized. The analysis metrics (including weight norm, feature norm, ELR) also make sense in the context.

**Other Comments Or Suggestions:**

None

**Other Strengths And Weaknesses:**

#### Strengths

- SimbaV2 significantly improves the stability of learning dynamics in terms of metrics like ELR, especially for more challenging tasks in HBench-Hard.
- SimbaV2 is free of adding various learning-based/optimization-based regularization to address the non-stationarity issue, which offers better generality and feasibility.
- The comprehensive experiments can provide useful references for related studies.

#### Weaknesses

- The performance of SimbaV2 is based on SAC, leaving its effects on PPO and DQN-variants unknown (the authors also mentioned this).
- The environments used in the experiments include only proprioceptive observations (correct me if I misunderstand it)

**Questions For Authors:**

1. Are there experiments for evaluating the effects of the design choices “the inverted bottleneck MLP”, “Residual Connection → LERP” and “Linear → Linear + Scaler” in this work? It seems that I did not find direct empirical evidence for these ones.
2. In Simba, the observation is centered and rescaled by running mean and std, and in SimbaV2, the observation is rescaled by the L2 norm. If there is a variant of SimbaV2, that only replaces the L2-norm rescaling by the running std rescaling, how would it performs differently with SimbaV2?

**Relation To Broader Scientific Literature:**

The proposed architecture is of potential to be a norm for DRL, beyond the SAC base algorithm considered in this paper. The proposed methods and the experimental study can provide a useful reference to the related studies on learning under non-stationarity, e.g., continual RL, streaming/incremental RL.

**Theoretical Claims:**

Not applicable.

---

> ### Author Rebuttal · Authors · 2025-04-01
>
> Dear Reviewer ZJka,
>
> Thank you for your constructive feedback and positive support! We respond to each of your points below and would be happy to clarify further if needed.
>
>
> > **Q2.1:**
> > The necessity or effects of the design changes “Linear → Linear + Scaler” and “Residual Connection → LERP” are not supported with direct evidence.
> > Are there experiments for evaluating the effects of the design choices “the inverted bottleneck MLP”, “Residual Connection → LERP” and “Linear → Linear + Scaler” in this work?
>
> Thank you for highlighting this. Below we provide both the rationale and supporting evidence for these design choices.
>
> 1. **Linear → Linear + Scaler:**
> This modification enables two key benefits:
> (i) weight projection ensures control over parameter norm and effective learning rate, and
> (ii) the learnable scalar allows selective emphasis of important features.
>
> A standard linear layer without projection leads to unbounded parameter growth. On the other hand, projection without a scalar severely limits representational capacity. The Linear + Scaler combination is therefore necessary to balance norm control with expressivity.
>
> 2. **Residual Connection → LERP:**
>    Since all features in SimbaV2 are normalized to lie on a hypersphere, standard residual connections are no longer applicable. The closest analog would be fixed-$\alpha$ interpolation (e.g., $\alpha$=0.5). We conducted an ablation to compare **LERP** vs. fixed-$\alpha$ residual:
>
> | Method      | DMC-Hard | HBench-Hard |
> |-------------|----------|-------------|
> | LERP (learnable, $\alpha_{init}$=1/(L+1)) | 0.729 $\pm$ 0.065 | 0.946 $\pm$ 0.089 |
> | Residual (fixed, $\alpha_{init}$=0.5) | 0.687 $\pm$ 0.092 | 0.843 $\pm$ 0.123 |
>
> LERP achieves higher performance in both benchmarks. We attribute this to two factors: its learnable mixing coefficient, which enables adaptive interpolation, and its initialization ($\alpha$=1/(L+1)), which biases early training toward identity mapping, enhancing stability.
>
>
> > **Q2.2:**
> > There are some other related papers not included:
> > - Deep Policy Gradient Methods Without Batch Updates, Target Networks, or Replay Buffers. arXiv 2411.15370
> > - Adam on Local Time: Addressing Nonstationarity in RL with Relative Adam Timesteps. arXiv 2412.17113
> > - Improving Deep Reinforcement Learning by Reducing the Chain Effect of Value and Policy Churn. arXiv 2409.04792
>
> Thank you for pointing out these relevant works. We appreciate the suggestions and will include them in the related work section. We have also added the following relevant works:
>
> - Is High Variance Unavoidable in RL?., Bjorck et al., ICLR 2022
> - Understanding, Predicting, and Better Resolving Q-Value Divergence in Offline RL., Yue et al., NeurIPS 2023
> - Mixtures of Experts Unlock Parameter Scaling for Deep RL., Ceron et al., ICML 2024
> - Dissecting Deep RL with High Update Ratios., Hussing et al., RLC 2024
> - Don’t Flatten, Tokenize!., Sokar et al., ICLR 2025
>
>
> > **Q2.3:**
> > The performance of SimbaV2 is based on SAC, leaving its effects on PPO and DQN-variants unknown.
>
> Thank you for highlighting this. While extending to PPO and DQN is a valuable direction, these algorithms present challenges: PPO is inherently on-policy and difficult to scale to high update-to-data (UTD) ratios, while DQN is restricted to discrete action spaces.
>
> To assess the generality of SimbaV2 beyond SAC, we conducted additional experiments using DDPG, a widely adopted off-policy algorithm for continuous control:
>
> | Method   | DMC-Hard | HBench-Hard |
> |----------|----------|-------------|
> | SimbaV2  | 0.636 $\pm$ 0.087 | 0.693 $\pm$ 0.119 |
> | Simba    | 0.649 $\pm$ 0.089 | 0.445 $\pm$ 0.101 |
> | MLP      | 0.149 $\pm$ 0.034 | 0.115 $\pm$ 0.047 |
>
> In DMC-Hard, SimbaV2 performs competitively with Simba, both significantly outperforming the MLP baseline. In the more challenging HBench-Hard benchmark, SimbaV2 shows clear improvements over Simba, indicating enhanced stability and generalization beyond SAC.
>
>
> > **Q2.4:**
> > In Simba, the observation is centered and rescaled by running mean and std, and in SimbaV2, the observation is rescaled by the L2 norm. If there is a variant of SimbaV2 that only replaces the L2-norm rescaling with running std rescaling, how would it perform?
>
> We appreciate the opportunity to clarify. There appears to be a misunderstanding: both Simba and SimbaV2 use running mean and standard deviation (RSNorm) for input normalization.
>
> The key difference is architectural. SimbaV2 replaces LayerNorm with L2 normalization for internal features, not for input observations. This change targets internal stability and effective learning rates, while the input normalization strategy rem

---

### Official Review · Reviewer_R7VW · 2025-03-17

**Overall Recommendation:** 4

**Summary:**

The paper introduces SimbaV2, an RL architecture that stabilizes training and improves scalability through hyperspherical normalization and distributional value estimation with reward scaling. Built on Soft Actor-Critic (SAC), it achieves state-of-the-art performance across 57 continuous control tasks and scales effectively with increased model size and compute. Experiments confirm its stability, outperforming existing RL methods without requiring periodic weight reinitialization.

**Claims And Evidence:**

The paper provides strong empirical evidence to support its claims. The experiments are well-structured, covering scalability, stability, and performance comparisons across 57 continuous control tasks. The ablation studies confirm the importance of hyperspherical normalization and reward scaling, reinforcing the paper’s core contributions.

**Essential References Not Discussed:**

N/A

**Experimental Designs Or Analyses:**

I checked all the experimental designs and analyses, and there are no issues.

**Methods And Evaluation Criteria:**

The proposed methods and evaluation criteria are well-suited for the problem. Hyperspherical normalization and reward scaling directly address RL's instability and overfitting issues, making them relevant for scaling RL effectively. The evaluation is thorough, using continuous control tasks across four standard benchmarks (MuJoCo, DMC, MyoSuite, HumanoidBench), ensuring broad applicability. Comparisons with strong baselines and ablation studies further validate the approach.

**Other Comments Or Suggestions:**

It would be helpful if the authors could provide more theoretical justification or intuitive explanations for why previous methods struggle to scale while SimbaV2 does. Specifically, a deeper discussion on why hyperspherical normalization stabilizes training and why it eliminates the need for weight reinitialization would strengthen the paper’s contributions.

Could the authors disclose the computational resources used for this project, including hardware specifications and training time? This information would be valuable for the community to better understand the practicality and scalability of the approach.

**Other Strengths And Weaknesses:**

The paper's experimental design is a strong point, with thorough evaluations across multiple benchmarks and well-structured ablation studies that clearly isolate the contributions of each component. The writing is also clear and well-organized, making it easy to follow the ideas and their significance. I enjoyed reading this paper.

**Questions For Authors:**

N/A

**Relation To Broader Scientific Literature:**

The paper builds on prior work in RL scalability, regularization techniques, and normalization methods. It extends ideas from weight decay, dropout, and layer normalization, commonly used in supervised learning, by introducing hyperspherical normalization to stabilize RL training. The distributional value estimation approach aligns with prior work on distributional RL, enhancing gradient stability. Additionally, the paper addresses challenges seen in periodic weight reinitialization methods by providing an alternative that scales without overfitting. It contributes to the broader discussion on scaling laws in RL, challenging the notion that increasing model size and computation necessarily leads to instability.

**Theoretical Claims:**

This paper does not contain theoretical proofs.

---

> ### Author Rebuttal · Authors · 2025-04-01
>
> Dear reviewer R7VW,
>
> Thank you for your thoughtful and constructive feedback. We address your concerns in detail below and would be happy to clarify any remaining questions.
>
>
> > **Question 1.1**
> It would be helpful if the authors could provide more theoretical justification or intuitive explanations for why previous methods struggle to scale while SimbaV2 does.
>
> > Why does hyperspherical normalization stabilize training and eliminate the need for weight reinitialization?
>
> We appreciate this insightful question. Our use of hyperspherical normalization is motivated by empirical observations in SimbaV1, where we identified unstable growth in key quantities—including feature norm, parameter norm, gradient norm, and effective learning rate per layer (Fig. 4). These instabilities are known to impair generalization and reduce plasticity in RL.
>
> Specifically:
> 1. **Feature Norm:** Prior work [1] shows that TD loss induces an implicit bias toward growing feature norms, which can cause overfitting and reduced feature rank. This is often driven by a few dominant dimensions, leading to loss of plasticity [2]. Techniques such as feature norm regularization [1] and hyperspherical normalization [3,4] were proposed to mitigate this.
> 2. **Parameter Norm and Effective Learning Rate:**  As parameter norms grow, the effective learning rate (gradient norm divided by parameter norm) declines, which hampers gradient flow and learning dynamics [5,6]. In SimbaV1, we observed this specifically in the encoder, where the effective learning rate collapses over time (Fig. 4e). Simply increasing the global learning rate is not viable, as other layers (e.g., the predictor) may already be operating at high effective rates.
>
> SimbaV2 addresses these challenges through the following design choices:
> 1. **Feature Norm:** Hyperspherical normalization to control feature norm growth.
> 2. **Parameter Norm:** Weight projection onto a hypersphere to control parameter norm.
> 3. **Gradient Norm:** Distributional critic with reward scaling to regulate gradient norm
>
> Together, these mechanisms ensure stable learning dynamics (i.e., stable effective learning rate across layers) and sustained plasticity, thereby eliminating the need for weight reinitialization. We recognize that our original draft did not clearly explain these intuitions and will revise the introduction accordingly to highlight these design motivations.
>
> - [1] DR3: Value-Based Deep Reinforcement Learning Requires Explicit Regularization, Kumar et al, ICLR'22.
> - [2] Understanding and Preventing Capacity Loss in Reinforcement Learning, Lyle et al, ICLR'22.
> - [3] Is high variance unavoidable in rl? a case study in continuous control, Bjorck et al, ICLR'22.
> - [4] Dissecting Deep RL with High Update Ratios: Combatting Value Divergence, Hussing et al, RLC'24.
> - [5] Loss of plasticity in deep continual learning, Dohare et al, Nature'24.
> - [6] Normalization and effective learning rates in reinforcement learning, Lyle et al, NeurIPS'24.
>
>
> > **Question 1.2**
> Could the authors disclose the computational resources used for this project, including hardware specifications and training time?
>
> All experiments were conducted using NVIDIA RTX 3090 GPUs and an AMD EPYC 7402 24-Core Processor.
>
> For wall-clock time, a single run of SimbaV2 with UTD=2 typically takes 1.6 hours. This varies by environment: around 1.0 hour for simpler tasks like Cartpole (no collisions, minimal joints) and up to 2.5 hours for more complex tasks like Dog (many joints and intricate interactions).

---

> > ### Comment · Reviewer_R7VW · 2025-04-02
> >
> > Thanks for the authors' response, which solved my concerns. I have increased the rating.

---

### Decision · Program_Chairs · 2025-05-01

**Decision:**

Accept (spotlight poster)

**Comment:**

There is a broad consensus that the paper deserves acceptance due to the strong empirical evidence to support its claims and SOTA performance on a range of tasks. The paper is seen as a useful step towards scaling existing deep RL approaches. A minor weakness is that this paper might be related on many aspects to SimbaV1 (reviewer 4564).